# How Does Variance Shape the Regret in Contextual Bandits?

**Zeyu Jia**
Massachusetts Institute of Technology
zyjia@mit.edu

**Jian Qian**
Massachusetts Institute of Technology
jianqian@mit.edu

**Alexander Rakhlin**
Massachusetts Institute of Technology
rakhlin@mit.edu

**Chen-Yu Wei**
University of Virginia
chenyu.wei@virginia.edu

## Abstract

We consider realizable contextual bandits with general function approximation, investigating how small reward variance can lead to better-than-minimax regret bounds. Unlike in minimax regret bounds, we show that the eluder dimension $d_{\text{elu}}$—a measure of the complexity of the function class—plays a crucial role in variance-dependent bounds. We consider two types of adversary:

- Weak adversary: The adversary sets the reward variance before observing the learner's action. In this setting, we prove that a regret of $\Omega(\sqrt{\min\{A, d_{\text{elu}}\}\Lambda} + d_{\text{elu}})$ is unavoidable when $d_{\text{elu}} \leq \sqrt{AT}$, where $A$ is the number of actions, $T$ is the total number of rounds, and $\Lambda$ is the total variance over $T$ rounds. For the $A \leq d_{\text{elu}}$ regime, we derive a nearly matching upper bound $\tilde{\mathcal{O}}(\sqrt{A\Lambda} + d_{\text{elu}})$ for the special case where the variance is revealed at the beginning of each round.

- Strong adversary: The adversary sets the reward variance after observing the learner's action. We show that a regret of $\Omega(\sqrt{d_{\text{elu}}\Lambda} + d_{\text{elu}})$ is unavoidable when $\sqrt{d_{\text{elu}}\Lambda} + d_{\text{elu}} \leq \sqrt{AT}$. In this setting, we provide an upper bound of order $\tilde{\mathcal{O}}(d_{\text{elu}}\sqrt{\Lambda} + d_{\text{elu}})$.

Furthermore, we examine the setting where the function class additionally provides distributional information of the reward, as studied by Wang et al. (2024). We demonstrate that the regret bound $\tilde{\mathcal{O}}(\sqrt{d_{\text{elu}}\Lambda} + d_{\text{elu}})$ established in their work is unimprovable when $\sqrt{d_{\text{elu}}\Lambda} + d_{\text{elu}} \leq \sqrt{AT}$. However, with a slightly different definition of the total variance and with the assumption that the reward follows a Gaussian distribution, one can achieve a regret of $\tilde{\mathcal{O}}(\sqrt{A\Lambda} + d_{\text{elu}})$.

## 1  Introduction

We consider the contextual bandit problem that models repeated interactions between the learner and the environment. In each round, the learner chooses an action based on the received context, and observes the reward of the chosen action. Algorithms designed to achieve minimax regret guarantees under a variety of statistical assumptions and computational models have been extensively studied (Auer et al., 2002; Dudik et al., 2011; Agarwal et al., 2012, 2014; Foster and Rakhlin, 2020; Xu and Zeevi, 2020; Simchi-Levi and Xu, 2022; Zhang, 2022).

However, these algorithms often fail to leverage the potentially benign nature of the environment. In this work, we refine the regret bound by considering the variance of the reward. Such variance-dependent regret bounds, also known as second-order regret bounds, have been primarily studied under linear function approximation (Zhang et al., 2021; Kim et al., 2022; Zhao et al., 2023). Notably,

Zhao et al. (2023) first established a near-optimal $\tilde{\mathcal{O}}(d\sqrt{\Lambda} + d)$ regret bound for linear contextual bandits, where $d$ represents the feature dimension and $\Lambda$ the sum of the reward variances.

For contextual bandits with general function approximation, the recent work by Wang et al. (2024) obtained a second-order bound assuming access to a model class containing distributional information about the reward. They showed a regret bound of $\tilde{\mathcal{O}}(\sqrt{d_{\mathrm{elu}}\Lambda\log|\mathcal{M}|} + d_{\mathrm{elu}}\log|\mathcal{M}|)$, where $|\mathcal{M}|$ is the size of the model class, and $d_{\mathrm{elu}} = d_{\mathrm{elu}}(\mathcal{M})$ is its eluder dimension. As noted in Wang et al. (2024), the dependence on $d_{\mathrm{elu}}$ is undesirable, and when the number of actions $A$ is much smaller than $d_{\mathrm{elu}}$, this bound can potentially be improved. This conjecture is supported by Foster and Rakhlin (2020), who showed that the upper bound $\tilde{\mathcal{O}}(\sqrt{AT\log|\mathcal{F}|} + A\log|\mathcal{F}|)$ is achievable regardless of the eluder dimension, where $T \geq \Lambda$ is the number of rounds, and $|\mathcal{F}|$ is the size of the function class containing mean reward information. It is tempting to conjecture that the regret can smoothly scale with $\Lambda$, resulting in a bound of $\tilde{\mathcal{O}}(\sqrt{A\Lambda\log|\mathcal{F}|} + A\log|\mathcal{F}|)$. Such variance-dependent regret bounds that replace the dependence on the number of rounds $T$ by the total variance $\Lambda$ have been shown in multi-armed bandits (Audibert et al., 2009), linear bandits (Ito and Takemura, 2023), and linear contextual bandits (Zhao et al., 2023).

In this paper, we show, surprisingly, that the aforementioned conjecture is not true in general. Specifically, for any $A$ and any $d_{\mathrm{elu}} \leq \sqrt{AT}$, one can construct a problem instance with lower bound $\Omega(\sqrt{\min\{A, d_{\mathrm{elu}}\}\Lambda} + d_{\mathrm{elu}})$ with $\log|\mathcal{F}| = \mathcal{O}(\log T)$ and $d_{\mathrm{elu}}(\mathcal{F}) = d_{\mathrm{elu}}$. This rules out the possibility of achieving $\tilde{\mathcal{O}}(\sqrt{A\Lambda\log|\mathcal{F}|} + A\log|\mathcal{F}|)$ for all $A$ because we can always make $d_{\mathrm{elu}} = \sqrt{AT}$, resulting in a lower bound $\Omega(\sqrt{AT})$ even with $\Lambda = 0$. Our primary goal is to design algorithms that achieve the near-optimal regret bound $\tilde{\mathcal{O}}(\sqrt{\min\{A, d_{\mathrm{elu}}\}\Lambda\log|\mathcal{F}|} + d_{\mathrm{elu}}\log|\mathcal{F}|)$.

The lower bound $\sqrt{\min\{A, d_{\mathrm{elu}}\}\Lambda} + d_{\mathrm{elu}}$ indicates that the complexity of contextual bandits arises from two parts. The first part accounts for local estimation of the true function, where the complexity is due to the variance of the reward and the local structure of the function set around the ground truth function $f^\star \in \mathcal{F}$. This results in the term $\sqrt{\min\{A, d_{\mathrm{elu}}\}\Lambda}$, with the leading coefficient $\min\{A, d_{\mathrm{elu}}\}$ corresponding to the decision-estimation coefficient (Foster et al., 2021). The second part accounts for global search for the true function, in which the complexity is due to a more global structure of the function set and can be quantified by the *disagreement* among the functions. The complexity of this part scales with $d_{\mathrm{elu}}$, even when $A = 2$. The contribution of the global part is usually overshadowed by the local part when only considering regret bounds with constant variance. Our work highlights its role by studying the variance-dependent bound. The fundamental role of disagreement is also discussed in Foster et al. (2020) for gap-dependent bounds. Specifically, they also showed that when trying to obtain the gap-dependent bound that has logarithmic dependence on $T$, the complexity must scale with some disagreement measure over the function class, instead of just the number of actions.

The previous work by Wei et al. (2020) also derived a set of results for general contextual bandits showing that the tight second-order regret bound is strictly larger than merely replacing the $T$ in the minimax bound by the second-order error. They consider the more general agnostic setting but the tight regret bounds are only established for the $|\mathcal{F}| = 1$ case. Their result for $|\mathcal{F}| > 1$ can be applied to our setting, though it only gives highly sub-optimal bounds. Overall, our work refines theirs in the realizable setting.

When preparing our camera-ready version, the concurrent work of Pacchiano (2024), which studied exactly the same problem as ours, was posted on arXiv. We provide a comparison with their work in Section 3.4. More related works are discussed in Appendix A.

## 2 Preliminaries

A contextual bandit problem consists of a context space $\mathcal{X}$, an action space $\mathcal{A}$, the total number of rounds $T$ and a class of functions $\mathcal{F} \subset [0,1]^{\mathcal{X}\times\mathcal{A}}$. At round $t$, the learner observes a context $x_t \in \mathcal{X}$, then makes a decision $a_t \in \mathcal{A}$ based on the current context $x_t$ and history, and observes a reward $r_t$. We assume that these rewards $r_t$ are given by

$$r_t = f^\star(x_t, a_t) + \epsilon_t, \tag{1}$$

where $f^\star : \mathcal{X} \times \mathcal{A} \to [0,1]$ is some function unknown to the learner, and $\epsilon_t$ are independent zero-mean random variables with variance $\sigma_t^2$ such that $r_t \in [0,1]$.[1] We denote $\Lambda = \sum_{t=1}^{T} \sigma_t^2$. The learner

---

[1] We assume bounded noise for simplicity. The extension to 1-sub-Gaussian noise is straightforward.

aims to optimize the total expected regret $R_T$, defined as

$$R_T = \sum_{t=1}^{T} \left( \max_{a \in \mathcal{A}} f^\star(x_t, a) - f^\star(x_t, a_t) \right).$$

We make the following realizability assumption:

**Assumption 2.1** (Function Realizability). *Assume that $f^\star$ in Eq. (1) satisfies $f^\star \in \mathcal{F}$.*

We finish this section with the definition of eluder dimension:

**Definition 2.1** (Eluder Dimension (Russo and Van Roy, 2014)). *For function class $\mathcal{F}$ defined on space $\mathcal{Z}$, we define the eluder dimension of $\mathcal{F}$ at scale $\alpha \geq 0$, denoted by $d_{elu}(\mathcal{F}; \alpha)$, as the length of the longest sequence of tuples $(z_1, f_1, f_1'), ..., (z_m, f_m, f_m') \in \mathcal{Z} \times \mathcal{F} \times \mathcal{F}$ such that there exists $\alpha_0 \geq \alpha$ making the following hold for all $i = 1, .., m$:*

$$\sum_{j < i} (f_i(z_j) - f_i'(z_j))^2 \leq \alpha_0^2, \text{ and } |f_i(z_i) - f_i'(z_i)| > \alpha_0.$$

*Throughout the paper, if $\alpha$ is not specified, we take the default value $\alpha = 1/T^2$. We also omit the dependence on $\mathcal{F}$ when it is clear from the context.*

## 3 Results Overview

We describe our three settings in the following three subsections and summarize the results in Table 1. In the following, $\mathcal{F}$ denotes the function class that only contains reward mean information, and $\mathcal{M}$ the model class that contains reward distribution information.

### 3.1 Weak Adversary Case with Variance Revealing (Section 4)

First, we consider the case where the adversary is *weak*. This means that the variance $\sigma_t$ only depends on the history up to round $t - 1$, which aligns with the standard "adaptive adversary" assumption. For this case, we show that for any $A$ and $d$, one can find an instance of contextual problem problem with $|\mathcal{F}| \leq \sqrt{AT}$, such that the regret is at least

$$\Omega \left( \sqrt{\min\{A, d_{\text{elu}}\} \Lambda} + \min\{d_{\text{elu}}, \sqrt{AT}\} \right). \tag{2}$$

For upper bounds in the weak adversary case, we focus on the regime $A \leq d_{\text{elu}} \leq \sqrt{AT}$, where the lower bound can be written as $\Omega(\sqrt{A\Lambda} + d_{\text{elu}})$.[2] While our ultimate goal is to obtain a nearly matching upper bound $\tilde{\mathcal{O}}(\sqrt{A\Lambda \log |\mathcal{F}|} + d_{\text{elu}} \log |\mathcal{F}|)$, we have not achieved it yet in full generality. In Section 4, we provide an algorithm which operates under the assumption that the variance $\sigma_t$ is revealed to the learner at the beginning of round $t$, and show that it achieves the matching upper bound. An initial attempt to remove this assumption is discussed in Section 7, where we show that when $\sigma_t \in \{0, 1\}$ for all $t$ and $\sigma_t$ is revealed to the learner at the *end* of round $t$, the matching upper bound can also be achieved.

### 3.2 Strong Adversary Case (Section 5)

Next, we consider the case where the adversary is *strong*. This means the adversary can decide $\sigma_t$ *after* seeing the action $a_t$ chosen by the learner at round $t$. In this case, the lower bound becomes

$$\Omega \left( \min \left\{ \sqrt{d_{\text{elu}}\Lambda} + d_{\text{elu}}, \sqrt{AT} \right\} \right). \tag{3}$$

The difference with Eq. (2) is that the scaling in front of $\Lambda$ changes from $\min\{A, d_{\text{elu}}\}$ to $d_{\text{elu}}$. This shows the even more crucial role of eluder dimension in the strong adversary case. For this setting, we give an upper bound of $\tilde{\mathcal{O}}(d_{\text{elu}} \sqrt{\Lambda \log |\mathcal{F}|} + d_{\text{elu}} \log |\mathcal{F}|)$, which is off from the lower bound by a $\sqrt{d_{\text{elu}}}$ factor along with other logarithmic factors.

---

[2] $A \leq d_{\text{elu}} \leq \sqrt{AT}$ is a more challenging and elusive regime, and we focus our study here. When $d_{\text{elu}} > \sqrt{AT}$, we can use SquareCB (Foster and Rakhlin, 2020) to achieve the tight bound $\sqrt{AT}$; when $d_{\text{elu}} < A$, we can use the algorithm in Section 5 to get $d_{\text{elu}} \sqrt{\Lambda} + d_{\text{elu}}$, which is tight up to a $\sqrt{d_{\text{elu}}}$ factor.

Table 1: Results overview. $d_{\mathrm{elu}}$ refers to either $d_{\mathrm{elu}}(\mathcal{F})$ or $d_{\mathrm{elu}}(\mathcal{M})$ depending on the settings. In Section 4 and Section 5, $\Lambda := \sum_{t=1}^{T} \sigma_t^2$. In Section 6, $\Lambda_\infty := \sum_{t=1}^{T} \max_a \sigma_{M^\star}(x_t, a)^2$ and $\Lambda_\circ := \sum_{t=1}^{T} \sigma_{M^\star}(x_t, a_t)^2$, where $M^\star \in \mathcal{M}$ is the underlying true model, and $\sigma_M(x, a)^2$ is the reward variance for $(x, a)$ predicted by model $M$.

**Notice**: For simplicity, this table only considers the case $A \leq d_{\mathrm{elu}}$, and all lower bounds should be further taken a minimum with $\sqrt{AT}$.

| | Upper bound (omitting $\log T$ factors) | Lower bound |
|---|---|---|
| Weak adversary + revealing $\sigma_t$ (Section 4) | $\sqrt{A\Lambda \log |\mathcal{F}|} + d_{\mathrm{elu}} \log |\mathcal{F}|$ (Corollary 4.1) | $\sqrt{A\Lambda} + d_{\mathrm{elu}}$ (Theorem 4.1) |
| Strong adversary (Section 5) | $d_{\mathrm{elu}} \sqrt{\Lambda \log |\mathcal{F}|} + d_{\mathrm{elu}} \log |\mathcal{F}|$ (Theorem 5.2) | $\sqrt{d_{\mathrm{elu}}\Lambda} + d_{\mathrm{elu}}$ (Theorem 5.1) |
| Learning with model class (Section 6) | $\sqrt{A\Lambda_\infty \log |\mathcal{M}|} + d_{\mathrm{elu}} \log |\mathcal{M}|$ for Gaussian noise (Theorem 6.1) | $\sqrt{A\Lambda_\infty} + d_{\mathrm{elu}}$ (Theorem 6.2) |
| Learning with model class (Section 6) | $\sqrt{d_{\mathrm{elu}}\Lambda_\circ \log |\mathcal{M}|} + d_{\mathrm{elu}} \log |\mathcal{M}|$ (Wang et al. (2024)) | $\sqrt{d_{\mathrm{elu}}\Lambda_\circ} + d_{\mathrm{elu}}$ (Theorem 6.3) |

## 3.3 Learning with a Model Class (Section 6)

In Section 6, we assume that the function class provides information on the distribution of the reward rather than just the mean. Such a function class is usually called a *model class*. More precisely, the learner is provided with a model class $\mathcal{M}$ that includes the true model $M^\star \in \mathcal{M}$ so that $M^\star(x, a)$ specifies reward distribution for the context-action pair $(x, a)$. Compared to the scenario studied in Section 4, here we do not require variance to be revealed to the learner. This becomes possible because with a model class, the learner can now obtain variance information (though not precise) through the context. Under the assumption that the noise is Gaussian, we provide an $\tilde{\mathcal{O}}(\sqrt{A\Lambda_\infty \log |\mathcal{M}|} + d_{\mathrm{elu}} \log |\mathcal{M}|)$ upper bound where $\Lambda_\infty = \sum_t \max_a \sigma_{M^\star}(x_t, a)^2$, and a matching lower bound, where $\sigma_M(x, a)$ is the reward variance for the context-action pair $(x, a)$ predicted by $M \in \mathcal{M}$.

The work of Wang et al. (2024) also studied second-order contextual bandits with a model class. They use $\Lambda_\circ = \sum_t \sigma_{M^\star}(x_t, a_t)^2$, i.e., the reward variance of the chosen actions, as the variance measure. They obtain $\tilde{\mathcal{O}}(\sqrt{d_{\mathrm{elu}}\Lambda_\circ \log |\mathcal{M}|} + d_{\mathrm{elu}} \log |\mathcal{M}|)$ upper bound. We show a nearly matching lower bound $\Omega(\min\{\sqrt{d_{\mathrm{elu}}\Lambda_\circ} + d_{\mathrm{elu}}, \sqrt{AT}\})$, similar to the lower bound for the strong adversary case studied in Section 5. The lower bound indicates that, in general, the bound of Wang et al. (2024) cannot be improved even when $A < d_{\mathrm{elu}}$.

## 3.4 Comparison with Pacchiano (2024)

The work of Pacchiano (2024) also studied variance-dependent bounds for realizable contextual bandits. They also consider two settings, which can be mapped to those in our Section 4 and Section 5, respectively. For the weak adversary setting with revealed $\sigma_t$ (Section 4), they give an upper bound of $\tilde{\mathcal{O}}(\sqrt{d_{\mathrm{elu}}\Lambda \log |\mathcal{F}|} + d_{\mathrm{elu}} \log |\mathcal{F}|)$,[3] which is incomparable to our $\tilde{\mathcal{O}}(\sqrt{A\Lambda \log |\mathcal{F}|} + d_{\mathrm{elu}} \log |\mathcal{F}|)$. However, a full picture of this setting can be obtained by combining their upper bound and our upper bound and the lower bound in Eq. (2). For the strong adversary setting (Section 5), they derive exactly the same upper bound as in our Theorem 5.2. Our work makes additional contribution in the lower bounds and the extension to the distributional setting (Section 6).

## 4 Weak Adversary Case with Variance Revealing

In this section, we consider cases where the variance $\sigma_t^2$ at round $t$ is given to the learner at the beginning of round $t$ together with the context $x_t$.

---

[3]Although Pacchiano (2024) presents this result by assuming $\sigma_t = \sigma$ for some known $\sigma$, it is straightforward to extend it to the case where $\sigma_t$ are different but revealed before each round, just like in our Section 4.3.

## 4.1 Lower Bound

The regret lower bound is shown with identical and known variance. The construction is similar to those in Wei et al. (2020). Concretely, we have the following theorem.

**Theorem 4.1** (Main lower bound). *For any integer $d, A \geq 2$, any positive real number $\sigma \in [0,1]$, and time $T > 0$, there exists a context space $\mathcal{X}$ and a contextual bandit problem $\mathcal{F} \subset (\mathcal{X} \times \mathcal{A} \to \mathbb{R})$ with eluder dimension $d_{elu}(0) \leq d$, action set $\mathcal{A}$ with $|\mathcal{A}| \leq A$, and variance $\sigma_t \leq \sigma$ for all $t \in [T]$ such that any algorithm will suffer a regret at least $\Omega(\sqrt{\sigma^2 \min\{A, d\}T} + \min\{d, \sqrt{AT}\})$.*

**Proof sketch.** The full proof is deferred to Appendix C. The two parts in the lower bound came from the following two different hardness: (1) The first part of the lower bound with $\Omega(\sqrt{\sigma^2 \min\{A, d\}T})$ is a natural lower bound with variance $\sigma$ due to estimation of the mean values. (2) For the second part, we consider the following function class. In this function class, there is a "good" action that serves as the default choice with a reward of 1/2 for all contexts. For each of the other $A - 1$ "bad" actions, for each context, there is one function that obtains a reward of 1 but obtains 0 for all the other contexts. When $d < \sqrt{AT}$, this function class forces the learner to guess for each context which action to choose. So even if the reward is deterministic, i.e., variance $\sigma = 0$, any learner would have to suffer a regret scaling with the number of contexts times the number of actions, which in total coincide with the eluder dimension. When $d \geq \sqrt{AT}$, the learner can simply commit to the "good" action and suffer $\sqrt{AT}$ but no better than this. □

This lower bound is rather surprising for the following consequences: (1) The most significant implication from this lower bound is that improving the minimax regret bound with the knowledge of the variance is only possible if $d < \sqrt{AT}$. (2) Even when $d < \sqrt{AT}$, any learner would have to pay for the eluder dimension as a lower-order term. These are non-trivial because the second-order bounds are usually obtained from changing Hoeffding concentration to Bernstein concentration which usually only scales the regret bounds by $\sigma$. This lower bound shows that the second-order contextual bandit is not one of the usual cases. In the next section, we will match this lower bound from the upper bound side by combining several algorithmic techniques.

## 4.2 Upper Bound with Known and Fixed Variance

Motivated by the lower bound in Theorem 4.1, we wonder whether there is an algorithm which can achieve a matching upper bound of $\tilde{\mathcal{O}}(\sqrt{\sigma^2 \min\{A, d\}T} + \min\{d, \sqrt{AT}\})$, if the learner is provided with information of variance at the beginning of each round. In this subsection, we answer this question affirmly. To begin with, we consider the case when all the variance are identical, i.e. $\sigma_1 = \sigma_2 = \cdots = \sigma_T = \sigma$, and $\sigma$ is given to the learner. Later (Section 4.3), we will discuss how to generalize this result to the case with nonidentical variances across different rounds.

We assume that $r_t = f^\star(x_t, a_t) + \epsilon_t \in [0, 1]$ and $\mathrm{Var}(\epsilon_t) \leq \sigma_t^2$ for every $1 \leq t \leq T$. Our results can be easily extended to subgaussian random noise (at the cost of a $\log T$ factor) since for such variables, with probability at least $1 - \delta$, $|\epsilon_t| \leq C\sqrt{\log(1/\delta)}$ for a constant $C$.

### 4.2.1 Algorithm and Analysis for Identical Variance

We first consider the case with identical variance, i.e. $\sigma_t^2 = \sigma^2$ for all $t \in [T]$. We propose Algorithm 1, and show that it has regret upper bound $\tilde{\mathcal{O}}(\sqrt{\sigma^2 AT \log |\mathcal{F}|} + d_{\mathrm{elu}} \log |\mathcal{F}|)$. The algorithm is adapted from SquareCB of Foster and Rakhlin (2020), but additionally maintains a confidence function set, and has mechanisms to learn faster when the functions in the confidence set has larger disagreement. It has the following elements:

**1. Restricting action set (Line 4)** At the beginning of round $t$ (Line 4), the learner restricts the action set to $\mathcal{A}_t$, which only includes those actions that is the best action of some functions in the function class $\mathcal{F}_t$. If we assume that $f^\star$ is always in the function class $\mathcal{F}_t$, by doing this we remove the unnecessary possibility of choosing actions that can never be the best action.

**2. Checking disagreement (Line 5-Line 7)** The next step of the algorithm is to check whether there is an action in $\mathcal{A}_t$ such that two functions in the function class have large value differences (Line 6). We called such actions "discriminative actions". Roughly speaking, we are seeking an action $a \in \mathcal{A}_t$ such that

$$\exists f, f' \in \mathcal{F}_t, \qquad |f(x_t, a) - f'(x_t, a)| \gtrsim \Delta \approx \sigma^2.$$

---

**Algorithm 1** VarCB (Variance-aware Contextual Bandits)

---

**Input:** $\delta \in [0, 1]$, $\sigma \in [1/AT, 1]$.

1: Let $L = \log \frac{|\mathcal{F}|T^2}{\delta\sigma^2}$ and $\Delta = \frac{\sigma^2}{11\sqrt{L}}$ and $\mathcal{F}_1 = \mathcal{F}$.

2: **for** $t = 1, 2\ldots,$ **do**

3:     Receive context $x_t$.

4:     Define $\mathcal{A}_t = \{a \in \mathcal{A} : \exists f \in \mathcal{F}_t, \ a \in \arg\max_{a\in\mathcal{A}} f(x_t, a)\}$

5:     Define

$$g_t(a) = \sup_{f, f' \in \mathcal{F}_t} \frac{|f(x_t, a) - f'(x_t, a)|}{\sqrt{1 + \sum_{\tau=1}^{t-1} w_\tau (f(x_\tau, a_\tau) - f'(x_\tau, a_\tau))^2}}, \qquad \forall a \in \mathcal{A}. \qquad (4)$$

6:     **if** $\max_{a\in\mathcal{A}_t} g_t(a) \geq \Delta$ **then**

7:         Choose action $a_t = \arg\max_{a\in\mathcal{A}_t} g_t(a)$ and receive $r_t$.

8:     **else**

9:         Call online regression oracle (Algorithm 4) with input $(\mathcal{F}_t, x_t)$ and obtain $f_t$.

10:        Let $b_t = \arg\max_{a\in\mathcal{A}_t} f_t(x_t, a)$ (pick an arbitrary maximizer if there are multiple).

11:        Draw $a_t \sim p_t$ and receive $r_t$, where

$$p_t(a) = \begin{cases} 0 & \text{if } a \in \mathcal{A} \setminus \mathcal{A}_t, \\ \frac{1}{|\mathcal{A}_t| + \gamma(f_t(x_t, b_t) - f_t(x_t, a))} & \text{if } a \in \mathcal{A}_t \setminus \{b_t\}, \\ 1 - \sum_{a' \neq a} p_t(a') & \text{if } a = b_t. \end{cases} \qquad \text{(inverse gap weighting)}$$

12:     Define $w_t = \min\left\{\frac{1}{\sigma^2}, \frac{1}{g_t(a_t)\sqrt{L}}\right\}$ and update the confidence set:

$$\mathcal{F}_{t+1} = \left\{ f \in \mathcal{F}_t : \sum_{\tau=1}^{t} w_\tau (f(x_\tau, a_\tau) - \hat{f}_{t+1}(x_\tau, a_\tau))^2 \leq 102L \right\}, \qquad (5)$$

$$\text{where } \hat{f}_{t+1} = \arg\min_{f\in\mathcal{F}_t} \sum_{\tau=1}^{t} w_\tau (f(x_\tau, a_\tau) - r_\tau)^2. \qquad (6)$$

---

If such an action exists, then the learner chooses this action at round $t$. By selecting such an action that can discriminate disagreed functions, the function set $\mathcal{F}_t$ can more quickly shrink. To prevent this action to incur overly large regret, it is important to perform Step 1 (Restricting action set). The regret incurred in rounds choosing discriminative actions is of order $\tilde{\mathcal{O}}(d_{\text{elu}} \log |\mathcal{F}|)$.

**3. Inverse gap weighting (Line 8-Line 11)** At round $t$, if there is no discriminative action, then the learner performs inverse gap weighting as in the SquareCB algorithm (Foster and Rakhlin (2020)). Inverse gap weighting requires the learner to have access to an online regression oracle that generates online estimations $f_t$ and ensures that the estimation error $\sum_t (f_t(x_t, a_t) - f^\star(x_t, a_t))^2$ is small. In the original SquareCB, the requirement for the online regression oracle is

$$R_{\mathsf{sq}} = \sum_{t=1}^{T} (f_t(x_t, a_t) - r_t)^2 - \sum_{t=1}^{T} (f^\star(x_t, a_t) - r_t)^2 \lesssim \log |\mathcal{F}|, \qquad \text{(F\&R's condition)}$$

which only allows for a $\sqrt{AT \log |\mathcal{F}|}$ regret bound that does not meet our goal. To improve this, we design an online regression oracle that ensures

$$R_{\mathsf{sq}} = \sum_{t\in\mathcal{T}_{\mathsf{IGW}}} (f_t(x_t, a_t) - r_t)^2 - \sum_{t\in\mathcal{T}_{\mathsf{IGW}}} (f^\star(x_t, a_t) - r_t)^2 \lesssim (\sigma^2 + \tilde{\Delta}) \log |\mathcal{F}|, \quad \text{(our condition)}$$

where $\mathcal{T}_{\mathsf{IGW}}$ is the set of rounds that we run inverse gap weighting (i.e., entering the else case in Line 8), and $\tilde{\Delta}$ is an upper bound for $\max_{a\in\mathcal{A}_t} \max_{f, f' \in \mathcal{F}_t} |f(x_t, a) - f'(x_t, a)|$, i.e., the maximum disagreement among the function set $\mathcal{F}_t$ for the context $x_t$. Thanks to Step 2, we only run inverse gap weighting when $\tilde{\Delta} \lesssim \Delta \approx \sigma^2$. Thus, with the refined $R_{\mathsf{sq}}$ guarantee and standard squareCB arguments, we can get a regret bound of order $\sigma\sqrt{AT \log |\mathcal{F}|}$ for the rounds in $\mathcal{T}_{\mathsf{IGW}}$.

The way to achieve "(our condition)" is an interesting part of our algorithm. A standard way to ensure F&R's condition is by aggregating over the function set through exponential weights. Exponential weights ensures $R_{\text{sq}} = \mathcal{O}(\log |\mathcal{F}|/\eta)$ as long as the functions to be aggregated are $\eta$-mixable. Thus, in order to show $R_{\text{sq}} = \mathcal{O}(\sigma^2 \log |\mathcal{F}|)$, we need to argue $\eta = \Omega(1/\sigma^2)$. However, because the potential range of $r_t$ is $[0, 1]$ even though the variance $\sigma^2$ and and the disagreement $\Delta$ are both much smaller than 1, the best mixability coefficient $\eta$ we can show for squared loss is still $\Theta(1)$.

To address this, we resort to the use of the Prod algorithm (Cesa-Bianchi and Lugosi, 2006) with a properly chosen surrogate loss to perform aggregation. This algorithm has a different second-order approximation for the loss compared to the exponential weight algorithm, which is crucial in obtaining the desired bound. The regret analysis is also no longer through mixability. Our online regression oracle is provided in Algorithm 4 in Appendix D. We remark without giving details that in the linear case, such a guarantee can also be obtained through Online Newton Step (Hazan et al., 2007).

**4. Updating function set (Line 12)**    After finishing selecting the action $a_t$ for round $t$, the learner updates the confidence function set $\mathcal{F}_t$ to prepare for the next round. The construction of the confidence set utilizes the idea of weighted regression that has been widely used in previous variance-aware or corruption-robust contextual bandit or RL algorithms (He et al., 2022; Zhao et al., 2023; Ye et al., 2023; Agarwal et al., 2023). This has the effect of controlling the relative importance of different samples and is crucial in controlling the regret incurred in Step 2.

By putting these building blocks together, we arrive at Algorithm 1. The regret of Algorithm 1 is described in Theorem 4.2, whose proof is deferred to Appendix D.

**Theorem 4.2.** *Algorithm 1 ensures with probability at least $1 - \delta$,*

$$\sum_{t=1}^{T} (\max_{a \in \mathcal{A}} f^\star(x_t, a) - f^\star(x_t, a_t)) = \tilde{\mathcal{O}}\left( \sqrt{\sigma^2 AT \log\left(|\mathcal{F}|/\delta\right)} + d_{elu} \log\left(|\mathcal{F}|/\delta\right) \right).$$

**Comparison with AdaCB of Foster et al. (2020)**    Our VarCB (Algorithm 1) shares some similarities with the AdaCB algorithm from Foster et al. (2020), which aims to achieve a $\tilde{\mathcal{O}}(\frac{d \log |\mathcal{F}|}{\text{GAP}})$ regret bound. Here, $d$ is a disagreement coefficient of $\mathcal{F}$, which takes the same role as our $d_{\text{elu}}$, and GAP represents the minimal reward gap between the best and second-best decisions. Specifically, both algorithms include a step to remove irrelevant actions (our Step 1). The action selection rule of AdaCB also depends on the amount of disagreement over the function class, which is superficially related to the if-else separation in VarCB. However, we find that the case separations in the two algorithms do not have a clear correspondence to each other, possibly due to the different objectives of the two algorithms. Also, the two algorithms operate under quite different settings: AdaCB works in the setting where the contexts are i.i.d., while VarCB allows for adversarial contexts. On the other hand, AdaCB is parameter-free, but VarCB requires the information of $\sigma$. Developing a more unified version for these two better-than-minimax algorithms is an interesting future direction.

### 4.3 Algorithm and Analysis for Heteroscedastic Noise

Next, we will discuss how to generalize our algorithm to heteroscedastic case, i.e. when the noise of different rounds are different. Based on the values of the variance, we classify each round into the following $(\log(AT) + 1)$ sets: if $\sigma_t \in [0, \frac{1}{AT}]$, we classify $t$ into $\mathcal{T}_0$, and for $\sigma_t \in (\frac{2^{i-1}}{AT}, \frac{2^i}{AT}]$, we classify $t$ into $\mathcal{T}_i$ for $2 \le i \le \log(AT)$, i.e., if $\sigma_t$ falls into the $i$-th intervals in the following,

$$\Sigma_0 = [0, \tfrac{1}{AT}], \quad \Sigma_1 = (\tfrac{1}{AT}, \tfrac{2}{AT}], \quad \Sigma_2 = (\tfrac{2}{AT}, \tfrac{4}{AT}], \quad \cdots, \quad \Sigma_{\log(AT)} = (1/2, 1], \quad (7)$$

we classify $t$ into $\mathcal{T}_i$. For each set $\mathcal{T}_i$, we maintain an algorithm $\mathscr{A}_i$ of Algorithm 1 in parallel. At the beginning at round $t$, when observing that $t \in \mathcal{T}_i$, only $\mathscr{A}_i$ is updated, while $\mathscr{A}_j$ remains the same for $j \ne i$. According to Theorem 4.2, we have for any $0 \le i \le \log T$,

$$\sum_{t \in [\mathcal{T}_i]} (\max_{a \in \mathcal{A}} f^\star(x_t, a) - f^\star(x_t, a_t)) = \tilde{\mathcal{O}}\left( \sqrt{A|\mathcal{T}_i| \cdot \left(\tfrac{2^i}{AT}\right)^2 \log |\mathcal{F}|} + d_{\text{elu}} \log |\mathcal{F}| \right),$$

we can bound the total regret by

$$\sum_{i=1}^{\log(AT)} \tilde{\mathcal{O}}\left( \sqrt{A|\mathcal{T}_i| \cdot \left(\tfrac{2^i}{AT}\right)^2 \log |\mathcal{F}|} + d_{\text{elu}} \log |\mathcal{F}| \right) = \tilde{\mathcal{O}}\left( \sqrt{A \sum_{i=1}^{T} \sigma_i^2 \log |\mathcal{F}|} + d_{\text{elu}} \log |\mathcal{F}| \right).$$

---

**Algorithm 2** Algorithm for Heteroscedastic Noise

---

**Input:**
1: Initialize instances $\mathscr{A}_i$ of VarCB (Algorithm 1) with $\sigma = \frac{2^i}{AT}$ for $0 \le i \le \log(AT)$.
2: **for** $t = 1 : T$ **do**
3:     Receive $\sigma_t \in [0, 1]$, and suppose that $\sigma_t \in \Sigma_i$, where $\Sigma_i$ is defined in Eq. (7).
4:     Receive context $x_t$, and inject $x_t$ into algorithm $\mathscr{A}_i$. to obtain action $a_t$.
5:     Play action $a_t$ and update algorithm $\mathscr{A}_i$.

---

The formal algorithm for heteroscedastic cases is given in Algorithm 2, and we have the following corollary on the second-order regret bound of Algorithm 2.

**Corollary 4.1.** The output $a_t$ of Algorithm 2 in round $t$ satisfies that with probability at least $1 - \delta$,

$$\sum_{t=1}^{T} (\max_{a \in \mathcal{A}} f^\star(x_t, a) - f^\star(x_t, a_t)) = \tilde{\mathcal{O}}\left( \sqrt{A \sum_{i=1}^{T} \sigma_i^2 \log(|\mathcal{F}|/\delta)} + d_{\text{elu}} \log(|\mathcal{F}|/\delta) \right).$$

The proof of Corollary 4.1 is given in Section E.

## 5 Strong Adversary Case

In this section, we consider the case where the adversary decides the variance $\sigma_t$ *after* seeing the action $a_t$ chosen by the learner. We provide regret lower and upper bounds matching up to a factor of $\sqrt{d_{\text{elu}}}$ and other logarithmic factors. More importantly, the minimax regret bounds differ with the weak adversary case (Section 4) as discussed in Section 3.2, demonstrating the even more crucial role of eluder dimension in this case.

**Regret lower bound**    In this strong adversary case, we first show that the adversary's power is enhanced in terms of the achievable minimax regret bounds. Concretely, we have the following theorem.

**Theorem 5.1.** *For any integer $d, A, T \ge 2$ and any positive real number $\Lambda \in [0, T]$, there exists a context space $\mathcal{X}$, a contextual bandit problem $\mathcal{F} \subset (\mathcal{X} \times \mathcal{A} \to \mathbb{R})$ with eluder dimension $d_{\text{elu}}(\mathcal{F}, 0) = d$ and action set $\mathcal{A} = [A]$ and an adversarial sequence of variances $\sigma_1^2, \ldots, \sigma_T^2$ with $\sum_{t=1}^{T} \sigma_t^2 \le \Lambda$ such that any algorithm will suffer a regret at least $\Omega(\min\{\sqrt{d\Lambda} + d, \sqrt{AT}\})$.*

The above theorem shows that the regret is at least $\Omega(\min\{\sqrt{d\Lambda} + d, \sqrt{AT}\})$ where $d = d_{\text{elu}}(\mathcal{F})$ even with $\log |\mathcal{F}| = \mathcal{O}(\log T)$. Recall that the bound in the weak adversary case (Section 4) can be written as $\Omega(\min\{\sqrt{\min\{A, d\}\Lambda} + d, \sqrt{AT}\})$. The power of the strong adversary is exactly the higher complexity $d$ in the $\Lambda$ term compared to $\min\{A, d\}$ in the weak adversary case. Now, we proceed to provide a matching upper bound up to a factor of $\sqrt{d}$.

**Regret upper bound**    For the strong adversary case, we adopt an optimism-based approach. In particular, we generalize the SAVE algorithm by Zhao et al. (2023), which achieves the tight $\tilde{\mathcal{O}}(d\sqrt{\Lambda} + d)$ bound for linear contextual bandits. We call the algorithm VarUCB and display it in Algorithm 5 of Appendix F. The algorithm combines the idea of weighted regression and multi-layer structure of SupLinUCB (Chu et al. (2011)) and refined variance-aware confidence set. Since this algorithm is a rather direct extension of Zhao et al. (2023)'s algorithm from the linear case to the non-linear case, we omit the detailed discussion on it and refer the readers to Zhao et al. (2023). Notice that for this algorithm, we do not need $\sigma_t$ to be revealed to the learner as in Section 4. In fact, we do not even need to know $\Lambda$. We have the following theorem for its regret guarantee.

**Theorem 5.2.** *When facing the strong adversary, Algorithm 5 guarantees a regret bound of $\tilde{\mathcal{O}}(d_{elu}\sqrt{\Lambda \log |\mathcal{F}|} + d_{elu} \log |\mathcal{F}|)$ with probability at least $1 - \delta$, where $\tilde{\mathcal{O}}(\cdot)$ hides $\log(T/\delta)$ factors.*

The proof is provided in Appendix F. Notice that when specializing Theorem 5.2 to the linear setting, the bound becomes $\tilde{\mathcal{O}}(\sqrt{d^3\Lambda} + d^2)$ since $\log |\mathcal{F}| = \Theta(d)$, which does not recover the bound of Zhao et al. (2023). Indeed, our analysis deviates from that of Zhao et al. (2023) due to the generality of non-linear function approximation. It is an interesting future direction to see whether our bound can be improved. We mention in passing that the work by Wang et al. (2024) obtained

---
**Algorithm 3** DistVarCB (Distributional Variance-aware Contextual Bandits)
---
1: Let $\mathcal{M}_1 = \mathcal{M}$, $M_1 = \frac{1}{|\mathcal{M}|}\sum_{M \in \mathcal{M}} M$, and $L = \Theta(\log(|\mathcal{M}|T/\delta))$.
2: **for** $t = 1, \ldots, T$ **do**
3:      Receive context $x_t$.
4:      **if** $\exists M, M' \in \mathcal{M}_t$ and $a \in \mathcal{A}$ such that $D_{\mathrm{H}}^2(M(x_t, a), M'(x_t, a)) \geq 1/2$ **then**
5:          Let $I_t = 1$ and pull $a_t = \arg\max_a \max_{M,M' \in \mathcal{M}_t} D_{\mathrm{H}}^2(M(x_t, a), M'(x_t, a))$.

6:      **else**
7:          Let $I_t = 2$ and pull $a_t \sim p_t$ where

$$p_t = \arg\min_{p \in \Delta(\mathcal{A})} \max_{M \in \mathcal{M}_t} \mathbb{E}_{a \sim p}\left[\max_{a'} f_M(x_t, a') - f_M(x_t, a) - \gamma \frac{(f_M(x_t, a) - f_{M_t}(x_t, a))^2}{\sigma_{M_t}^2(x_t, a)}\right].$$

8:      Receive $r_t$.
9:      Let $\mathcal{M}_{t+1} = \mathcal{M}_t \cap \left\{M : \sum_{s=1}^{t} D_{\mathrm{H}}^2(M(x_s, a_s), M_s(x_s, a_s)) \leq L\right\}$.
10:      Let $M_{t+1} = \frac{\sum_{M \in \mathcal{M}_{t+1}} q_t(M)M}{\sum_{M \in \mathcal{M}_{t+1}} q_t(M)}$,    where    $q_t(M) \propto \prod_{s=1}^{t} M(r_s | x_s, a_s)$.
---

$\sqrt{d\Lambda \log|\mathcal{M}|} + d \log|\mathcal{M}|$ upper bound where $d = d_{\mathrm{elu}}(\mathcal{M})$. However, the algorithm relies on having access to a model class. We study such a setting in our next section.

# 6 Learning with a Model Class

**Distributional setup** In this section, we consider the case where the learner is given a model class $\mathcal{M} \subset ((\mathcal{X} \times \mathcal{A}) \to \Delta(\mathbb{R}))$ where each model $M \in \mathcal{M}$ maps any context-action pair to a gaussian distribution, i.e., for any $x, a \in \mathcal{X} \times \mathcal{A}$,

$$M(x, a) = \mathcal{N}(f_M(x, a), \sigma_M(x, a)),$$

where $f_M(x, a)$ and $\sigma_M(x, a)$ are the mean and variance of the distribution $M(x, a)$. We assume that all the expected rewards and variances are bounded by $[0, 1]$. Recall, at round $t$, the reward is given by $r_t = f^\star(x_t, a_t) + \epsilon_t$. We further assume throughout this section that $\epsilon_t$ is Gaussian with variance $\sigma^\star(x_t, a_t)$ (since Gaussian is unbounded, we drop the assumption $r_t \in [0, 1]$ that we made in Section 2). Thus, the distribution of $r_t$ follows a true model $M^\star$ where $M^\star(x, a) = \mathcal{N}(f^\star(x, a), \sigma^\star(x, a))$.

**Assumption 6.1** (Model Realizability). *Assume $M^\star \in \mathcal{M}$.*

For this setup, it is useful to consider the Hellinger counterpart of the eluder dimension.

**Definition 6.1** (Hellinger Eluder Dimension). *For the model class $\mathcal{M}$ defined on the space $\mathcal{Z}$ (that is $\mathcal{M} \subset (\mathcal{Z} \to \Delta(\mathbb{R}))$), we define the Hellinger eluder dimension of $\mathcal{M}$ at scale $\alpha \geq 0$ as $d_{elu}^{\mathsf{H}}(\alpha)$ be the length of the longest sequence of tuples $(z_1, M_1, M_1'), ..., (z_m, M_m, M_m')$ and $\alpha_0 \geq \alpha$ such that for all $i = 1, .., m$, functions $M_i, M_i' \in \mathcal{M}$,*

$$\sum_{j<i} D_{\mathrm{H}}^2(M_i(z_j), M_i'(z_j)) \leq \alpha_0^2, \quad and \quad D_{\mathrm{H}}^2(M_i(z_i), M_i'(z_i)) > \alpha_0^2.$$

**Algorithm** Similar to Algorithm 1, we present Algorithm 3 tailored for the distributional case. At each round $t$, upon receiving the context $x_t$, the algorithm first checks if there exists an action $a$ such that two models within the localized model class $\mathcal{M}_t$ exhibit a significant divergence on the context-action pair $x_t, a$ measured by the squared Hellinger distance (Line 4). If such a difference is detected, the learner selects the action associated with the greatest divergence (Line 5). Conversely, if no action causes substantial divergence between models, the learner runs a variant of SquareCB (Foster and Rakhlin, 2020), employing adaptive variances to ensure low regret (Line 7). The major differences between Algorithm 1 and Algorithm 3 is that the latter measures the "disagreement" in terms of the squared Hellinger distance.

**Regret upper bound** We obtain the following distributional version regret bound for Algorithm 3.

**Theorem 6.1.** *For $d = d_{elu}^{\mathsf{H}}(1/\sqrt{T})$, the output $a_t$ of Algorithm 3 satisfies with probability at least $1 - \delta$,*

$$R_T = \tilde{\mathcal{O}}\left( \sqrt{A \sum_{t=1}^T \sigma_{M^\star}^2(x_t) \cdot \log(|\mathcal{M}|/\delta)} + d \log(|\mathcal{M}|/\delta) \right),$$

*where $\sigma_{M^\star}^2(x_t) = \max_{a \in A} \sigma_{M^\star}^2(x_t, a)$.*

A similar upper bound for a more general distributional case is obtained by Wang et al. (2024) in the form of $\tilde{\mathcal{O}}\left( \sqrt{d \sum_{t=1}^T \sigma_{M^\star}^2(x_t, a_t) \cdot \log |\mathcal{M}|} + d \log |\mathcal{M}| \right)$. In the leading term, our bound replaced the dependence of $d$ by the number of actions $A$ which is significantly smaller than $A$ in many cases of interest (e.g. linear, generalized linear). However, as a tradeoff, our bound also suffers a larger cumulative variance term. This tradeoff is necessary as we show in the following lower bound results that both our upper bound and their upper bound are optimal, i.e., matching lower bounds exist. Thus our result is at one end of the Pareto frontier.

**Regret lower bounds** We present the matching lower bound for our result as follows, which is essentially a rewrite of Theorem 4.1.

**Theorem 6.2.** *For any integer $d, A, T \geq 2$, any positive real number $\sigma \in [0, 1]$, there exists a context space $\mathcal{X}$ and a contextual bandit gaussian model class $\mathcal{M} \subset (\mathcal{X} \times \mathcal{A} \to \Delta(\mathbb{R}))$ with Hellinger eluder dimension $d_{elu}^{\mathsf{H}}(0) \leq d$, action set $\mathcal{A} = [A]$, and variances $\sigma_M(x, a) \leq \sigma$ for all $M \in \mathcal{M}$, $x, a \in \mathcal{X} \times \mathcal{A}$ such that any algorithm will suffer a regret at least $\Omega(\sqrt{\sigma^2 \min\{A, d\}T} + \min\{d, \sqrt{AT}\})$.*

Now we present the matching lower bound for the upper bound from Wang et al. (2024).

**Theorem 6.3.** *For any integer $d, A, T \geq 2$ and any positive real number $\Lambda \in [0, T]$, there exists a context space $\mathcal{X}$, a contextual bandit gaussian model class $\mathcal{M} \subset (\mathcal{X} \times \mathcal{A} \to \Delta(\mathbb{R}))$ with Hellinger eluder dimension $d_{elu}^{\mathsf{H}}(0) \leq d$ and action set $\mathcal{A} = [A]$ and the variances $\sum_{t=1}^T \sigma_{M^\star}(x_t, a_t)^2 \leq \Lambda$ such that any algorithm will suffer a regret at least $\Omega(\min\{\sqrt{d\Lambda} + d, \sqrt{AT}\})$.*

The lower bound obtained by Theorem 6.3 is an adaptation from Theorem 5.1 that crucially relies on the fact that the adversary can choose the variance according to the action $a_t$.

## 7 Open Questions

**Removing the revealing $\sigma_t$ assumption in the weak adversary setting** The assumption we made in Section 4 that the variance is revealed at the beginning of each round is rather restrictive, and ideally we would like to remove such an assumption. As a first step, we wonder whether the same regret bound $\tilde{O}(\sqrt{A\Lambda} + d_{elu})$ is achievable if the variance $\sigma_t$ is revealed at the *end* of round $t$. We answer this question affirmatively for the special case where $\sigma_t \in \{0, 1\}$. More details can be found in Appendix H. How to extend this result to general values of $\sigma_t$ is an interesting open question. Handling the case where $\sigma_t$ is never revealed is even more challenging but is the ultimate goal.

**Removing the Gaussian noise assumption in the distributional setting** Our Theorem 6.1 heavily relies on the assumption that the noise is Gaussian. We wonder whether such assumption can be relaxed or completely lifted. For example, can we obtain the same bound if the noise at round $t$ is just $\sigma_{M^\star}(x_t, a_t)$-sub-Gaussian? What if it is just a bounded noise with variance $\sigma_{M^\star}^2(x_t, a_t)$?

## Acknowledgement

We thank Dylan Foster for helpful discussions. We acknowledge support from ARO through award W911NF-21-1-0328, from the DOE through award DE-SC0022199, and from NSF through award DMS-2031883.

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

# Appendices

# A    Related Work

In this section, we review the literature in bandits/RL that obtains second-order regret bounds or other data-/instance-dependent regret bounds.

**Tabular/linear bandits and MDPs**    Second-order regret bounds for bandits have been extensively studied in non-contextual (fixed action set) settings, such as stochastic multi-armed bandits (Audibert et al., 2009), adversarial multi-armed bandits (Wei and Luo, 2018; Ito, 2021), and adversarial linear bandits (Hazan and Kale, 2011; Ito and Takemura, 2023). Key techniques in this line of work include replacing Hoeffding-style concentration bounds with Bernstein-style ones in value estimation, and using optimistic mirror descent (Rakhlin and Sridharan, 2013) to achieve variance reduction. Extending beyond non-contextual settings, a series of recent works (Zhang et al., 2021; Kim et al., 2022; Zhao et al., 2023) have focused on obtaining tight variance-dependent bounds for linear contextual bandits. The techniques developed for variance-dependent regret bounds in bandits have been extended to MDPs, leading to either variance-dependent or horizon-free bounds (Talebi and Maillard, 2018; Zanette and Brunskill, 2019; Zhang et al., 2021; Kim et al., 2022; Zhao et al., 2023; Zhou et al., 2023; Zhang et al., 2024). Additionally, other settings such as dueling bandits (Di et al., 2024) and sparse linear bandits (Dai et al., 2023) have been explored.

**General contextual bandits**    For contextual bandits with general policy or function classes, second-order bounds have been explored in both agnostic settings (Wei et al., 2020) and realizable settings (Wang et al., 2024). Wei et al. (2020) focused on the case where the number of actions is small. They demonstrated that, unlike in multi-armed or linear bandits, the tight second-order bound for general contextual bandits is more complex than simply replacing the zeroth-order term $T$ in the minimax bound with the second-order measure, which aligns with our findings. For instance, even if the second-order error $\mathcal{E}$ is $\mathcal{O}(1)$, they showed that the regret could still grow with $\Omega(T^{1/4})$. On the other hand, Wang et al. (2024) focused on the case where the model class that provides distributional information for the reward has small eluder dimension. In contrast to the small action regime, their bounds smoothly scale with the second-order measure.

**Other instance-dependent bounds in general contextual bandits**    Beyond second-order bounds, other works have focused on other data-dependent or instance-dependent bounds. For example, Allen-Zhu et al. (2018), Foster and Krishnamurthy (2021), and Wang et al. (2023) studied first-order (small-loss) bounds for general contextual bandits and MDPs, in which the goal is to make the regret scale with $\sqrt{L^\star}$, where $L^\star$ is the cumulative loss of the best policy. These bounds are generally achievable with specialized algorithms. For general contextual bandits, gap-dependent bounds that exhibit logarithmic dependence on $T$ have been derived by Foster et al. (2020) and Dann et al. (2023) for realizable and agnostic settings, respectively. Notably, the algorithm of Foster et al. (2020) has some similarity with our main algorithm, which we discuss further in the main text.

# B    Technical Tools

**Lemma B.1.** *For any $0 \leq \sigma \leq 1/2$ and $0 \leq \varepsilon \leq \sigma/2$, define*

$$P_\sigma = \begin{cases} 2\sigma, & \text{with proba. } 1/2, \\ 0, & \text{with proba. } 1/2, \end{cases} \quad P_{\sigma,\epsilon}^+ = \begin{cases} 2\sigma, & \text{with proba. } \frac{\sigma+\epsilon}{2\sigma}, \\ 0, & \text{with proba. } \frac{\sigma-\epsilon}{2\sigma}, \end{cases} \quad P_{\sigma,\epsilon}^- = \begin{cases} 2\sigma, & \text{with proba. } \frac{\sigma-\epsilon}{2\sigma}, \\ 0, & \text{with proba. } \frac{\sigma+\epsilon}{2\sigma}. \end{cases}$$

*Denote by $h_\sigma, h_{\sigma,\epsilon}^+, h_{\sigma,\epsilon}^-$ the means of the three distributions respectively. We have*

$$h_\sigma = \sigma, \;\; h_{\sigma,\epsilon}^+ = \sigma + \epsilon, \text{ and } h_{\sigma,\epsilon}^- = \sigma - \epsilon.$$

*Let $V_\sigma, V_{\sigma,\epsilon}^+, V_{\sigma,\epsilon}^-$ be the variance of the three distributions respectively. We have*

$$V_\sigma, V_{\sigma,\epsilon}^+, V_{\sigma,\epsilon}^- \leq \sigma^2.$$

*Furthermore,*

$$D_{\mathrm{KL}}\big(P_{\sigma,\epsilon}^- \,\|\, P_{\sigma,\epsilon}^+\big) \leq \frac{4\epsilon^2}{\sigma^2}.$$

**Proof.**

$$D_{\mathrm{KL}}\big(P_{\sigma,\epsilon}^- \,\|\, P_{\sigma,\epsilon}^+\big) = \frac{\sigma-\epsilon}{2\sigma} \log \frac{\sigma-\epsilon}{\sigma+\epsilon} + \frac{\sigma+\epsilon}{2\sigma} \log \frac{\sigma+\epsilon}{\sigma-\epsilon}$$

$$= \frac{\epsilon}{2\sigma} \log\left(\frac{\sigma + \epsilon}{\sigma - \epsilon}\right)^2$$

$$= \frac{\epsilon}{\sigma} \log\left(1 + \frac{2\epsilon}{\sigma - \epsilon}\right)$$

$$\leq \frac{\epsilon}{\sigma} \cdot \frac{2\epsilon}{\sigma - \epsilon}$$

$$\leq \frac{4\epsilon^2}{\sigma^2}.$$

$\square$

**Lemma B.2.** *For any two gaussian distributions $\mathcal{N}(\mu_1, \sigma_1)$ and $\mathcal{N}(\mu_2, \sigma_2)$, if $|\log \sigma_1 - \log \sigma_2| > 3$, then*

$$D_{\mathrm{H}}^2(\mathcal{N}(\mu_1, \sigma_1), \mathcal{N}(\mu_2, \sigma_2)) \geq \frac{1}{2}.$$

**Proof of Lemma B.2.** Without loss of generality, assume $\sigma_1 > e^3 \sigma_2 > 8\sigma_2$. The Hellinger divergence between two gaussian distributions writes

$$D_{\mathrm{H}}^2(\mathcal{N}(\mu_1, \sigma_1), \mathcal{N}(\mu_2, \sigma_2)) = 1 - \sqrt{\frac{2\sigma_1\sigma_2}{\sigma_1^2 + \sigma_2^2}} \exp\left(-\frac{(\mu_1 - \mu_2)^2}{4(\sigma_1^2 + \sigma_2^2)}\right)$$

$$\geq 1 - \sqrt{\frac{2\sigma_1\sigma_2}{\sigma_1^2 + \sigma_2^2}}$$

$$\geq 1 - \sqrt{\frac{2\sigma_1/\sigma_2}{(\sigma_1/\sigma_2)^2 + 1}} \geq \frac{1}{2}.$$

$\square$

**Lemma B.3** (Lemma 4.2 of Wang et al. (2024)). *For any distribution $\mathbb{P}$ and $\mathbb{Q}$ on any space $\mathcal{X}$ such that $D_{\mathrm{H}}^2(\mathbb{P}, \mathbb{Q}) \leq \frac{1}{2}$, we have for any function $h : \mathcal{X} \to \mathbb{R}$,*

$$|\mathbb{E}_{\mathbb{P}}[h] - \mathbb{E}_{\mathbb{Q}}[h]| \leq 2\sqrt{(\mathrm{Var}_{\mathbb{P}}(h) + \mathrm{Var}_{\mathbb{Q}}(h))D_{\mathrm{H}}^2(\mathbb{P}, \mathbb{Q})}.$$

**Lemma B.4** (Lemma A.11 of Foster et al. (2021)). *For any distribution $\mathbb{P}$ and $\mathbb{Q}$ on any space $\mathcal{X}$, we have for any function $h : \mathcal{X} \to [0, R]$,*

$$|\mathbb{E}_{\mathbb{P}}[h] - \mathbb{E}_{\mathbb{Q}}[h]| \leq \sqrt{2R(\mathbb{E}_{\mathbb{P}}(h) + \mathbb{E}_{\mathbb{Q}}(h))D_{\mathrm{H}}^2(\mathbb{P}, \mathbb{Q})}.$$

*In particular,*

$$\mathbb{E}_{\mathbb{P}}[h] \leq 3\mathbb{E}_{\mathbb{Q}}[h] + 4R D_{\mathrm{H}}^2(\mathbb{P}, \mathbb{Q}).$$

**Lemma B.5** (Strengthened Freedman's inequality (Theorem 9 of Zimmert and Lattimore (2022))). *Let $X_1, X_2, \ldots, X_T$ be a martingale difference sequence with a filtration $\mathscr{F}_1 \subseteq \mathscr{F}_2 \subseteq \cdots$ such that $\mathbb{E}[X_t|\mathscr{F}_t] = 0$ and $\mathbb{E}[|X_t| \mid \mathscr{F}_t] < \infty$ almost surely. Then with probability at least $1 - \delta$,*

$$\sum_{t=1}^{T} X_t \leq 3\sqrt{V_T \log\left(\frac{2\max\{U_T, \sqrt{V_T}\}}{\delta}\right)} + 2U_T \log\left(\frac{2\max\{U_T, \sqrt{V_T}\}}{\delta}\right),$$

*where $V_T = \sum_{t=1}^{T} \mathbb{E}[X_t^2 \mid \mathscr{F}_t]$ and $U_T = \max\{1, \max_{t \in [T]} |X_t|\}$.*

# C Omitted Proofs in Section 4.1

**Lemma C.1** (Theorem 15.2 of Lattimore and Szepesvári (2020)). *For any integer $A \geq 2$, any positive real number $\sigma \in [0, 1]$, and time $T > 0$, there exists a context space $\mathcal{X}$ and a contextual bandit problem $\mathcal{F} \subset (\mathcal{X} \times \mathcal{A} \to \mathbb{R})$ with action set $\mathcal{A} = [A]$, $d_{elu}(\mathcal{F}; 0) = A$, and variances $\sigma_t \leq \sigma$ for all $t \in [T]$ such that any algorithm will suffer a regret at least $\Omega(\sqrt{\sigma^2 A T})$.*

**Proof of Lemma C.1.** Without loss of generality assume $\sigma < 1/2$. Fix the algorithm. The first lower bound construction follows the standard MAB lower bound. Let the context space $\mathcal{X} = \emptyset$ and the function class

$$\mathcal{F} = \{f_i(\cdot) = (\sigma + \varepsilon)\mathbb{1}(\cdot = i) + \sigma\mathbb{1}(\cdot = 1) \mid i \in \{2, 3, \ldots, A\}\} \cup \{f_1(\cdot) = (\sigma - \varepsilon)\mathbb{1}(\cdot = i) + \sigma\mathbb{1}(\cdot = 1)\},$$

where the gap $\varepsilon$ will be specified later with the constraint $\varepsilon \leq 2\sigma$. Let $P_\sigma$, $P_{\sigma,\epsilon}^+$, and $P_{\sigma,\epsilon}^-$ be defined as in Lemma B.1. For any function $f \in \mathcal{F}$, suppose the environment is such that if $f(i) = \sigma, \sigma + \epsilon, \sigma - \epsilon$, then the reward distribution is $P_\sigma$, $P_{\sigma,\epsilon}^+$, $P_{\sigma,\epsilon}^-$ respectively. Any environment and algorithm give rise to the distribution of history. Let $\mathbb{P}_i$ denote the distribution generated by the environment following $f_i \in \mathcal{F}$ together with the algorithm and expectations under $\mathbb{P}_i$ will be denoted by $\mathbb{E}_i$. Let $N_T(a) = \sum_{t=1}^T \mathbb{1}(a_t = a)$ where $a_t$ is the action taken at time step $t$ for $t \in [T]$. Let

$$i^\star = \arg\min_{j>1} \mathbb{E}_1[N_T(j)].$$

Since $\sum_{i=1}^A N_T(i) = T$, it holds that $\mathbb{E}_1[N_T(i^\star)] \leq T/(A-1)$. For the two environments induced by $f_1$ and $f_{i^\star}$, we have

$$\mathbb{E}_1[R_T] \geq \mathbb{P}_1(N_T(1) \leq T/2) \cdot \frac{T\varepsilon}{2} \quad \text{and} \quad \mathbb{E}_{i^\star}[R_T] \geq \mathbb{P}_{i^\star}(N_T(1) > T/2) \cdot \frac{T\varepsilon}{2}.$$

Then, by Bretagnolle-Huber inequality, we have

$$\mathbb{E}_1[R_T] + \mathbb{E}_{i^\star}[R_T] \geq \frac{T\varepsilon}{2}(\mathbb{P}_1(N_T(1) \leq T/2) + \mathbb{P}_{i^\star}(N_T(1) > T/2))$$

$$\geq \frac{T\varepsilon}{4}\exp(-D_{\mathrm{KL}}(\mathbb{P}_1 \parallel \mathbb{P}_{i^\star})).$$

Then by the chain rule of KL divergence and Lemma B.1, we have

$$D_{\mathrm{KL}}(\mathbb{P}_1 \parallel \mathbb{P}_{i^\star}) = \mathbb{E}_1[N_T(i^\star)]D_{\mathrm{KL}}(P_{\sigma,\epsilon}^- \parallel P_{\sigma,\epsilon}^+) \leq \frac{4T\varepsilon^2}{(A-1)\sigma^2}.$$

Thus we have

$$\mathbb{E}_1[R_T] + \mathbb{E}_{i^\star}[R_T] \geq \frac{T\varepsilon}{4}\exp\left(-\frac{4T\varepsilon^2}{(A-1)\sigma^2}\right).$$

Then by choosing $\varepsilon = \sqrt{(A-1)\sigma^2/4T}$, we have

$$\max_i\{\mathbb{E}_i[R_T]\} \geq \Omega(\sqrt{\sigma^2 AT}).$$

$\square$

**Lemma C.2.** *For any integer $N, A, T \geq 2$, there exists a context space $\mathcal{X}$ and a **deterministic** ($\sigma_t = 0$ for all $t \in [T]$) contextual bandit problem $\mathcal{F} \subset (\mathcal{X} \times \mathcal{A} \to \mathbb{R})$ with action set $\mathcal{A} = [A]$ and $d_{elu}(\mathcal{F}; 0) = N(A-1)$ such that any algorithm will suffer a regret at least $\Omega(\min\{T/N, AN\})$.*

**Proof of Lemma C.2.** Consider the function class $\mathcal{F} = \{f^{(0)}\} \cup \{f^{(i,j)}\}_{i \in [N], j \in [A-1]}$ with the space of contexts $\mathcal{X} = \{x^{(1)}, \ldots, x^{(N)}\}$ and the set of actions $\mathcal{A} = [A]$. For any $i \in [N], j \in [A-1]$, the function $f^{(i,j)}$ is defined as the following: For $i \in [N]$ and $j \in [A-1]$,

$$f^{(i,j)}(x^{(i)}, j) = 1,$$
$$f^{(i,j)}(x, k) = 0, \quad \forall x \neq x^{(i)} \text{ or } \forall k \in [A-1] \setminus \{j\}.$$
$$f^{(i,j)}(x, A) = \frac{1}{2}, \quad \forall x.$$

Meanwhile

$$f^{(0)}(x, j) = 0, \quad \forall x \text{ and } \forall j \in [A-1]$$
$$f^{(0)}(x, A) = \frac{1}{2}, \quad \forall x.$$

The eluder dimension of this function class is $N(A-1)$ since $f^{(i,j)}$ is uniquely identified by its value on $(x^{(i)}, j)$. We assume that $x_t$ is uniformly randomly chosen from $\mathcal{X}$, and $r_t = f_\star(x_t, a_t)$. That is, $\sigma_t = 0$ for all $t \in [T]$.

Fix any algorithm. Denote by $\mathbb{P}_0$ the probability distribution when $f^\star = f^{(0)}$ and $\mathbb{E}_0$ the expectation under $\mathbb{P}_0$. For any $i \in [N], j \in [A-1]$, denote by $\mathbb{P}_{(i,j)}$ the probability distribution when $f^\star = f^{(i,j)}$ and $\mathbb{E}_{(i,j)}$ the expectation under $\mathbb{P}_{(i,j)}$. Let $N_T(i,j) = \sum_{t=1}^T \mathbb{1}[x_t = x^{(i)}, a_t = j]$. Then the adversary decides $f_\star$ based on the following rule: if there exists $i \in [N], j \in [A-1]$ such that

$$\mathbb{E}_0\left[N_T(i,j)\right] \le \frac{1}{100} \tag{8}$$

then let $f_\star = f^{(i,j)}$. If no $i,j$ satisfies this, let $f_\star = f^{(0)}$. If $f_\star = f^{(0)}$, then we have

$$\mathbb{E}_0[R_T] \ge \frac{1}{2}\mathbb{E}_0\left[\sum_{i \in [N], j \in [A-1]} N_T(i,j)\right] \ge \frac{N(A-1)}{2} \cdot \min_{i,j} \mathbb{E}_0[N_T(i,j)] \ge \frac{N(A-1)}{200}.$$

On the other hand, if $f_\star = f^{(i,j)}$, then we have

$$\mathbb{P}_0(N_T(i,j) = 0) \ge \frac{99}{100}.$$

Then by Lemma B.4, we have

$$\mathbb{P}_0(N_T(i,j) = 0) \le 3\mathbb{P}_{(i,j)}(N_T(i,j) = 0) + 4D_{\mathrm{H}}^2\big(\mathbb{P}_0, \mathbb{P}_{(i,j)}\big).$$

Then by Lemma D.2 of Foster et al. (2024), we have

$$D_{\mathrm{H}}^2\big(\mathbb{P}_0, \mathbb{P}_{(i,j)}\big) \le 7\mathbb{E}_0[N_T(i,j)].$$

Altogether, we can obtain

$$\mathbb{P}_{(i,j)}(N_T(i,j) = 0) \ge \frac{1}{3}(\mathbb{P}_0(N_T(i,j) = 0) - 28\mathbb{E}_0[N_T(i,j)]) \ge 1/6.$$

This in turn implies that

$$\mathbb{E}_{(i,j)}[R_T] \ge \frac{1}{2}\mathbb{E}_{(i,j)}\left[\sum_{t=1}^T \mathbb{1}[x_t = x^{(i)}] - N_T(i,j)\right] \ge \frac{T}{2N} \cdot \mathbb{P}_{(i,j)}(N_T(i,j) = 0) \ge \frac{T}{12N}.$$

Combining the lower bounds for $\mathbb{E}_0[R_T]$ and $\mathbb{E}_{(i,j)}[R_T]$ finishes the proof. $\qquad\square$

**Proof of Theorem 4.1.** If $A > T$ and $d > T$, by Lemma C.2 with $N = 1$, we have lower bound $\Omega(T)$. Below, we assume $\min\{A, d\} \le T$.

In order to prove the lower bound, we only need to show that for any fixed $d, A, \sigma, T$ such that $\min\{A, d\} \le T$, there exists two classes where one has lower bound $\Omega(\sqrt{\sigma^2 \min\{A, d\}T})$ and the other has lower bound $\Omega(\min\{d, \sqrt{AT}\})$.

If $A \le d$, then we invoke Lemma C.1 to obtain the lower bound of $\Omega(\sqrt{\sigma^2 AT})$. Else if $d \le A$, we can again invoke Lemma C.1 with the action set $[d]$ and then expand the action set with dummy actions with all 0 rewards to obtain the lower bound of $\Omega(\sqrt{\sigma^2 dT})$. In all, we have shown that there is a lower bound of $\Omega(\sqrt{\sigma^2 \min\{A, d\}T})$.

If $d \le A$ (which implies $d \le T$ since we assume $\min\{A, d\} \le T$), then we invoke Lemma C.2 with $N = 1$ with action set $\mathcal{A}$ be $[d]$ plus $A - d$ dummy actions. Then we get a lower bound of $\Omega(\min\{T, d\}) = \Omega(d)$. Next, consider the case $d \ge A$. If $d \le \sqrt{AT}$, then we invoke Lemma C.2 with $N = d/A$ and obtain a lower bound of $\Omega(\min\{AT/d, d\}) = \Omega(d)$. Else we have $d > \sqrt{AT}$ (which implies $T \ge A$ since we assume $\min\{A, d\} \le T$). Then we consider the hard case from Lemma C.2 with $N = \sqrt{T/A}$ then the function class has Hellinger eluder dimension $\sqrt{AT}$. Embedding this function class into a more complex model class with a larger Hellinger eluder dimension, we can obtain the lower bound of $\Omega(\sqrt{AT})$. In all, we have shown that there is a lower bound of $\Omega(\min\{d, \sqrt{AT}\})$.

Consequently, we have shown a lower bound of $\Omega(\sqrt{\sigma^2 \min\{A, d\}T} + \min\{d, \sqrt{AT}\})$.

$\qquad\square$

**Algorithm 4** Prod-based online regression oracle

---

**Input:** Parameter $\eta$. Contextual Bandit Oracle gives function class $\mathcal{F}_t \subset \mathcal{F}$, action class $\mathcal{A}_t$ and context $x_t$ at round $t$. Contextual Bandit algorithm which takes online regression oracle and returns an action.

1: Let $q_1(f) = {}^1/_{|\mathcal{F}_1|}$ for every function $f \in \mathcal{F}_1$.
2: **for** $t = 1, 2, \dots$ **do**
3:     Generate
$$f_t(x_t, a) = \sum_{f \in \mathcal{F}_t} q_t(f) f(x_t, a) \tag{9}$$

4:     Output $\{f_t(x_t, a) : a \in \mathcal{A}_t\}$, and feed to Contextual Bandit algorithm to receive $a_t \in \mathcal{A}_t$.
5:     Call Contextual Bandit oracle to get $\mathcal{F}_{t+1}$ and $x_{t+1}$.
6:     Calculate
$$\tilde{\ell}_t(f) = 2(f(x_t, a_t) - f_t(x_t, a_t))(f_t(x_t, a_t) - r_t) \tag{10}$$

7:     and
$$q_{t+1}(f) = \frac{q_t(f)(1 - \eta \tilde{\ell}_t(f))}{\sum_{f \in \mathcal{F}_{t+1}} q_t(f)(1 - \eta \tilde{\ell}_t(f))} \quad \forall f \in \mathcal{F}_{t+1}. \tag{11}$$

---

# D  Omitted Proofs in Section 4.2

## D.1  Analysis of the Online Regression Oracle

**Lemma D.1** (Cesa-Bianchi et al. (2007)). *Fix some positive parameter $\eta > 0$. Suppose we have function sets $\mathcal{F}_1 \supset \mathcal{F}_2 \supset \cdots \supset \mathcal{F}_T$, and functions $\ell_t : \mathcal{F}_t \to \mathbb{R}$ satisfies that $\eta|\ell_t(f)| \leq \frac{1}{2}$ for any $f \in \mathcal{F}_t$. The prediction rule*

$$q_t(f) = \begin{cases} \frac{\prod_{\tau=1}^{t-1}(1 - \eta \ell_\tau(f))}{\sum_{g \in \mathcal{F}_t} \prod_{\tau=1}^{t-1}(1 - \eta \ell_\tau(g))} & f \in \mathcal{F}_t, \\ 0 & f \notin \mathcal{F}_t, \end{cases}$$

*ensures for any $f^\star \in \mathcal{F}_T$,*

$$\sum_{t=1}^{T} \left( \sum_{f \in \mathcal{F}_t} q_t(f)\ell_t(f) \right) - \sum_{t=1}^{T} \ell_t(f^\star) \leq \frac{\log|\mathcal{F}|}{\eta} + \eta \sum_{t=1}^{T} \ell_t(f^\star)^2.$$

**Proof.** Define $w_t(f) = \prod_{\tau=1}^{t}(1 - \eta \ell_\tau(f))$ and $W_t = \sum_{f \in \mathcal{F}_t} w_t(f)$.

$$\begin{aligned}
\log \frac{W_t}{W_{t-1}} &= \log \frac{\sum_{f \in \mathcal{F}_t} w_t(f)}{\sum_{f \in \mathcal{F}_{t-1}} w_{t-1}(f)} \\
&\leq \log \frac{\sum_{f \in \mathcal{F}_t} w_{t-1}(f)(1 - \eta \ell_t(f))}{\sum_{f \in \mathcal{F}_t} w_{t-1}(f)} \\
&\overset{(i)}{=} \log \left( \sum_{f \in \mathcal{F}_t} q_t(f)(1 - \eta \ell_t(f)) \right) \\
&\overset{(ii)}{=} \log \left( 1 - \eta \sum_{f \in \mathcal{F}_t} q_t(f)\ell_t(f) \right) \\
&\overset{(iii)}{=} -\eta \sum_{f \in \mathcal{F}_t} q_t(f)\ell_t(f),
\end{aligned}$$

where in $(i)$ we use the definition of $q_t$, in $(ii)$ we use the fact that $\sum_{f \in \mathcal{F}_t} q_t(f) = 1$, and in $(iii)$ we use the inequality $\log(1 - x) \leq -x$ for any $x < 1$. Therefore,

$$\sum_{t=1}^{T} \sum_{f \in \mathcal{F}_t} q_t(f)\ell_t(f) \leq \frac{1}{\eta} \log \frac{W_0}{W_T}$$

$$\overset{(i)}{\leq} \frac{\log |\mathcal{F}|}{\eta} - \frac{1}{\eta} \log w_T(f^\star)$$

$$= \frac{\log |\mathcal{F}|}{\eta} - \frac{1}{\eta} \sum_{t=1}^{T} \log(1 - \eta \ell_t(f^\star))$$

$$\overset{(ii)}{\leq} \frac{\log |\mathcal{F}|}{\eta} - \frac{1}{\eta} \sum_{t=1}^{T} (-\eta \ell_t(f^\star) - \eta^2 \ell_t(f^\star)^2)$$

$$= \frac{\log |\mathcal{F}|}{\eta} + \sum_{t=1}^{T} \ell_t(f^\star) + \eta \sum_{t=1}^{T} \ell_t(f^\star)^2,$$

where in $(i)$ we use the fact that $W_0 = |\mathcal{F}|$ and $W_T \geq w_T(f^\star)$ for any $f^\star \in \mathcal{F}_t$, and in $(ii)$ we use the fact that $\eta|\ell_t(f^\star)| \leq \frac{1}{2}$ and also the inequality $\log(1 - x) \geq -x - x^2$ for any $|x| \leq \frac{1}{2}$. $\qquad\square$

**Lemma D.2.** *Suppose for any $x \in \mathcal{X}, a \in \mathcal{A}, f \in \mathcal{F}$, we always have $f(x, a) \in [0, 1]$, the reward $r_t \in [0, 1]$ and for any $f, f' \in \mathcal{F}_t$, we always have*

$$\max_{a \in \mathcal{A}_t} |f(x_t, a) - f'(x_t, a)| \leq \tilde{\Delta}.$$

*Then for the output $f_t$ according to Algorithm 4, we have with probability at least $1 - \delta$,*

$$\sum_{t=1}^{T} (f_t(x_t, a_t) - r_t)^2 - \sum_{t=1}^{T} (f^\star(x_t, a_t) - r_t)^2 \leq 16(\sigma^2 + \tilde{\Delta}) \log (|\mathcal{F}|/\delta).$$

**Proof.** We first notice that with our choice of $\eta = \frac{1}{4(\sigma^2 + \tilde{\Delta})}$, for any $f \in \mathcal{F}_t$ we have

$$\eta|\tilde{\ell}_t(f)| \leq 2\eta|f(x_t, a_t) - f_t(x_t, a_t)| \leq 2\eta \max_{f' \in \mathcal{F}_t} |f(x_t, a_t) - f'(x_t, a_t)| \leq \frac{2\tilde{\Delta}}{4(\sigma^2 + \tilde{\Delta})} \leq \frac{1}{2}$$

where $\tilde{\ell}_t$ is defined in Eq. (10). According to Lemma D.1, we have

$$\sum_{t=1}^{T} \sum_{f \in \mathcal{F}_t} q_t(f)\tilde{\ell}_t(f) - \sum_{t=1}^{T} \tilde{\ell}_t(f^\star) \leq \frac{\log |\mathcal{F}|}{\eta} + \eta \sum_{t=1}^{T} \tilde{\ell}_t(f^\star)^2,$$

By the definition of $f_t$ in Eq. (9) and $\tilde{\ell}_t$ in Eq. (10), we have

$$\sum_{f \in \mathcal{F}_t} q_t(f)\tilde{\ell}_t(f) = 2(f_t(x_t, a_t) - r_t) \cdot \left( \sum_{f \in \mathcal{F}_t} q_t(f)f(x_t, a_t) - f_t(x_t, a_t) \right) = 0.$$

Next, we observe that

$$\tilde{\ell}_t(f^\star) = (f^\star(x_t, a_t) - r_t)^2 - (f_t(x_t, a_t) - r_t)^2 - (f_t(x_t, a_t) - f^\star(x_t, a_t))^2,$$

which implies

$$\sum_{t=1}^{T} (f_t(x_t, a_t) - r_t)^2 - (f^\star(x_t, a_t) - r_t)^2$$

$$= -\sum_{t=1}^{T} \tilde{\ell}_t(f^\star) - \sum_{t=1}^{T} (f_t(x_t, a_t) - f^\star(x_t, a_t))^2$$

$$\leq \frac{\log |\mathcal{F}|}{\eta} + 4\eta \sum_{t=1}^{T} \tilde{\ell}_t(f^\star)^2 - \sum_{t=1}^{T} (f_t(x_t, a_t) - f^\star(x_t, a_t))^2. \tag{12}$$

We notice that

$$\tilde{\ell}_t(f^\star) = 2(f_t(x_t, a_t) - f^\star(x_t, a_t))(r_t - f^\star(x_t, a_t))$$

is a martingale difference sequence, and we further have $|\tilde{\ell}_t(f^\star)^2| \le 4\tilde{\Delta}^2$ and

$$
\begin{aligned}
\mathrm{Var}_t(\tilde{\ell}_t(f^\star)^2) &\le 4(f_t(x_t, a_t) - f^\star(x_t, a_t))^4 \mathbb{E}_t[(r_t - f^\star(x_t, a_t))^4] \\
&\le 4(f_t(x_t, a_t) - f^\star(x_t, a_t))^4 \mathbb{E}_t[(r_t - f^\star(x_t, a_t))^2] \\
&= 4(f_t(x_t, a_t) - f^\star(x_t, a_t))^4 \sigma^2.
\end{aligned}
$$

Hence according to Freedman inequality (Freedman, 1975), we have with probability $1 - \delta$,

$$
\begin{aligned}
\sum_{t=1}^{T} \tilde{\ell}_t(f^\star)^2 &\le 2\sqrt{\sum_{t=1}^{T} 4(f_t(x_t, a_t) - f^\star(x_t, a_t))^4 \sigma^2 \log \frac{1}{\delta}} + 2 \cdot 4\tilde{\Delta}^2 \log \frac{1}{\delta} \\
&\le 2\sqrt{\sum_{t=1}^{T} (f_t(x_t, a_t) - f^\star(x_t, a_t))^2 \sigma^2 \cdot 4\tilde{\Delta}^2 \log \frac{1}{\delta}} + 2 \cdot 4\tilde{\Delta}^2 \log \frac{1}{\delta} \\
&\le \sum_{t=1}^{T} (f_t(x_t, a_t) - f^\star(x_t, a_t))^2 \sigma^2 + 12\tilde{\Delta}^2 \log \frac{1}{\delta},
\end{aligned}
$$

where in the second inequality we use the fact that $|f_t(x_t, a_t) - f^\star(x_t, a_t)| \le \tilde{\Delta}$, and in the last inequality we use the AM-GM inequality. Hence, with our choice of $\eta = \frac{1}{4(\sigma^2 + \tilde{\Delta})}$, we have

$$
\begin{aligned}
\sum_{t=1}^{T} &(f_t(x_t, a_t) - r_t)^2 - (f^\star(x_t, a_t) - r_t)^2 \\
&\le \frac{\log |\mathcal{F}|}{\eta} + 4\eta \sum_{t=1}^{T} \tilde{\ell}_t(f^\star)^2 - \sum_{t=1}^{T} (f_t(x_t, a_t) - f^\star(x_t, a_t))^2 \\
&\le 4(\sigma^2 + \tilde{\Delta}) \log |\mathcal{F}| + 12\tilde{\Delta}^2 \log \frac{1}{\delta} \\
&\le 16(\sigma^2 + \tilde{\Delta}) \log \frac{|\mathcal{F}|}{\delta}
\end{aligned}
$$

where in the last inequality we use the fact that $\tilde{\Delta} \le 1$. $\qquad\square$

**Lemma D.3.** *Suppose for any $x \in \mathcal{X}, a \in \mathcal{A}, f \in \mathcal{F}$, we always have $f(x, a) \in [0, 1]$, the reward $r_t \in [0, 1]$ and for any $f, f' \in \mathcal{F}_t$, we always have*

$$
\max_{a \in \mathcal{A}_t} |f(x_t, a) - f'(x_t, a)| \le \tilde{\Delta}.
$$

*Then for the output $f_t$ according to Algorithm 4, we have with probability at least $1 - \delta$,*

$$
\sum_{t=1}^{T} \sum_{a \in \mathcal{A}} p_t(a)(f_t(x_t, a) - f^\star(x_t, a))^2 \le 48(\sigma^2 + \tilde{\Delta}) \log\left(2|\mathcal{F}|/\delta\right).
$$

**Proof.** We use $\mathscr{F}_t$ denote the filtration constructed by $\mathscr{F}_t = \sigma(x_{1:t}, a_{1:t}, r_{1:t})$. And we let

$$
M_t = (f_t(x_t, a_t) - r_t)^2 - (f^\star(x_t, a_t) - r_t)^2.
$$

Then we have $|M_t| \le 2\tilde{\Delta}$. According to (Foster and Rakhlin, 2020, Lemma 1, Lemma 4), we have with probability at least $1 - \delta/2$,

$$
\sum_{t=1}^{T} \sum_{a \in \mathcal{A}} p_t(a)(f_t(x_t, a) - f^\star(x_t, a))^2 \le 2 \sum_{t=1}^{T} \left((f_t(x_t, a_t) - r_t)^2 - (f^\star(x_t, a_t) - r_t)^2\right) + 16\tilde{\Delta} \log\left(\frac{2}{\delta}\right).
$$

Next, according to Lemma D.2, we have with probability at least $1 - \delta/2$,

$$
\sum_{t=1}^{T} \left((f_t(x_t, a_t) - r_t)^2 - (f^\star(x_t, a_t) - r_t)^2\right) \le 16(\sigma^2 + \tilde{\Delta}) \log\left(\frac{2|\mathcal{F}|}{\delta}\right).
$$

Hence we obtain that with probability at least $1 - \delta$,

$$\sum_{t=1}^{T} \sum_{a \in \mathcal{A}} p_t(a)(f_t(x_t, a) - f^{\star}(x_t, a))^2 \leq 48(\sigma^2 + \tilde{\Delta}) \log(2|\mathcal{F}|/\delta).$$

$\square$

### D.2 Proof of Theorem 4.2

For simplicity, we define the following sets based on Algorithm 1:

$$\mathcal{T}_1 = \{t \in [T], \text{'if' condition in Line 6 holds in Algorithm 1}\}, \quad \text{and} \quad \mathcal{T}_2 = [T] \backslash \mathcal{T}_1. \tag{13}$$

We first show that with high probability, the true model $f^{\star} \in \mathcal{F}_t$, where $\mathcal{F}_t$ is defined in Eq. (5) in Algorithm 1.

**Lemma D.4.** *When the function class $\mathcal{F}_t$ iteratively defined in Eq. (5), with probability at least $1 - \delta$, we have $f^{\star} \in \mathcal{F}_t$ for any $1 \leq t \leq T$.*

**Proof.** We will prove the result by induction on $t$. For $t = 1$, since $\mathcal{F}_1 = \mathcal{F}$ and according to Assumption 2.1 we have $f^{\star} \in \mathcal{F}_1$. Next, we will assume that $f^{\star} \in \mathcal{F}_{t-1}$ and attempt to prove $f^{\star} \in \mathcal{F}_t$. Since $r_{\tau} = f^{\star}(x_{\tau}, a_{\tau}) + \epsilon_t$ for any $1 \leq \tau \leq t - 1$, we have

$$\sum_{\tau=1}^{t-1} w_{\tau}(f_t(x_{\tau}, a_{\tau}) - f^{\star}(x_{\tau}, a_{\tau}))^2$$

$$= \sum_{\tau=1}^{t-1} w_{\tau}(f_t(x_{\tau}, a_{\tau}) - r_{\tau})^2 - \sum_{\tau=1}^{t-1} w_{\tau}(f^{\star}(x_{\tau}, a_{\tau}) - r_{\tau})^2$$

$$+ 2 \sum_{\tau=1}^{t-1} w_{\tau}(f_t(x_{\tau}, a_{\tau}) - f^{\star}(x_{\tau}, a_{\tau}))(r_{\tau} - f^{\star}(x_{\tau}, a_{\tau}))$$

$$\leq 2 \sum_{\tau=1}^{t-1} w_{\tau} \epsilon_{\tau}(f_t(x_{\tau}, a_{\tau}) - f^{\star}(x_{\tau}, a_{\tau})),$$

where the last inequality uses the definition that $f_t$ is the minimizer of $\sum_{\tau=1}^{t-1} w_{\tau}(f(x_{\tau}, a_{\tau}) - r_{\tau})^2$ in (6).

We notice that $w_{\tau}(f_t(x_{\tau}, a_{\tau}) - f^{\star}(x_{\tau}, a_{\tau}))\epsilon_{\tau}$ is a martingale difference sequence, and since $|w_{\tau}| \leq \frac{1}{\sigma^2}$,

$$\max_{1 \leq \tau \leq t-1} |w_{\tau}(f_t(x_{\tau}, a_{\tau}) - f^{\star}(x_{\tau}, a_{\tau}))\epsilon_{\tau}| \leq \frac{1}{\sigma^2},$$

$$\sum_{\tau=1}^{t-1} (w_{\tau}(f_t(x_{\tau}, a_{\tau}) - f^{\star}(x_{\tau}, a_{\tau}))\epsilon_{\tau}|)^2 \leq \frac{T}{\sigma^4}.$$

According to the Strengthened Freeman's Inequality (Lemma B.5), and noticing that $\mathbb{E}_{\tau-1}[\epsilon_{\tau}^2] = \sigma^2$ for any $1 \leq \tau \leq t - 1$, with probability at least $1 - \frac{\delta}{T}$, for any $f \in \mathcal{F}$,

$$\sum_{\tau=1}^{t-1} w_{\tau} \epsilon_{\tau}(f_t(x_{\tau}, a_{\tau}) - f^{\star}(x_{\tau}, a_{\tau}))$$

$$\leq 3 \sqrt{\sum_{\tau=1}^{t-1} w_{\tau}^2 \sigma^2 (f_t(x_{\tau}, a_{\tau}) - f^{\star}(x_{\tau}, a_{\tau}))^2 \log \frac{|\mathcal{F}|T^2}{\delta \sigma^2}} + 2 \max_{1 \leq \tau \leq t-1} |w_{\tau} \epsilon_{\tau}(f_t(x_{\tau}, a_{\tau}) - f^{\star}(x_{\tau}, a_{\tau}))| \log \frac{|\mathcal{F}|T^2}{\delta \sigma^2}$$

$$\overset{(i)}{\leq} 3 \sqrt{\sum_{\tau=1}^{t-1} w_{\tau}(f_t(x_{\tau}, a_{\tau}) - f^{\star}(x_{\tau}, a_{\tau})^2 L + 2 \sup_{\tau \leq t-1} w_{\tau}|f_t(x_{\tau}, a_{\tau}) - f^{\star}(x_{\tau}, a_{\tau})|L}$$

$$
\overset{(ii)}{\leq} 3\sqrt{\sum_{\tau=1}^{t-1} w_\tau (f_t(x_\tau, a_\tau) - f^\star(x_\tau, a_\tau))^2 L} + 2 \sup_{\tau \leq t-1} \sqrt{L + \sum_{s=1}^{\tau-1} w_s (f_t(x_s, a_s) - f^\star(x_s, a_s))^2 L}
$$

$$
\leq 5\sqrt{\sum_{\tau=1}^{t-1} w_\tau (f_t(x_\tau, a_\tau) - f^\star(x_\tau, a_\tau))^2 L + L}
$$

$$
\overset{(iii)}{\leq} \frac{1}{2} \sum_{\tau=1}^{t-1} w_\tau (f_t(x_\tau, a_\tau) - f^\star(x_\tau, a_\tau))^2 + 51L,
$$

where in $(i)$ we use the fact that $w_\tau \leq 1/\sigma^2$ for any $\tau$ according to the definition of $w_\tau$ in Algorithm 1 and using the definition $L \triangleq \frac{|\mathcal{F}|T^2}{\delta \sigma^2}$, in $(ii)$ we use the fact that

$$
w_\tau \leq \frac{\sqrt{1 + \sum_{s=1}^{\tau-1} w_s (f_s(x_s, a_s) - f^\star(x_s, a_s))^2}}{|f_\tau(x_\tau, a_\tau) - f^\star(x_\tau, a_\tau)|\sqrt{L}}
$$

according to the definition of $w_\tau$ in Algorithm 1 since $f_\tau, f^\star \in \mathcal{F}_\tau$ by induction hypothesis, and finally in $(iii)$ we use AM-GM inequality. And this proves the induction hypothesis of $f^\star \in \mathcal{F}_t$.

Therefore, we obtain that with probability at least $1 - \delta$ for any $1 \leq t \leq T$, $f^\star \in \mathcal{F}_t$. $\qquad\square$

The following lemma is a useful result of Eluder dimension.

**Lemma D.5.** *For any $\lambda \geq 0$ and $\alpha \in (0, 1]$, we have*

$$
\sum_{t=1}^{T} \min \left\{ \lambda, \sup_{f, f' \in \mathcal{F}_t} \frac{(f(x_t, a_t) - f'(x_t, a_t))^2}{\alpha^2 T + \sum_{\tau=1}^{t-1} (f(x_\tau, a_\tau) - f'(x_\tau, a_\tau))^2} \right\} \leq d_{elu}(\alpha) \left( \lambda + \log T \right).
$$

**Proof.** This proof follows Lemma 5.1 of Ye et al. (2023). Create $T$ bins, and call them $B_1, \ldots, B_T$. Each bin is empty at the beginning. Below we will add elements in $\{1, 2, \ldots, T\}$ to the bins.

Suppose that $\{1, 2, \ldots, t-1\}$ have been assigned to their bins. To assign $t$ to a bin, we find the smallest $n \in [T]$ such that "$\exists \epsilon \geq \alpha$, $(x_t, a_t)$ is $\epsilon$-independent to the elements in $B_n$ with respect to $\mathcal{F}_t$." We let $n_t$ to be the $n$ we found, and put element $i$ into bin $B_{n_i}$.

By the procedure above, we can conclude that for each $t$, "$\forall \epsilon \geq \alpha$, $(x_t, a_t)$ is $\epsilon$-dependent on all of $\{B_1, \ldots, B_{n_t-1}\}$ with respect to $\mathcal{F}_t$." Next, we show that this necessitates the following: for any $f, f' \in \mathcal{F}_t$,

$$
(f(x_t, a_t) - f'(x_t, a_t))^2 \leq \alpha^2 + \frac{1}{n_t - 1} \sum_{\tau=1}^{t-1} (f(x_\tau, a_\tau) - f'(x_\tau, a_\tau))^2
$$

This is because if otherwise, then there exists $f, f' \in \mathcal{F}_t$ and a bin $B_n \in \{B_1, \ldots, B_{n_t-1}\}$ which $(x_t, a_t)$ is $\epsilon$-dependent on $\forall \epsilon \geq \alpha$, but

$$
(f(x_t, a_t) - f'(x_t, a_t))^2 > \alpha^2 + \sum_{\tau \in B_n} (f(x_\tau, a_\tau) - f'(x_\tau, a_\tau))^2.
$$

Choose

$$
\xi = \max \left\{ \max_{\tau \in B_n} |f(x_\tau, a_\tau) - f'(x_\tau, a_\tau)|, \alpha \right\} \geq \alpha.
$$

Clearly, $|f(x_\tau, a_\tau) - f'(x_\tau, a_\tau)| \leq \xi$ for all $\tau \in B_n$, but $|f(x_t, a_t) - f'(x_t, a_t)| > \alpha$, contradicting that $(x_t, a_t)$ is $\epsilon$-dependent on $B_n$ $\forall \epsilon \geq 1/\sqrt{T}$.

We obtain that

$$
\sup_{f, f' \in \mathcal{F}_t} \frac{(f(x_t, a_t) - f'(x_t, a_t))^2}{\alpha^2 T + \sum_{\tau=1}^{t-1} (f(x_\tau, a_\tau) - f'(x_\tau, a_\tau))^2} \leq \frac{1}{n_t - 1}.
$$

Let $|B_1|, \ldots, |B_T|$ be the size of the bins after all elements are added. By the eluder dimension definition (Definition 2.1), we know $|B_n| \leq d_{\text{elu}}(\alpha)$ for all $n$. Therefore, we have

$$\sum_{t=1}^{T} \min\left\{\lambda, \sup_{f,f' \in \mathcal{F}_t} \frac{(f(x_t, a_t) - f'(x_t, a_t))^2}{1 + \sum_{\tau=1}^{t-1}(f(x_\tau, a_\tau) - f'(x_\tau, a_\tau))^2}\right\}$$

$$= \sum_{t:n_t=1} \lambda + \sum_{n=2}^{T} \sum_{t:n_t=n} \frac{1}{n-1}$$

$$\leq \lambda d_{\text{elu}}(\alpha) + \sum_{n=2}^{T} \frac{d_{\text{elu}}(\alpha)}{n-1}$$

$$\leq \lambda d_{\text{elu}}(\alpha) + d_{\text{elu}}(\alpha) \log T.$$

$\square$

**Lemma D.6.** *Suppose we have positive number $B$ such that for any $t \in [T]$, $w_t \in [0, B]$. Then for any $\lambda \geq 0$, we have*

$$\sum_{t=1}^{T} \min\left\{\lambda, \sup_{f,f' \in \mathcal{F}_t} \frac{w_t(f(x_t, a_t) - f'(x_t, a_t))^2}{1 + \sum_{\tau=1}^{t-1} w_\tau(f(x_\tau, a_\tau) - f'(x_\tau, a_\tau))^2}\right\}$$

$$\leq 3d_{elu}(1/\sqrt{BT})(\lambda + \log T)\log(BT).$$

**Proof.** We define set $\mathcal{T}_i = \{t \in [T] : 2^{i-1}/T \leq w_t \leq 2^i/T\}$ for any $1 \leq i \leq \log(BT)$. and $\mathcal{T}_0 = \{t \in [T] : w_t \in [0, 1/T]\}$. Then we have

$$[T] \subset \cup_{i=1}^{\log(B/A)} \mathcal{T}_i.$$

Additionally, we notice that for any $1 \leq i \leq \log(BT)$ and $t \in \mathcal{T}_i$, we have

$$\frac{w_t(f(x_t, a_t) - f'(x_t, a_t))^2}{1 + \sum_{\tau=1}^{t-1} w_\tau(f(x_\tau, a_\tau) - f'(x_\tau, a_\tau))^2}$$

$$\leq \frac{w_t(f(x_t, a_t) - f'(x_t, a_t))^2}{1 + \sum_{t \in \mathcal{T}_i} w_\tau(f(x_\tau, a_\tau) - f'(x_\tau, a_\tau))^2}$$

$$\overset{(i)}{\leq} 2 \cdot \frac{(f(x_t, a_t) - f'(x_t, a_t))^2}{1/(2^i/T) + \sum_{\tau \in \mathcal{T}_i, \tau < t}(f(x_\tau, a_\tau) - f'(x_\tau, a_\tau))^2}$$

$$\overset{(ii)}{\leq} 2 \cdot \frac{(f(x_t, a_t) - f'(x_t, a_t))^2}{1/B + \sum_{\tau \in \mathcal{T}_i, \tau < t}(f(x_\tau, a_\tau) - f'(x_\tau, a_\tau))^2},$$

where in $(i)$ we use the fact that for any $\tau \in \mathcal{T}_i$, $w_\tau \in [2^{i-1}/T, 2^i/T]$, and in $(ii)$ we use the fact that $2^i/T \leq B$. According to Lemma D.5, we have for any $1 \leq i \leq \log(BT)$.

$$\sum_{t \in \mathcal{T}_i} \min\left\{\lambda, \sup_{f,f' \in \mathcal{F}_t} \frac{(f(x_t, a_t) - f'(x_t, a_t))^2}{1/B + \sum_{\tau \in \mathcal{T}_i, \tau < t}(f(x_\tau, a_\tau) - f'(x_\tau, a_\tau))^2}\right\} \leq d_{\text{elu}}(1/\sqrt{BT})(\lambda + \log|\mathcal{T}_i|)$$

$$\leq d_{\text{elu}}(1/\sqrt{BT})(\lambda + \log T).$$

Next we notice that for those $t \in \mathcal{T}_0$, we have $w_t \leq 1/T$, which implies that

$$\sum_{t \in \mathcal{T}_i} \min\left\{\lambda, \sup_{f,f' \in \mathcal{F}_t} \frac{w_t(f(x_t, a_t) - f'(x_t, a_t))^2}{1 + \sum_{\tau < t} w_\tau(f(x_\tau, a_\tau) - f'(x_\tau, a_\tau))^2}\right\} \leq T \cdot 1/T = 1.$$

$$\sum_{i=1}^{T} \min\left\{\lambda, \sup_{f,f' \in \mathcal{F}_t} \frac{w_t(f(x_t, a_t) - f'(x_t, a_t))^2}{1 + \sum_{\tau=1}^{t-1} w_\tau(f(x_\tau, a_\tau) - f'(x_\tau, a_\tau))^2}\right\}$$

$$\leq 2d_{\text{elu}}(1/\sqrt{BT})(\lambda + \log T)\log(BT) + 1 \leq 3d_{\text{elu}}(1/\sqrt{BT})(\lambda + \log T)\log(BT).$$

$\square$

Next, we present a lemma bounding the single-step regret in terms of $g_t$ defined in Eq. (4) in Algorithm 1.

**Lemma D.7.** *Suppose $f^\star \in \mathcal{F}_t$ for any $1 \leq t \leq T$. For any $1 \leq t \leq T$ we have*

$$\max_{a \in \mathcal{A}_t} f^\star(x_t, a) - f^\star(x_t, a_t) \leq 21\sqrt{L} \cdot \max_a g_t(a)$$

**Proof.** In the following, we assume $f^\star \in \mathcal{F}_t$ always holds. Let

$$a_t^{\mathsf{ucb}} = \arg\max_{a \in \mathcal{A}_t} \max_{f \in \mathcal{F}_t} f(x_t, a).$$

Then we have

$$\max_{a \in \mathcal{A}_t} f^\star(x_t, a) - f^\star(x_t, a_t)$$

$$\overset{(i)}{\leq} \max_{f \in \mathcal{F}_t} f(x_t, a_t^{\mathsf{ucb}}) - f^\star(x_t, a_t)$$

$$= \max_{f \in \mathcal{F}_t} f(x_t, a_t^{\mathsf{ucb}}) - \min_{f \in \mathcal{F}_t} f(x_t, a_t^{\mathsf{ucb}}) + \min_{f \in \mathcal{F}_t} f(x_t, a_t^{\mathsf{ucb}}) - \max_{f \in \mathcal{F}_t} f(x_t, a_t) + \max_{f \in \mathcal{F}_t} f(x_t, a_t) - f^\star(x_t, a_t)$$

$$\leq 2 \max_{f, f' \in \mathcal{F}_t} \max_{a \in \mathcal{A}_t} |f(x_t, a) - f'(x_t, a)| + \min_{f \in \mathcal{F}_t} f(x_t, a_t^{\mathsf{ucb}}) - \max_{f \in \mathcal{F}_t} f(x_t, a_t)$$

$$\overset{(ii)}{\leq} 2 \max_{f, f' \in \mathcal{F}_t} \max_{a \in \mathcal{A}_t} |f(x_t, a) - f'(x_t, a)|$$

$$\overset{(iii)}{\leq} 21 \max_{f, f' \in \mathcal{F}_t} \max_{a \in \mathcal{A}_t} \frac{|f(x_t, a) - f'(x_t, a)|}{\sqrt{1 + \sum_{\tau < t} w_\tau (f(x_\tau, a_\tau) - f'(x_\tau, a_\tau))^2}} \times \sqrt{L}.$$

Here in $(i)$ we use the fact that $f^\star \in \mathcal{F}_t$, and in $(ii)$ we first use the definition of $\mathcal{A}_t$ in Line 4 of Algorithm 1 that there exists $f' \in \mathcal{F}_t$ such that $a_t = \max_{a \in \mathcal{A}} f'(x_t, a)$, which implies

$$\min_{f \in \mathcal{F}_t} f(x_t, a_t^{\mathsf{ucb}}) \leq f'(x_t, a_t^{\mathsf{ucb}}) \leq f'(x_t, a_t) \leq \max_{f \in \mathcal{F}_t} f(x_t, a_t).$$

In $(iii)$ we use the definition of Eq. (5) that for any $f, f' \in \mathcal{F}_t$,

$$1 + \sum_{\tau < t} w_\tau (f(x_\tau, a_\tau) - f'(x_\tau, a_\tau))^2 \leq 1 + 102L \leq 103L.$$

$\square$

Our next lemma provides an upper bound to the expectation of regret for rounds falling into $\mathcal{T}_1$.

**Lemma D.8.** *Suppose $f^\star \in \mathcal{F}_t$ for any $1 \leq t \leq T$. We have*

$$\sum_{t \in \mathcal{T}_1} (\max_{a \in \mathcal{A}} f^\star(x_t, a) - f^\star(x_t, a_t)) \leq \tilde{O}\left(\frac{\sigma^2 d_{elu}\sqrt{L}}{\Delta} + d_{elu}L\right),$$

*where we use $\tilde{\mathcal{O}}$ to hide constants and factors of $\log\left(\frac{T}{\sigma\Delta}\right)$.*

**Proof.** We first define $M = \log \frac{1}{\Delta\sqrt{T}}$ and $\beta = 21\sqrt{L}$ and recall the definition of $g_t$ in Eq. (4).

$$g_t(a_t) = \sup_{f, f' \in \mathcal{F}_t} \frac{|f(x_t, a_t) - f'(x_t, a_t)|}{\sqrt{1 + \sum_{\tau=1}^{t-1} w_\tau (f(x_\tau, a_\tau) - f'(x_\tau, a_\tau))^2}}.$$

In the following, when with no ambiguity, we write $g_t = g_t(a_t)$. Since for any $f \in \mathcal{F}$, we have $f(x, a) \in [0, 1]$, we have $|g_t| \leq 1$. Also notice that for any $t \in \mathcal{T}_1$, we have $g_t \geq \Delta$ according to Algorithm 1.

We further define subsets $\mathcal{T}_{11h}, \mathcal{T}_{12h} \subset [T]$ for every $h = \Delta \cdot 2^i$ with $i = 0, 1, 2, \cdots, M-1$ as

$$\mathcal{T}_{11h} \triangleq \mathcal{T}_1 \cap \{t : h \leq g_t \leq 2h\} \cap \left\{t : w_t = \frac{1}{\sigma^2}\right\},$$

$$\text{and} \quad \mathcal{T}_{12h} \triangleq \mathcal{T}_1 \cap \{t : h \leq g_t \leq 2h\} \cap \left\{t : w_t \neq \frac{1}{\sigma^2}\right\}.$$

Since $\Delta \leq g_t \leq 1$ for $t \in \mathcal{T}_1$, we have

$$\mathcal{T}_1 \subset \bigcup_{i=0}^{M-1} \mathcal{T}_{11(2^i\Delta)} \bigcup_{i=0}^{M-1} \mathcal{T}_{12(2^i\Delta)}. \tag{14}$$

Next, we fix $h$, we analyze $\mathcal{T}_{11h}$ and $\mathcal{T}_{12h}$ separately. For those $t \in \mathcal{T}_{11h}$, we have:

$$\sum_{t \in \mathcal{T}_{11h}} (\max_{a \in \mathcal{A}_t} f^\star(x_t, a) - f^\star(x_t, a_t))$$

$$\overset{(i)}{\leq} \sum_{t \in \mathcal{T}_{11h}} \min\left\{1, \beta \max_{f,f' \in \mathcal{F}_t} \frac{|f(x_t, a_t) - f'(x_t, a_t)|}{\sqrt{1 + \sum_{\tau < t} w_\tau (f(x_\tau, a_\tau) - f'(x_\tau, a_\tau))^2}}\right\}$$

$$\overset{(ii)}{=} \sum_{t \in \mathcal{T}_{11h}} \min\{1, \beta g_t\}$$

$$\overset{(iii)}{\leq} \min\{1, 2\beta h\}|\mathcal{T}_{11h}|,$$

where in $(i)$ we use Lemma D.7 and the fact that $|f^\star(x, a) - f^\star(x, a')| \leq 1$ for any action $a, a' \in \mathcal{A}$, in $(ii)$ we use the definition of $g_t$ in Eq. (4) and the simplification $g_t := g_t(a_t)$, and in $(iii)$ we use the definition of $\mathcal{T}_{11h}$ that for any $t \in \mathcal{T}_{11h}$ we always have $g_t \leq 2h$. Next, we bound the cardinality of each $\mathcal{T}_{11h}$:

$$|\mathcal{T}_{11h}| \overset{(i)}{\leq} \sum_{t \in \mathcal{T}_{11h}} \mathbb{I}\{g_t \geq h\} \mathbb{I}\left\{w_t = \frac{1}{\sigma^2}\right\}$$

$$\overset{(ii)}{\leq} \frac{1}{h} \sum_{t \in \mathcal{T}_{11h}} \min\{h, g_t\} \mathbb{I}\left\{w_t = \frac{1}{\sigma^2}\right\}$$

$$\overset{(iii)}{\leq} \frac{\sigma}{h} \sum_{t \in \mathcal{T}_{11}} \min\left\{\frac{h}{\sigma}, \sqrt{w_t} g_t\right\}$$

$$\overset{(iv)}{\leq} \frac{\sigma}{h} \sqrt{|\mathcal{T}_{11h}|} \sqrt{\sum_{t \in \mathcal{T}_{11h}} \min\left\{\frac{h^2}{\sigma^2}, w_t g_t^2\right\}}$$

$$\overset{(v)}{=} \frac{\sigma}{h} \sqrt{|\mathcal{T}_{11h}|} \sqrt{\sum_{t \in \mathcal{T}_{11h}} \min\left\{\frac{h^2}{\sigma^2}, \sup_{f,f' \in \mathcal{F}_t} \frac{w_t(f(x_t, a_t) - f'(x_t, a_t))^2}{1 + \sum_{\tau \leq t-1} w_\tau (f(x_\tau, a_\tau) - f'(x_\tau, a_\tau))^2}\right\}}$$

$$\overset{(vi)}{\leq} \frac{\sigma}{h} \sqrt{3|\mathcal{T}_{11h}| d_{\text{elu}}\left(\frac{\sigma}{\sqrt{T}}\right) \left(\log T + \frac{h^2}{\sigma^2}\right) \log\left(\frac{T}{\sigma^2}\right)}$$

where in $(i)$ we use the definition of $\mathcal{T}_{11h}$, in $(ii)$ we merely use the inequality that $\mathbb{I}\{g \geq h\} \leq \frac{1}{h}\min\{g, h\}$ for any $g, h \geq 0$, in $(iii)$ we use the the fact that $\mathbb{I}\{w_t = 1/\sigma^2\} \leq \sqrt{w_t}\sigma$ since $w_t \leq 1/\sigma^2$ always holds, in $(iv)$ we use Cauchy-Schwarz inequality, in $(v)$ we use the definition of $g_t$ in Eq. (4), and finanly in $(vi)$ we use Lemma D.6 with $\lambda = h^2/\sigma^2$ and also $w_t \in [0, 1/\sigma^2]$. Therefore, we obtain that

$$|\mathcal{T}_{11h}| \leq 3\left(\frac{\sigma^2 \log T}{h^2} + 1\right) d_{\text{elu}}\left(\frac{\sigma}{\sqrt{T}}\right) \log\left(\frac{T}{\sigma^2}\right),$$

which implies

$$\sum_{t \in \mathcal{T}_{11h}} (\max_{a \in \mathcal{A}_t} f^\star(x_t, a) - f^\star(x_t, a_t))$$

$$\leq \min\{6\beta h, 3\} \left(\frac{\sigma^2 \log T}{h^2} + 1\right) d_{\text{elu}}\left(\frac{\sigma}{\sqrt{T}}\right) \log\left(\frac{T}{\sigma^2}\right)$$

$$\leq \left(\frac{6\beta\sigma^2 \log T}{h} + 3\right) d_{\text{elu}}\left(\frac{\sigma}{\sqrt{T}}\right) \log\left(\frac{T}{\sigma^2}\right)$$

$$\leq \left(\frac{126\sigma^2 \log T \sqrt{L}}{h} + 3\right) d_{\text{elu}}\left(\frac{\sigma}{\sqrt{T}}\right) \log\left(\frac{T}{\sigma^2}\right).$$

Next, we will deal with those $t$ in $\mathcal{T}_{12h}$. Similar to the proof for $\mathcal{T}_{11h}$, for any fixed $h$ we have

$$\sum_{t \in \mathcal{T}_{12h}} (\max_{a \in \mathcal{A}_t} f^\star(x_t, a) - f^\star(x_t, a_t)) \leq \min\{1, 2\beta h\}|\mathcal{T}_{12h}|. \tag{15}$$

And we can upper bound the cardinality of $\mathcal{T}_{12h}$ as

$$|\mathcal{T}_{12h}| = \sum_{t \in \mathcal{T}_{12h}} \mathbb{I}\{g_t \geq h\} \mathbb{I}\left\{w_t \neq \frac{1}{\sigma^2}\right\}$$

$$\leq \frac{1}{h} \sum_{t \in \mathcal{T}_{12h}} \min\{h, g_t\} \mathbb{I}\left\{w_t \neq \frac{1}{\sigma^2}\right\}$$

$$\overset{(i)}{=} \frac{1}{h} \sum_{t \in \mathcal{T}_{12h}} \min\left\{h, \sup_{f,f' \in \mathcal{F}_t} \frac{|f(x_t, a_t) - f'(x_t, a_t)|}{\sqrt{1 + \sum_{\tau=1}^{t-1} w_\tau (f(x_\tau, a_\tau) - f'(x_\tau, a_\tau))^2}}\right\} \mathbb{I}\left\{w_t \neq \frac{1}{\sigma^2}\right\}$$

$$\overset{(ii)}{\leq} \frac{1}{h} \sum_{t \in \mathcal{T}_{12h}} \min\left\{h, \sqrt{L} \sup_{f,f' \in \mathcal{F}_t} \frac{w_t(f(x_t, a_t) - f'(x_t, a_t))^2}{1 + \sum_{\tau=1}^{t-1} w_\tau (f(x_\tau, a_\tau) - f'(x_\tau, a_\tau))^2}\right\}$$

$$\overset{(iii)}{\leq} \frac{3}{h} d_{\text{elu}}\left(\frac{\sigma}{\sqrt{T}}\right)\left(\sqrt{L} \log T + h\right) \log\left(\frac{T}{\sigma^2}\right),$$

where in $(i)$ we use the definition of $g_t$ in Eq. (4), in $(ii)$ we use the definition of $w_t$ that when $w_t \neq 1/\sigma^2$, we always have

$$w_t = \min_{f,f' \in \mathcal{F}_t} \frac{\sqrt{1 + \sum_{\tau=1}^{t-1} w_\tau (f(x_\tau, a_\tau) - f'(x_\tau, a_\tau))^2}}{|f(x_t, a_t) - f'(x_t, a_t)|\sqrt{L}},$$

and $(iii)$ is according to Lemma D.6 and the fact that $w_t \in [0, 1/\sigma^2]$.

Therefore, according to Eq. (15), we obtain that

$$\sum_{t \in \mathcal{T}_{12h}} (\max_{a \in \mathcal{A}_t} f^\star(x_t, a) - f^\star(x_t, a_t)) \leq \min\{1, 2\beta h\}|\mathcal{T}_{12h}|$$

$$\leq \left(6\beta\sqrt{L} \log T + 3\right) d_{\text{elu}}\left(\frac{\sigma}{\sqrt{T}}\right) \log\left(\frac{T}{\sigma^2}\right)$$

$$= (126 L \log T + 3) d_{\text{elu}}\left(\frac{\sigma}{\sqrt{T}}\right) \log\left(\frac{T}{\sigma^2}\right).$$

Finally, recalling Eq. (14), if we sum the regret obtained above over $h$, we obtain

$$\sum_{t \in \mathcal{T}_1} (\max_{a \in \mathcal{A}_t} f^\star(x_t, a) - f^\star(x_t, a_t))$$

$$= \sum_{i=0}^{M-1} \left[\sum_{t \in \mathcal{T}_{11(2^i\Delta)}} \left(\max_{a \in \mathcal{A}_t} f^\star(x_t, a) - f^\star(x_t, a_t)\right) + \sum_{t \in \mathcal{T}_{12(2^i\Delta)}} \left(\max_{a \in \mathcal{A}_t} f^\star(x_t, a) - f^\star(x_t, a_t)\right)\right]$$

$$\lesssim \sum_{i=0}^{M-1} \left[\frac{d_{\text{elu}}\sigma^2\sqrt{L}}{(2^i\Delta)} + d_{\text{elu}} + d_{\text{elu}}L + d_{\text{elu}}\right] \log\left(\frac{T}{\sigma}\right)$$

(using $d_{\text{elu}} \triangleq d_{\text{elu}}(1/T^2)$ from Section 2 and the fact that $d_{\text{elu}}(\alpha)$ is decreasing in $\alpha$)

$$= \tilde{\mathcal{O}}\left(\frac{\sigma^2 d_{\text{elu}}\sqrt{L}}{\Delta} + d_{\text{elu}}L\right).$$

$\square$

Finally, we provide a lemma which upper bounds the expectation of regret for rounds falling into $\mathcal{T}_2$.

**Lemma D.9.** *Suppose that for any $t \in [T]$, we have $f^\star \in \mathcal{F}_t$. With $\tilde{\Delta} = 11\Delta\sqrt{L}$, we have with probability at least $1 - \delta$,*

$$\sum_{t \in \mathcal{T}_2}(\max_a f^\star(x_t, a) - f^\star(x_t, a_t)) \leq 16\sqrt{(\sigma^2 + \tilde{\Delta})T\log\left(\frac{4|\mathcal{F}|}{\delta}\right)} + 3\log\left(\frac{2}{\delta}\right).$$

**Proof.** We let $a_t^\star = \arg\max_a f^\star(x_t, a)$. According to (Foster and Rakhlin, 2020, Lemma 3), we have

$$\sum_{a \in \mathcal{A}} p_t(a)(f^\star(x_t, a_t^\star) - f^\star(x_t, a)) \leq \frac{2A}{\gamma} + \frac{\gamma}{4}\sum_{a \in \mathcal{A}} p_t(a)\left[(f_t(x_t, a) - f^\star(x_t, a))^2\right]$$

Summing this inequality up for all $t \in \mathcal{T}_2$, we obtain

$$\sum_{t \in \mathcal{T}_2}\sum_{a \in \mathcal{A}} p_t(a)(f^\star(x_t, a_t^\star) - f^\star(x_t, a)) \leq \frac{2A|\mathcal{T}_2|}{\gamma} + \frac{\gamma}{4}\sum_{t \in \mathcal{T}_2}\sum_{a \in \mathcal{A}} p_t(a)(f_t(x_t, a_t) - f^\star(x_t, a))^2.$$

We notice that for any $t \in \mathcal{T}_2$, we have

$$\sup_{f,f' \in \mathcal{F}_t}\max_{a \in \mathcal{A}}\frac{|f(x_t, a) - f'(x_t, a)|}{\sqrt{1 + \sum_{\tau=1}^{t-1} w_\tau(f(x_\tau, a_\tau) - f'(x_\tau, a_\tau))^2}} \leq \Delta.$$

According to the definition of $\mathcal{F}_t$, for any $f, f' \in \mathcal{F}_t$,

$$\sum_{\tau=1}^{t-1} w_\tau(f(x_\tau, a_\tau) - f'(x_\tau, a_\tau))^2 \leq 102L,$$

which implies that for any $f, f' \in \mathcal{F}_t$,

$$\max_{a \in \mathcal{A}}|f(x_t, a) - f'(x_t, a)| \leq \Delta\sqrt{1 + 102L} \leq \tilde{\Delta}.$$

Hence according to Lemma D.2, with probability at least $1 - \delta/2$, we have

$$\sum_{t=1}^{T}\sum_{a \in \mathcal{A}} p_t(a)(f_t(x_t, a) - f^\star(x_t, a))^2 \leq 48(\sigma^2 + \tilde{\Delta})\left(\frac{4|\mathcal{F}|}{\delta}\right).$$

Further noticing that $|\mathcal{T}_2| \leq T$, with choice $\gamma = \sqrt{\frac{AT}{4(\tilde{\Delta}+\sigma^2)\log(|\mathcal{F}|/\delta)}}$, we have with probability at least $1 - \delta/2$,

$$\sum_{t \in \mathcal{T}_2}\sum_{a \in \mathcal{A}} p_t(a)(f^\star(x_t, a_t^\star) - f^\star(x_t, a)) \leq \sqrt{48(\sigma^2 + \tilde{\Delta})T\log\left(\frac{4|\mathcal{F}|}{\delta}\right)}.$$

Finally, since we always have $0 \leq f^\star(x_t, a_t^\star) - f^\star(x_t, a) \leq 1$, according to Bernstein inequality we have with probability at least $1 - \delta/2$,

$$\left|\sum_{t \in \mathcal{T}_2}(f^\star(x_t, a_t^\star) - f^\star(x_t, a_t)) - \sum_{t \in \mathcal{T}_2}\sum_{a \in \mathcal{A}} p_t(a)(f^\star(x_t, a_t^\star) - f^\star(x_t, a))\right|$$

$$\leq 2\sqrt{\sum_{t \in \mathcal{T}_2}\sum_{a \in \mathcal{A}} p_t(a)(f^\star(x_t, a_t^\star) - f^\star(x_t, a))^2\log\left(\frac{2}{\delta}\right)} + 2\log\left(\frac{2}{\delta}\right)$$

$$\leq \sum_{t\in\mathcal{T}_2}\sum_{a\in\mathcal{A}} p_t(a)(f^\star(x_t,a_t^\star)-f^\star(x_t,a))^2 + 3\log\left(\frac{2}{\delta}\right)$$

$$\leq \sum_{t\in\mathcal{T}_2}\sum_{a\in\mathcal{A}} p_t(a)(f^\star(x_t,a_t^\star)-f^\star(x_t,a)) + 3\log\left(\frac{2}{\delta}\right),$$

where in the second inequality we use the AM-GM inequality and in the last inequality we use the fact that $0 \leq f^\star(x_t,a_t^\star) - f^\star(x_t,a) \leq 1$. Therefore, we obtain that with probability at least $1-\delta$,

$$\sum_{t\in\mathcal{T}_2}(f^\star(x_t,a_t^\star)-f^\star(x_t,a_t)) \leq 2\sum_{t\in\mathcal{T}_2}\sum_{a\in\mathcal{A}} p_t(a)(f^\star(x_t,a_t^\star)-f^\star(x_t,a)) + 3\log\left(\frac{2}{\delta}\right)$$

$$\leq 16\sqrt{(\sigma^2+\tilde{\Delta})T\log\left(\frac{4|\mathcal{F}|}{\delta}\right)} + 3\log\left(\frac{2}{\delta}\right).$$

$\square$

**Proof of Theorem 4.2.** Lemma D.4 and Lemma D.8 gives that with probability at least $1-\delta/2$ we have

$$\sum_{t\in\mathcal{T}_1}(\max_{a\in\mathcal{A}} f^\star(x_t,a) - f^\star(x_t,a_t)) = \tilde{O}\left(\frac{\sigma^2 d_{\text{elu}}\sqrt{L}}{\Delta} + d_{\text{elu}}L\right).$$

Additionally, according to Lemma D.9, we have with probability at least $1-\delta/2$

$$\sum_{t\in\mathcal{T}_2}(\max_{a\in\mathcal{A}} f^\star(x_t,a) - f^\star(x_t,a_t)) \leq 16\sqrt{(\sigma^2+\tilde{\Delta})TL} + 3L.$$

Hence when $\Delta = \sigma^2/(11\sqrt{L})$ (so $\tilde{\Delta}=\sigma^2$), with probability at least $1-\delta$, we have

$$\sum_{t\in[T]}(\max_{a\in\mathcal{A}} f^\star(x_t,a) - f^\star(x_t,a_t))$$

$$= \sum_{t\in\mathcal{T}_1}(\max_{a\in\mathcal{A}} f^\star(x_t,a) - f^\star(x_t,a_t)) + \sum_{t\in\mathcal{T}_2}(\max_{a\in\mathcal{A}} f^\star(x_t,a) - f^\star(x_t,a_t))$$

$$= \tilde{\mathcal{O}}\left(\sqrt{\sigma^2 ATL} + d_{\text{elu}}L\right).$$

Then noticing that $L = \log\frac{|\mathcal{F}|T^2}{\sigma^2\delta}$ and that $\sigma \geq \frac{1}{AT}$ finishes the proof. $\square$

# E Omitted Proofs in Section 4.3

In this section, we will prove Corollary 4.1.

**Proof of Corollary 4.1.** Based on the variance $\sigma_t^2$ at round $t$, Algorithm 2 classify rounds into $\mathcal{T}_0,\cdots,\mathcal{T}_{\log T}$. We will bound the regret of rounds in $\mathcal{T}_i$ separately.

According to Algorithm 2, for $t\in\mathcal{T}_i$ with $0 \leq i \leq \log(AT)$, we always have $1/AT \leq \sigma_t \leq 2^i/AT$. Hence, according to Theorem 4.2, with probability at least $1 - \delta/\log(AT)$, we have

$$\sum_{t\in[\mathcal{T}_i]}(\max_{a\in\mathcal{A}} f^\star(x_t,a) - f^\star(x_t,a_t)) = \tilde{\mathcal{O}}\left(\sqrt{A|\mathcal{T}_i|\cdot(2^i/(AT))^2\log\left(\frac{|\mathcal{F}|}{\delta}\right)} + d_{\text{elu}}\log\left(\frac{|\mathcal{F}|}{\delta}\right)\right).$$

We further observe that for $1 \leq i \leq \log(AT)$, $\sigma_t \geq 2^{i-1}/AT$, which implies that

$$|\mathcal{T}_i|\cdot(2^i/(AT))^2 \leq 4\sum_{t\in\mathcal{T}_i}\sigma_t^2.$$

And for $i=0$, we have

$$|\mathcal{T}_i|\cdot(2^i/(AT))^2 \leq T\cdot\frac{1}{(AT)^2} \leq \frac{1}{AT}.$$

Therefore, with probability at least $1 - \delta$, we have

$$\sum_{t=1}^{T} (\max_{a \in \mathcal{A}} f^{\star}(x_t, a) - f^{\star}(x_t, a_t))$$

$$= \tilde{\mathcal{O}} \left( \sum_{i=0}^{\log(AT)} \sqrt{A|\mathcal{T}_i| \cdot (2^i/(AT))^2 \log\left(\frac{|\mathcal{F}|}{\delta}\right)} + d_{\mathrm{elu}} \log\left(\frac{|\mathcal{F}|}{\delta}\right) \right)$$

$$= \tilde{\mathcal{O}} \left( \sum_{i=1}^{\log(AT)} \sqrt{A \sum_{t \in \mathcal{T}_i} \sigma_t^2 \log\left(\frac{|\mathcal{F}|}{\delta}\right)} + d_{\mathrm{elu}} \log\left(\frac{|\mathcal{F}|}{\delta}\right) \right)$$

$$\overset{(i)}{=} \tilde{\mathcal{O}} \left( \sqrt{\log(AT) \cdot \sum_{i=1}^{\log(AT)} A \sum_{t \in \mathcal{T}_i} \sigma_t^2 \log\left(\frac{|\mathcal{F}|}{\delta}\right)} + d_{\mathrm{elu}} \log\left(\frac{|\mathcal{F}|}{\delta}\right) \right)$$

$$= \tilde{\mathcal{O}} \left( \sqrt{A \sum_{t=1}^{T} \sigma_t^2 \log\left(\frac{|\mathcal{F}|}{\delta}\right)} + d_{\mathrm{elu}} \log\left(\frac{|\mathcal{F}|}{\delta}\right) \right),$$

where in $(i)$ we use Cauchy-Schwarz inequality. $\qquad \square$

---

**Algorithm 5** VarUCB

---

1: **Define**: $L = \Theta(\log(|\mathcal{F}|T/\delta))$, $\mathcal{F}_0 = \mathcal{F}$.
2: **Define**: $K \triangleq \lceil \log T \rceil$. Let $\Psi_{1,k} = \emptyset$ for $k = 1, 2, \ldots, K, K+1$.
3: **for** $t = 1, 2, \ldots$ **do**
4:     Define confidence set:

$$\mathcal{F}_t = \left\{ f \in \mathcal{F}_{t-1} : \forall k \in [K], \sum_{\tau \in \Psi_{t,k}} w_\tau^2 (f(x_\tau, a_\tau) - \hat{f}_{t,k}(x_\tau, a_\tau))^2 \leq \beta_{t,k}^2 \right\},$$

where

$$\hat{f}_{t,k} = \arg\min_{f \in \mathcal{F}_t} \sum_{\tau \in \Psi_{t,k}} w_\tau^2 (f(x_\tau, a_\tau) - r_\tau)^2, \tag{16}$$

$$\beta_{t,k} = \begin{cases} 10 \cdot 2^{-k} \sqrt{\sum_{\tau \in \Psi_{t,k}} w_\tau^2 (r_\tau - \hat{f}_{t,k}(x_\tau, a_\tau))^2 L + L^2} & \text{if } k \leq K \text{ and } 2^{2k} \geq 80L \\ \sqrt{|\Psi_{t,k}|} & \text{if } k \leq K \text{ and } 2^{2k} < 80L \end{cases}$$

5:     Receive $x_t$, and define $\mathcal{A}_t = \{a \in \mathcal{A} : \exists f \in \mathcal{F}_t : a \in \arg\max_{a' \in \mathcal{A}} f(x_t, a')\}$.
6:     Define

$$g_{t,k}(a) = \max_{f, f' \in \mathcal{F}_t} \frac{|f(x_t, a) - f'(x_t, a)|}{\sqrt{2^{-2k}L^2 + \sum_{\tau \in \Psi_{t,k}} w_\tau^2 (f(x_\tau, a_\tau) - f'(x_\tau, a_\tau))^2}}.$$

7:     Let $k_t$ be the smallest $k \in [K]$ such that $\max_{a \in \mathcal{A}_t} g_{t,k}(a) \geq 2^{-k}$.
        (Let $k_t = K+1$ if such $k_t$ does not exist)
8:     Play $a_t = \begin{cases} \arg\max_{a \in \mathcal{A}_t} g_{t,k_t}(a) & \text{if } k \leq K \\ \arg\max_{a \in \mathcal{A}_t} \max_{f \in \mathcal{F}_t} f(x_t, a) & \text{otherwise} \end{cases}$, and receive $r_t$.
9:     Define $g_t = g_{t,k_t}(a_t)$. Let $w_t = 2^{-k_t}/g_t$ if $k_t \leq K$ and $w_t = 1$ if $k_t = K+1$.
10:    Update $\Psi_{t+1,k_t} \leftarrow \Psi_{t,k_t} \cup \{t\}$ and $\Psi_{t+1,k} \leftarrow \Psi_{t,k}$ for $k \neq k_t$.

---

# F   Algorithm for the Strong Adversary Case and Omitted Proofs in Section 5

## F.1   Upper Bound

In this section, we introduce and analyze Algorithm 5, an algorithm that achieves $\tilde{\mathcal{O}}(d_{\text{elu}}\sqrt{\Lambda \log |\mathcal{F}|} + d_{\text{elu}} \log |\mathcal{F}|)$ regret bound against strong adversary.

Algorithm 5 is an extension of the SAVE algorithm (Zhao et al., 2023) from linear function approximation to general function approximation. The algorithm maintains $K + 1 = \Theta(\log T)$ bins denoted as $\{\Psi_{t,k}\}$, $k = 1, 2, \ldots, K+1$, which forms a partition of $[t-1]$ (i.e., every time index $\tau < t$ falls into exactly one of these bins). Each bin $k$ can form a confidence function set like in standard LinUCB using samples in $\Psi_{t,k}$. The overall confidence set $\mathcal{F}_t$ is the intersection of the confidence sets of individual bins (Line 4).

Upon receiving the context $x_t$, the learner form an active action set that contains plausible actions (Line 5). The next step is to decide which bin $t$ should go to. This is done by leveraging the uncertainty measure $g_{t,k}(a)$ for bin $k$ and action $a$ (Line 6), which measures how uncertain the reward of action $a$ is, given prior samples in bin $k$. The measure $g_{t,k}(a)$ corresponds to the quantity $\|a\|_{\Sigma_{t,k}^{-1}}$ usually seen in linear contextual bandits, where $\Sigma_{t,k}$ is the covariance matrix formed by the samples in $\Psi_{t,k}$. The algorithm finds the smallest $k$ such that there exists an action with relative large uncertainty $g_{t,k}(a) \geq 2^{-k}$ (Line 7). The learner would then choose this action in order to gain relatively large shrinking in bin $k$'s confidence set (Line 8), and put time $t$ in bin $k$. The sample at time $t$ is assigned a weight $w_t$ that is inversely proportional to the uncertainty measure (Line 9). This ensures that the importance of the samples within each bin is more balanced.

Before proving the main theorem Theorem 5.2, we first establish lemmas Lemma F.1–Lemma F.5.

**Lemma F.1.** *Suppose that $f^\star \in \mathcal{F}_{t-1}$. Then with probability at least $1 - \delta/T$, for all $k \in [K]$ we have*

$$\sum_{\tau \in \Psi_{t,k}} w_\tau^2 (\hat{f}_{t,k}(x_\tau, a_\tau) - f^\star(x_\tau, a_\tau))^2 \le 10 \cdot 2^{-2k} \left( \sum_{\tau \in \Psi_{t,k}} w_\tau^2 \sigma_\tau^2 L + L^2 \right).$$

**Proof.** Since $r_\tau = f^*(x_\tau, a_\tau) + \epsilon_t$, we have

$$\sum_{\tau \in \Psi_{t,k}} w_\tau^2 (\hat{f}_{t,k}(x_\tau, a_\tau) - f^\star(x_\tau, a_\tau))^2$$

$$= \sum_{\tau \in \Psi_{t,k}} w_\tau^2 \left( \hat{f}_{t,k}(x_\tau, a_\tau) - r_\tau \right)^2 - \sum_{\tau \in \Psi_{t,k}} w_\tau^2 \left( f^\star(x_\tau, a_\tau) - r_\tau \right)^2$$

$$+ 2 \sum_{\tau \in \Psi_{t,k}} w_\tau^2 (\hat{f}_{t,k}(x_\tau, a_\tau) - f^\star(x_\tau, a_\tau))(r_\tau - f^\star(x_\tau, a_\tau))$$

$$\le 2 \sum_{\tau \in \Psi_{t,k}} w_\tau^2 \epsilon_\tau (\hat{f}_{t,k}(x_\tau, a_\tau) - f^\star(x_\tau, a_\tau)), \tag{17}$$

where the last inequality is by the optimality of $\hat{f}_{t,k}$ given in Eq. (16). According to the strengthened Freeman's Inequality (Lemma B.5), with $L = C \log(|\mathcal{F}|T/\delta)$ for some large enough universal constant $C$ (specified in Line 1 of Algorithm 5), the last expression in Eq. (17) can be further bounded by

$$\sqrt{\sum_{\tau \in \Psi_{t,k}} w_\tau^4 \sigma_\tau^2 (\hat{f}_{t,k}(x_\tau, a_\tau) - f^\star(x_\tau, a_\tau))^2 L} + \max_{\tau \in \Psi_{t,k}} \left| w_\tau^2 \epsilon_\tau (\hat{f}_{t,k}(x_\tau, a_\tau) - f^\star(x_\tau, a_\tau)) \right| L$$

$$\le \sqrt{\sum_{\tau \in \Psi_{t,k}} w_\tau^4 \sigma_\tau^2 (\hat{f}_{t,k}(x_\tau, a_\tau) - f^\star(x_\tau, a_\tau)^2 L} + \max_{\tau \in \Psi_{t,k}} w_\tau |\hat{f}_{t,k}(x_\tau, a_\tau) - f^\star(x_\tau, a_\tau)| L,$$

$$(w_t \le 1 \text{ because } g_t = g_{t,k_t}(a_t) \ge 2^{-k_t} \text{ for } k \in [K], \text{ and } |\epsilon_t| \le 1)$$

$$\le \left( \max_{\tau \in \Psi_{t,k}} w_\tau |\hat{f}_{t,k}(x_\tau, a_\tau) - f^\star(x_\tau, a_\tau)| \right) \left( \sqrt{\sum_{\tau \in \Psi_{t,k}} w_\tau^2 \sigma_\tau^2 L} + L \right) \tag{18}$$

for all $k$ with probability at least $1 - \delta/T$ by a union bound over $k$ and $\hat{f}_{t,k}$.

For any $\tau \in \Psi_{t,k}$, by the definition of $w_\tau$ (Line 9 of Algorithm 5), we have

$$w_\tau |\hat{f}_{t,k}(x_\tau, a_\tau) - f^\star(x_\tau, a_\tau)|$$

$$\le 2^{-k}/g_\tau \cdot |\hat{f}_{t,k}(x_\tau, a_\tau) - f^\star(x_\tau, a_\tau)|$$

$$= 2^{-k} \min_{f, f' \in \mathcal{F}_\tau} \frac{\sqrt{2^{-2k} L^2 + \sum_{s \in \Psi_{\tau,k}} w_s^2 (f(x_s, a_s) - f'(x_s, a_s))^2}}{|f(x_\tau, a_\tau) - f'(x_\tau, a_\tau)|} \cdot |\hat{f}_{t,k}(x_\tau, a_\tau) - f^\star(x_\tau, a_\tau)|$$

$$\le 2^{-k} \sqrt{2^{-2k} L^2 + \sum_{s \in \Psi_{\tau,k}} w_s^2 (\hat{f}_{t,k}(x_s, a_s) - f^\star(x_s, a_s))^2}.$$

$$(f^\star, \hat{f}_{t,k} \in \mathcal{F}_{t-1} \subset \mathcal{F}_\tau \text{ by assumption})$$

Combining this and Eq. (18), we get

$$\sum_{\tau \in \Psi_{t,k}} w_\tau^2 (\hat{f}_{t,k}(x_\tau, a_\tau) - f^\star(x_\tau, a_\tau))^2$$

$$\le 2^{-k} \left( \max_{\tau \in \Psi_{t,k}} \sqrt{2^{-2k} L^2 + \sum_{s \in \Psi_{\tau,k}} w_s^2 (\hat{f}_{t,k}(x_s, a_s) - f^\star(x_s, a_s))^2} \right) \left( \sqrt{\sum_{\tau \in \Psi_{t,k}} w_\tau^2 \sigma_\tau^2 L} + L \right)$$

$$\leq 2^{-k} \left( \sqrt{2^{-2k}L^2 + \sum_{\tau \in \Psi_{t,k}} w_\tau^2 (\hat{f}_{t,k}(x_\tau, a_\tau) - f^\star(x_\tau, a_\tau))^2} \right) \left( \sqrt{\sum_{\tau \in \Psi_{t,k}} w_\tau^2 \sigma_\tau^2 L} + L \right).$$

Solving the inequality yields

$$\sum_{\tau \in \Psi_{t,k}} w_\tau^2 (\hat{f}_{t,k}(x_\tau, a_\tau) - f^\star(x_\tau, a_\tau))^2 \leq 10 \cdot 2^{-2k} \left( \sum_{\tau \in \Psi_{t,k}} w_\tau^2 \sigma_\tau^2 L + L^2 \right).$$

$\square$

**Lemma F.2.** *Suppose that $f^\star \in \mathcal{F}_{t-1}$. Then for all $k \in [K]$ satisfying $2^{2k} \geq 80L$, we have with probability at least $1 - \delta/T$,*

$$\sum_{\tau \in \Psi_{t,k}} w_\tau^2 \sigma_\tau^2 \leq 8 \sum_{\tau \in \Psi_{t,k}} w_\tau^2 \left( r_\tau - \hat{f}_{t,k}(x_\tau, a_\tau) \right)^2 + 4L,$$

$$\sum_{\tau \in \Psi_{t,k}} w_\tau^2 \left( r_\tau - \hat{f}_{t,k}(x_\tau, a_\tau) \right)^2 \leq 2 \sum_{\tau \in \Psi_{t,k}} w_\tau^2 \sigma_\tau^2 + L.$$

**Proof.** With $L = C \log(|\mathcal{F}|T/\delta)$ for some large enough universal constant $C$, we have

$$\sum_{\tau \in \Psi_{t,k}} w_\tau^2 \sigma_\tau^2 \leq 2 \sum_{\tau \in \Psi_{t,k}} w_\tau^2 \epsilon_\tau^2 + L \hspace{3cm} \text{(Freedman's inequality)}$$

$$\leq 4 \sum_{\tau \in \Psi_{t,k}} w_\tau^2 \left( r_\tau - \hat{f}_{t,k}(x_\tau, a_\tau) \right)^2 + 4 \sum_{\tau \in \Psi_{t,k}} w_\tau^2 \left( \hat{f}_{t,k}(x_\tau, a_\tau) - f^\star(x_\tau, a_\tau) \right)^2 + L$$

$$\leq 4 \sum_{\tau \in \Psi_{t,k}} w_\tau^2 \left( r_\tau - \hat{f}_{t,k}(x_\tau, a_\tau) \right)^2 + 40 \cdot 2^{-2k} \left( \sum_{\tau \in \Psi_{t,k}} w_\tau^2 \sigma_\tau^2 L + L^2 \right) + L$$

$$\text{(by Lemma F.1)}$$

$$\leq 4 \sum_{\tau \in \Psi_{t,k}} w_\tau^2 \left( r_\tau - \hat{f}_{t,k}(x_\tau, a_\tau) \right)^2 + 2L + \frac{1}{2} \sum_{\tau \in \Psi_{t,k}} w_\tau^2 \sigma_\tau^2.$$

Rearranging gives the first inequality. For the second inequality, note that we have

$$\sum_{\tau \in \Psi_{t,k}} w_\tau^2 \left( r_\tau - \hat{f}_{t,k}(x_\tau, a_\tau) \right)^2 \leq \sum_{\tau \in \Psi_{t,k}} w_\tau^2 \left( r_\tau - f^\star(x_\tau, a_\tau) \right)^2 \hspace{1cm} \text{(by the optimality of } \hat{f}_{t,k})$$

$$\leq 2 \sum_{\tau \in \Psi_{t,k}} w_\tau^2 \sigma_\tau^2 + L. \hspace{2cm} \text{(Freedman's inequality)}$$

$\square$

**Lemma F.3.** *With probability at least $1 - \delta$, $f^\star \in \mathcal{F}_t$ for all $t$.*

**Proof.** We prove by induction. Assume that $f^\star \in \mathcal{F}_{t-1}$. Then for all $k \in [K]$ such that $2^{2k} \geq 80L$, by Lemma F.1 and Lemma F.2, with probability at least $1 - \delta/T$,

$$\sum_{\tau \in \Psi_{t,k}} w_\tau^2 (\hat{f}_{t,k}(x_\tau, a_\tau) - f^\star(x_\tau, a_\tau))^2 \leq 10 \cdot 2^{-2k} \left( \sum_{\tau \in \Psi_{t,k}} w_\tau^2 \sigma_\tau^2 L + L^2 \right)$$

$$\leq 10 \cdot 2^{-2k} \left( 8 \sum_{\tau \in \Psi_{t,k}} w_\tau^2 \left( r_\tau - \hat{f}_{t,k}(x_\tau, a_\tau) \right)^2 L + 4L^2 \right)$$

$$\leq \beta_{t,k}^2.$$

For $k \in [K]$ such that $2^{2k} < 80L$, we bound trivially

$$\sum_{\tau \in \Psi_{t,k}} w_\tau^2 (\hat{f}_{t,k}(x_\tau, a_\tau) - f^\star(x_\tau, a_\tau))^2 \leq |\Psi_{t,k}| = \beta_{t,k}^2. \qquad (w_\tau \leq 1)$$

In both cases, we have $f^\star \in \mathcal{F}_t$ by the definition of $\mathcal{F}_t$. By induction, we conclude that with probability at least $1 - \delta$, $f^\star \in \mathcal{F}_t$ for all $t$. $\qquad \square$

**Lemma F.4.** *For $k \in [K]$, we have $|\Psi_{T+1,k}| \leq \tilde{\mathcal{O}}(2^{2k} d_{elu})$.*

**Proof.** For $t$ such that $k_t \in [K]$, by the definition of $w_t$ we have

$$1 = 2^{k_t} w_t g_t = 2^{k_t} \max_{f,f' \in \mathcal{F}_t} \frac{w_t |f(x_t, a_t) - f'(x_t, a_t)|}{\sqrt{2^{-2k_t} L^2 + \sum_{\tau \in \Psi_{t,k_t}} w_\tau^2 (f(x_\tau, a_\tau) - f'(x_\tau, a_\tau))^2}}.$$

Thus,

$$
\begin{aligned}
|\Psi_{T+1,k}| &= \sum_{t \in \Psi_{T+1,k}} \min \left\{ 1, \ 2^k \max_{f,f' \in \mathcal{F}_t} \frac{w_t |f(x_t, a_t) - f'(x_t, a_t)|}{\sqrt{2^{-2k} L^2 + \sum_{\tau \in \Psi_{t,k}} w_\tau^2 (f(x_\tau, a_\tau) - f'(x_\tau, a_\tau))^2}} \right\}^2 \\
&= 2^{2k} \sum_{t \in \Psi_{T+1,k}} \min \left\{ 2^{-2k}, \ \max_{f,f' \in \mathcal{F}_t} \frac{w_t^2 (f(x_t, a_t) - f'(x_t, a_t))^2}{2^{-2k} L^2 + \sum_{\tau \in \Psi_{t,k}} w_\tau^2 (f(x_\tau, a_\tau) - f'(x_\tau, a_\tau))^2} \right\} \\
&\leq \tilde{\mathcal{O}}(2^{2k} d_{\text{elu}}). \qquad \text{(by Lemma D.5)}
\end{aligned}
$$

$\square$

**Lemma F.5.** *For $t$ such that $2^{2(k_t-1)} \geq 80L$, we have*

$$\max_{a \in \mathcal{A}_t} f^\star(x_t, a) - f^\star(x_t, a_t) \leq 200 \cdot 2^{-2k_t} \left( \sqrt{\Lambda L} + L \right).$$

**Proof.**

$$\max_{a \in \mathcal{A}_t} f^\star(x_t, a) - f^\star(x_t, a_t)$$

$$\leq \max_{f \in \mathcal{F}_t} f(x_t, a_t^{\text{ucb}}) - f^\star(x_t, a_t) \qquad (\text{define } a_t^{\text{ucb}} = \arg\max_{a \in \mathcal{A}_t} \max_{f \in \mathcal{F}_t} f(x_t, a))$$

$$= \max_{f \in \mathcal{F}_t} f(x_t, a_t^{\text{ucb}}) - \min_{f \in \mathcal{F}_t} f(x_t, a_t^{\text{ucb}}) + \min_{f \in \mathcal{F}_t} f(x_t, a_t^{\text{ucb}}) - \max_{f \in \mathcal{F}_t} f(x_t, a_t) + \max_{f \in \mathcal{F}_t} f(x_t, a_t) - f^\star(x_t, a_t)$$

$$\leq 2 \max_{f,f' \in \mathcal{F}_t} \max_{a \in \mathcal{A}_t} |f(x_t, a) - f'(x_t, a)| + \min_{f \in \mathcal{F}_t} f(x_t, a_t^{\text{ucb}}) - \max_{f \in \mathcal{F}_t} f(x_t, a_t)$$

$$\overset{(i)}{\leq} 2 \max_{f,f' \in \mathcal{F}_t} \max_{a \in \mathcal{A}_t} |f(x_t, a) - f'(x_t, a)|$$

$$\overset{(ii)}{\leq} 2 \max_{f,f' \in \mathcal{F}_t} \max_{a \in \mathcal{A}_t} \frac{|f(x_t, a) - f'(x_t, a)|}{\sqrt{2^{-2k_t+2} L^2 + \sum_{\tau \in \Psi_{t,k_t-1}} w_\tau^2 (f(x_\tau, a_\tau) - f'(x_\tau, a_\tau))^2}} \times \sqrt{5} \beta_{t,k_t-1}$$

$$= 2\sqrt{5} \cdot \max_{a \in \mathcal{A}_t} g_{t,k_t-1}(a) \beta_{t,k_t-1}$$

$$\leq 2\sqrt{5} \cdot 2^{-k_t+1} \beta_{t,k_t-1} \qquad \text{(by the definition of } k_t)$$

$$\leq 2\sqrt{5} \cdot 2^{-k_t+1} \cdot 10 \cdot 2^{-k_t+1} \sqrt{\sum_{\tau \in \Psi_{t,k_t-1}} w_\tau^2 (r_\tau - \hat{f}_{t,k_t-1}(x_\tau, a_\tau))^2 L + L^2}$$

$$\text{(by the definition of } \beta_{t,k})$$

$$\leq 200 \cdot 2^{-2k_t} \sqrt{\sum_{\tau \in \Psi_{t,k_t-1}} w_\tau^2 \sigma_\tau^2 L + L^2} \qquad \text{(by Lemma F.2)}$$

$$\leq 200 \cdot \left(2^{-2k_t}\sqrt{\Lambda L} + 2^{-2k_t}L\right). \qquad\qquad (w_\tau \leq 1)$$

$$(19)$$

Here, in $(i)$ we use the definition of $\mathcal{A}_t$ that there exists $f' \in \mathcal{F}_t$ such that $a_t = \max_{a \in \mathcal{A}} f'(x_t, a)$, which implies

$$\min_{f \in \mathcal{F}_t} f(x_t, a_t^{\mathsf{ucb}}) \leq f'(x_t, a_t^{\mathsf{ucb}}) \leq f'(x_t, a_t) \leq \max_{f \in \mathcal{F}_t} f(x_t, a_t).$$

In $(ii)$ we use the fact that for $f, f' \in \mathcal{F}_t$, $k \leq K$ and $2^{2k} \geq 80L$, we have

$$2^{-2k}L^2 + \sum_{\tau \in \Psi_{t,k}} w_\tau^2 (f(x_\tau, a_\tau) - f'(x_\tau, a_\tau))^2$$

$$\leq \beta_{t,k}^2 + 2\sum_{\tau \in \Psi_{t,k}} w_\tau^2 (f(x_\tau, a_\tau) - \hat{f}_{t,k}(x_\tau, a_\tau))^2 + 2\sum_{\tau \in \Psi_{t,k}} w_\tau^2 (f'(x_\tau, a_\tau) - \hat{f}_{t,k}(x_\tau, a_\tau))^2$$

$$\text{(by the definition of } \beta_{t,k})$$

$$\leq 5\beta_{t,k}^2. \qquad\qquad\qquad\qquad\qquad\qquad\qquad \text{(by the fact that } f, f' \in \mathcal{F}_t)$$

$\square$

**Proof of Theorem 5.2.** Without loss of generality, assume $2^{2K} \geq 80L$, which is equivalent to $T^2 \geq \Theta(\log(|\mathcal{F}|T/\delta))$. Notice that $[T] = \bigcup_{k=1}^{K+1} \Psi_{T+1,k}$.

**Bound the regret in $\Psi_{T+1,K+1}$.** Notice that by assumption, for $t \in \Psi_{T+1,K+1}$, we have $2^{2(k_t-1)} = 2^{2K} \geq 80L$. Thus, by Lemma F.5,

$$\sum_{t \in \Psi_{T+1,K+1}} \left(\max_{a \in \mathcal{A}_t} f^\star(x_t, a) - f^\star(x_t, a_t)\right) \leq 200|\Psi_{T+1,K+1}| \cdot 2^{-2(K+1)}\left(\sqrt{\Lambda L} + L\right)$$

$$\leq \mathcal{O}\left(T \times \frac{1}{T^2} \times (\Lambda L + L)\right) = \mathcal{O}(1).$$

**Bound the regret in $\Psi_{T+1,k}$ with $2^{2k} \geq 80L$.** By Lemma F.5,

$$\sum_{t \in \Psi_{T+1,k}} \left(\max_{a \in \mathcal{A}_t} f^\star(x_t, a) - f^\star(x_t, a_t)\right) \leq 200|\Psi_{T+1,k}| \cdot 2^{-2k}\left(\sqrt{\Lambda L} + L\right)$$

$$\leq \tilde{\mathcal{O}}\left(d_{\mathsf{elu}}\left(\sqrt{\Lambda L} + L\right)\right) \qquad \text{(by Lemma F.4)}$$

$$= \tilde{\mathcal{O}}\left(d_{\mathsf{elu}}\sqrt{\Lambda \log |\mathcal{F}|} + d_{\mathsf{elu}} \log |\mathcal{F}|\right).$$

**Bound the regret in $\Psi_{T+1,k}$ with $2^{2k} < 80L$.**

$$\sum_{t \in \Psi_{T+1,k}} \left(\max_{a \in \mathcal{A}_t} f^\star(x_t, a) - f^\star(x_t, a_t)\right) \leq |\Psi_{T+1,k}|$$

$$\leq \tilde{\mathcal{O}}\left(2^{2k} d_{\mathsf{elu}}\right) \qquad \text{(by Lemma F.4)}$$

$$\leq \tilde{\mathcal{O}}(d_{\mathsf{elu}}L)$$

$$= \tilde{\mathcal{O}}\left(d_{\mathsf{elu}} \log |\mathcal{F}|\right).$$

Combining all parts proves the desired bound.

$\square$

## F.2 Lower Bound

**Lemma F.6.** *For any integer $A, T \geq 2$, $N \leq c\sqrt{T/A}$ and positive number $\Lambda > AN/c$ with some $c \leq 1$, there exists a context space $\mathcal{X}$, a contextual bandit problem $\mathcal{F} \subset (\mathcal{X} \times \mathcal{A} \to \mathbb{R})$ with eluder dimension $d_{elu}(\mathcal{F}, 0) = N(A-1)$ and action set $\mathcal{A} = [A]$, and adversarially assigned variances $\sigma_1^2, \ldots, \sigma_T^2$ that $\sum_{t=1}^T \sigma_t^2 \leq \Lambda$ such that any algorithm will suffer at least $\Omega(\min\{\sqrt{NA\Lambda}, \sqrt{AT}\})$.*

**Proof of Lemma F.6.** Let $\varepsilon^2 \in [2N/T, 1]$ be a parameter to be decided later. Consider the function class $\mathcal{F} = \{f^{(0)}\} \cup \{f^{(i,j)}\}_{i \in [N], j \in [A-1]}$ with the space of contexts $\mathcal{X} = \{x^{(1)}, \ldots, x^{(N)}\}$ and the set of actions $\mathcal{A} = [A]$. For any $i \in [N], j \in [A-1]$, the function $f^{(i,j)}$ is defined as the following: For $i \in [N]$ and $j \in [A-1]$,

$$f^{(i,j)}(x^{(i)}, j) = \frac{1}{2} + \varepsilon,$$

$$f^{(i,j)}(x, k) = \frac{1}{2} - \varepsilon, \quad \forall x \neq x^{(i)} \text{ or } \forall k \in [A-1] \setminus \{j\}.$$

$$f^{(i,j)}(x, A) = \frac{1}{2}, \quad \forall x.$$

Meanwhile

$$f^{(0)}(x, j) = \frac{1}{2} - \varepsilon, \quad \forall x \text{ and } \forall j \in [A-1]$$

$$f^{(0)}(x, A) = \frac{1}{2}, \quad \forall x.$$

The eluder dimension of this function class is $N(A-1)$ since $f^{(i,j)}$ is uniquely identified by its value on $(x^{(i)}, j)$.

We assume that $x_t$ is uniformly randomly chosen from $\mathcal{X}$, and $r_t = f_\star(x_t, a_t) + \epsilon\sigma_t$, where $\epsilon \sim \mathcal{N}(0, 1)$ and $\sigma_t$ is defined in the following way. Fix the algorithm. Let $N_t(i,j) = \sum_{s=1}^t \mathbb{1}(x_s = x^{(i)}, a_s = j)$ for any $x \in \mathcal{X}$ and the action $b$. The adversary assigns

$$\sigma_t = \mathbb{1}(a_t = j, N_t(x_t, j) \leq 1/\varepsilon^2),$$

that is, it assigns variance $1$ when the algorithm chooses action $b$ and the context-action pair $x_t, b$ are not played for more than $1/\varepsilon^2$ times.

Fix any algorithm. Denote by $\mathbb{P}_0$ the probability distribution when $f^\star = f^{(0)}$ and $\mathbb{E}_0$ the expectation under $\mathbb{P}_0$. For any $i \in [N], j \in [A-1]$, denote by $\mathbb{P}_{(i,j)}$ the probability distribution when $f^\star = f^{(i,j)}$ and $\mathbb{E}_{(i,j)}$ the expectation under $\mathbb{P}_{(i,j)}$. Then the adversary decides $f_\star$ based on the following rule: if there exists $i \in [N], j \in [A-1]$ such that

$$\mathbb{E}_0[N_T(i,j)] \leq \frac{1}{1000\varepsilon^2} \tag{20}$$

then let $f_\star = f^{(i,j)}$. If no $i, j$ satisfies this, let $f_\star = f^{(0)}$.

If $f^\star = f^{(0)}$, then we have

$$\mathbb{E}_0[R_T] \geq \varepsilon \cdot \mathbb{E}_0\left[\sum_{i \in [N], j \in [A-1]} N_T(i,j)\right] \geq \varepsilon N(A-1) \cdot \min_{i,j} \mathbb{E}_0[N_T(i,j)] \geq \frac{N(A-1)}{1000\varepsilon}.$$

On the other hand, if $f_\star = f^{(i,j)}$, then we have

$$\mathbb{P}_0\left(N_T(i,j) < \frac{1}{\varepsilon^2}\right) \geq \frac{999}{1000}.$$

Then by Lemma B.4, we have

$$\mathbb{P}_0\left(N_T(i,j) < \frac{1}{\varepsilon^2}\right) \leq 3\mathbb{P}_{i,j}\left(N_T(i,j) < \frac{1}{\varepsilon^2}\right) + 4D_{\mathrm{H}}^2(\mathbb{P}_0, \mathbb{P}_{i,j}).$$

Then by Lemma D.2 of Foster et al. (2024), we have

$$D_{\mathrm{H}}^2(\mathbb{P}_0, \mathbb{P}_{i,j}) \leq 7\mathbb{E}_0\big[N_T(i,j) \wedge (1/\varepsilon^2)\big] \cdot 4\varepsilon^2 \leq \frac{7}{250}.$$

Altogether, we can obtain

$$\mathbb{P}_{i,j}\Big(N_T(i,j) < \frac{1}{\varepsilon^2}\Big) \geq \frac{1}{3}\Big(\mathbb{P}_0\Big(N_T(i,j) < \frac{1}{\varepsilon^2}\Big) - 28/250\Big) \geq 1/6.$$

This in turn, implies that with the choice $\varepsilon^2 \geq 2N/T$, that

$$\mathbb{E}_{i,j}[R_T] \geq \varepsilon \cdot \mathbb{E}_{i,j}\left[\sum_{t=1}^T \mathbb{1}[x_t = x^{(i)}] - N_T(x^{(i)}, j)\right] \geq \varepsilon \cdot \Big(\frac{T}{N} - \frac{1}{\varepsilon^2}\Big)\mathbb{P}_{i,j}\Big(N_T(i,j) < \frac{1}{\varepsilon^2}\Big) \geq \frac{T\varepsilon}{12N}.$$

Thus if $\sqrt{AT} > \sqrt{NA\Lambda}$, then we set $\varepsilon = \sqrt{NA/\Lambda}$. We have $\varepsilon \leq 1$ due to the assumption that $\Lambda \geq AN/c$. Then we verify

$$\sum_{t=1}^T \sigma_t^2 = \sum_{t=1}^T \mathbb{1}(x_t = x^{(i)}, a_t = j, N_t(i,j) \leq 1/\varepsilon^2) \leq N(A-1)/\varepsilon^2 \leq \Lambda.$$

Furthermore, $\sqrt{AT} > \sqrt{NA\Lambda}$ implies $T\varepsilon/N > AN/\varepsilon$, and thus

$$\sup_{i \in \{0\} \cup [N]} \mathbb{E}_i[R_T] \geq \Omega\Big(\min\Big\{\frac{AN}{\varepsilon}, \frac{T\varepsilon}{N}\Big\}\Big) \geq \frac{AN}{\varepsilon} = \Omega(\sqrt{NA\Lambda}).$$

Otherwise if $\sqrt{AT} < \sqrt{NA\Lambda}$, then we have $T/N \leq \Lambda$ and we set $\varepsilon = N\sqrt{A/T}$. We have $\varepsilon \leq 1$ due to the assumption that $N \leq c\sqrt{T/A}$. Then we verify

$$\sum_{t=1}^T \sigma_t^2 = \sum_{t=1}^T \mathbb{1}(x_t = x^{(i)}, a_t = j, N_t(i,j) \leq 1/\varepsilon^2) \leq N(A-1)/\varepsilon^2 \leq T/N \leq \Lambda.$$

Finally, we have the lower bounds

$$\sup_{i \in \{0\} \cup [N]} \mathbb{E}_i[R_T] \geq \Omega\Big(\min\Big\{\frac{AN}{\varepsilon}, \frac{T\varepsilon}{N}\Big\}\Big) \geq \Omega(\sqrt{AT}).$$

Overall, we have proven that

$$\sup_{i \in \{0\} \cup [N]} \mathbb{E}_i[R_T] \geq \Omega\Big(\min\Big\{\frac{AN}{\varepsilon}, \frac{T\varepsilon}{N}\Big\}\Big) \geq \Omega(\min\{\sqrt{NA\Lambda}, \sqrt{AT}\}).$$

$\square$

**Proof of Theorem 5.1.**

We first deal with some corner cases:

Case 1: If $A > T$ and $d > T$, then by Lemma C.2 with $N = 1$, we have a lower bound of $\Omega(T)$.

Case 2: If $A > T$, $d < T$, and $d > \Lambda$, then by Lemma C.2 with $N = 1$ and the number of actions in Lemma C.2 set to $d$, we have a lower bound of $\Omega(d)$.

Case 3: If $A > T$, $d < T$, and $d < \Lambda$, then by Lemma C.1 with the number of actions in Lemma C.1 set to $d$, we have a lower bound of $\Omega(\sqrt{d\Lambda})$.

Case 4: If $A < T$, $d > T$, then by Lemma C.2 with $N = \sqrt{T/A}$, we have a lower bound of $\Omega(\sqrt{AT})$.

Case 5: If $A < T$, $d < T$, $d \leq A$, and $d \geq \Lambda$ then by Lemma C.2 with $N = 1$ and the number of actions in Lemma C.2 set to be $d$, we have a lower bound of $\Omega(d)$.

Case 6: If $A < T$, $d < T$, $d \leq A$, and $d > \Lambda$, then by Lemma C.1 with the number of actions in Lemma C.1 set to be $d$, we have a lower bound of $\Omega(\sqrt{d\Lambda})$.

Now, we consider our main cases where $A, d \leq T$ and $d \geq A$. We consider the following two subcases:

Subcase 1: If $d \geq \Omega(\sqrt{d\Lambda})$ or $d \geq \Omega(\sqrt{AT})$, then we invoke Lemma C.2 with $N = \min\{d/A, \sqrt{T/A}\}$ and obtain a lower bound of $\Omega(\min\{d, \sqrt{AT}\}) \geq \Omega(\min\{\sqrt{d\Lambda} + d, \sqrt{AT}\})$.

Subcase 2: If $d < c\sqrt{d\Lambda}$ and $d < c\sqrt{AT}$ for a small enough constant $0 < c < 1$. Then we invoke Lemma F.6 with $N = d/A$ and obtain a lower bound of $\Omega(\min\{\sqrt{d\Lambda} + d, \sqrt{AT}\})$.

Thus, we conclude our proof.

$\square$

# G   Omitted Proofs in Section 6

We revise the proof of Lemma 3 of Foster and Rakhlin (2020) to show the following guarantee.

**Lemma G.1.** *We have for any $t \in [T]$,*

$$
\inf_{p \in \Delta(\mathcal{A})} \max_{M \in \mathcal{M}_t} \mathbb{E}_{a \sim p}\left[\max_{a'} f_M(x_t, a') - f_M(x_t, a) - \gamma \frac{(f_M(x_t, a) - f_{M_t}(x_t, a))^2}{\sigma_{M_t}^2(x_t, a)}\right] \lesssim \frac{A\sigma_{M_t}^2(x_t)}{\gamma},
$$

*where $\sigma_{M_t}^2(x_t) = \sup_{a \in A} \sigma_{M_t}^2(x_t, a)$.*

**Proof of Lemma G.1.** Fix $t \in [T]$. Let $q \in \Delta(\mathcal{A})$ be the policy such that

$$
q(x_t, a) = \frac{1}{\lambda + \gamma/\sigma_{M_t}^2(x_t, a) \cdot (\max_{a' \in \mathcal{A}} f_{M_t}(x_t, a') - f_{M_t}(x_t, a))},
$$

where $\lambda$ is such that $\sum_{a \in \mathcal{A}} q(x_t, a) = 1$. We show that such $\lambda$ exists and $\lambda \in (0, A]$. Let $h(\lambda) = \sum_{a \in \mathcal{A}} \frac{1}{\lambda + \gamma/\sigma_{M_t}^2(x_t, a) \cdot (\max_{a' \in \mathcal{A}} f_{M_t}(x_t, a') - f_{M_t}(x_t, a))}$. Then we have $h(\lambda)$ is monotonically decreasing with $h(0) = \infty$ and $h(A) < 1$. Thus there exists $\lambda \in (0, A]$ such that $h(\lambda) = 1$ that corresponds to $q(x_t, a)$.

We first separate the regret with respect to any fixed $M$ into four parts as the following

$$
\mathbb{E}_{a \sim q}\left[\max_{a'} f_M(x_t, a') - f_M(x_t, a)]\right]
$$
$$
= \mathbb{E}_{a \sim q}\left[\max_{a'} f_{M_t}(x_t, a') - f_{M_t}(x_t, a)]\right] + \mathbb{E}_{a \sim q}[f_{M^\star}(x_t, a) - f_{M_t}(x_t, a)]]
$$
$$
+ (f_M(x_t, a^\star) - f_{M_t}(x_t, a^\star)) + \left(f_{M_t}(x_t, a^\star) - \max_{a'} f_{M_t}(x_t, a')\right), \quad (21)
$$

where $a^\star \in \arg\max_{a \in \mathcal{A}} f_{M^\star}(x_t, a)$. Firstly, by the definition of $q$, we have

$$
\mathbb{E}_{a \sim q}\left[\max_{a'} f_{M_t}(x_t, a') - f_{M_t}(x_t, a)]\right] = \sum_{a \in \mathcal{A}} \frac{\max_{a' \in \mathcal{A}} f_{M_t}(x_t, a') - f_{M_t}(x_t, a)}{\lambda + \gamma/\sigma_{M_t}^2(x_t, a) \cdot (\max_{a' \in \mathcal{A}} f_{M_t}(x_t, a') - f_{M_t}(x_t, a))}
$$
$$
\leq \sum_{a \in \mathcal{A}} \frac{\sigma_{M_t}^2(x_t, a)}{\gamma} \leq \frac{A\sigma_{M_t}^2(x_t)}{\gamma}. \quad (22)
$$

Secondly, by the AM-GM inequality, we have

$$
\mathbb{E}_{a \sim q}\left[f_{M^\star}(x_t, a) - f_{M_t}(x_t, a) - \frac{\gamma}{2} \frac{(f_M(x_t, a) - f_{M_t}(x_t, a))^2}{\sigma_{M_t}^2(x_t, a)}\right]
$$
$$
\leq \mathbb{E}_{a \sim q}\left[\frac{\sigma_{M_t}^2(x_t, a)}{\gamma}\right] \leq \frac{\sigma_{M_t}^2(x_t)}{\gamma}. \quad (23)
$$

Thirdly, again by the AM-GM inequality and the definition of $q$, we have

$$f_M(x_t, a^\star) - f_{M_t}(x_t, a^\star) - q(x_t, a^\star)\frac{\gamma}{2}\frac{(f_M(x_t, a^\star) - f_{M_t}(x_t, a^\star))^2}{\sigma^2_{M_t}(x_t, a^\star)}$$

$$\leq \frac{\sigma^2_{M_t}(x_t, a^\star)}{\gamma q(x_t, a^\star)}$$

$$= \frac{(\lambda + \gamma/\sigma^2_{M_t}(x_t, a^\star) \cdot (\max_{a' \in \mathcal{A}} f_{M_t}(x_t, a') - f_{M_t}(x_t, a))) \cdot \sigma^2_{M_t}(x_t, a^\star)}{\gamma}$$

$$\leq \frac{A\sigma^2_{M_t}(x_t)}{\gamma} + \max_{a' \in \mathcal{A}} f_{M_t}(x_t, a') - f_{M_t}(x_t, a). \tag{24}$$

Plug the inequality (22), (23), and (24) in the equality (21) to obtain the desired bound of

$$\mathbb{E}_{a \sim q}\left[\max_{a'} f_M(x_t, a') - f_M(x_t, a) - \gamma\frac{(f_M(x_t, a) - f_{M_t}(x_t, a))^2}{\sigma^2_{M_t}(x_t, a)}\right] \lesssim \frac{A\sigma^2_{M_t}(x_t)}{\gamma}$$

$\square$

**Lemma G.2.** *Whenever $I_t = 2$ and $M^\star \in \mathcal{M}_t$, we have for any action $a \in [A]$,*

$$\sigma^2_{M^\star}(x_t, a) \lesssim \sigma^2_{M_t}(x_t, a) \lesssim \sigma^2_{M^\star}(x_t, a).$$

**Proof.** Since for any $M, M' \in \mathcal{M}_t$ and $a \in \mathcal{A}$, one has $D^2_H(M(x_t, a), M'(x_t, a)) \leq 1/2$. Thus by Lemma B.2, we have that $\sigma_M(x_t, a) \lesssim \sigma_{M'}(x_t, a)$.

Then since $M^\star$ is in $\mathcal{M}_t$ and $M_t$ is a mixture of models in $\mathcal{M}_t$, we have by the total variacne law

$$\sigma^2_{M^\star}(x_t, a) \lesssim \min_{M \in \mathcal{M}_t} \sigma^2_M(x_t, a) \lesssim \sigma^2_{M_t}(x_t, a) \lesssim \max_{M \in \mathcal{M}_t} \sigma^2_M(x_t, a) \lesssim \sigma^2_{M^\star}(x_t, a).$$

$\square$

**Lemma G.3.** *Whenever $I_t = 2$ and $M^\star \in \mathcal{M}_t$, we have for any action $a \in \mathcal{A}$*

$$|f_{M^\star}(x_t, a) - f_{M_t}(x_t, a)| \lesssim \sqrt{\sigma^2_{M_t}(x_t, a)^2 D^2_H(M(x_t, a), M_t(x_t, a))}.$$

**Proof.** Since $M^\star$ is in $\mathcal{M}_t$ and $M_t$ is a mixture of models in $\mathcal{M}_t$, we have for any action $a \in [A]$,

$$D^2_H(M^\star(x_t, a), M_t(x_t, a)) \leq \max_{M \in \mathcal{M}_t} D^2_H(M^\star(x_t, a), M(x_t, a)) \leq 1/2.$$

Then by Lemma B.3, we have that

$$|f_{M^\star}(x_t, a) - f_{M_t}(x_t, a)| \lesssim \sqrt{(\sigma^2_{M^\star}(x_t, a) + \sigma^2_{M_t}(x_t, a))D^2_H(M(x_t, a), M_t(x_t, a))}.$$

Then by Lemma G.2, we have

$$|f_{M^\star}(x_t, a) - f_{M_t}(x_t, a)| \lesssim \sqrt{(\sigma^2_{M^\star}(x_t, a) + \sigma^2_{M_t}(x_t, a))D^2_H(M(x_t, a), M_t(x_t, a))}$$

$$\lesssim \sqrt{\sigma^2_{M_t}(x_t, a)^2 D^2_H(M(x_t, a), M_t(x_t, a))}.$$

$\square$

**Lemma G.4.** *The regrets accumalated on rounds where $I_t = 2$ are bounded with probability at least $1 - \delta$ by*

$$\sum_{t=1}^T \mathrm{reg}_t \mathbb{1}(I_t = 2) \lesssim \sqrt{A\sum_{t=1}^T \sigma^2_{M^\star}(x_t) \cdot \log(|\mathcal{M}|/\delta)},$$

*where $\sigma^2_{M^\star}(x_t) = \sup_{a \in A} \sigma^2_{M^\star}(x_t, a)$ and $\mathrm{reg}_t = \mathbb{E}_{a \sim p_t}[\max_{a'} f_{M^\star}(x_t, a') - f_{M^\star}(x_t, a)]$.*

**Proof.** Let $\mathbb{E}_t[\cdot] := \mathbb{E}[\cdot \mid \mathcal{H}_t]$ where $\mathcal{H}_t$ is the history up to time $t$. By Lemma G.1 and Lemma G.3, whenever $I_t = 2$ and $M^\star \in \mathcal{M}_t$, we have

$$\mathbb{E}_t[\mathrm{reg}_t \mathbb{1}(I_t = 2, M^\star \in \mathcal{M}_t)] = \mathbb{E}_{a \sim p_t}\left[\left(\max_{a'} f_{M^\star}(x_t, a') - f_{M^\star}(x_t, a)\right) \cdot \mathbb{1}(M^\star \in \mathcal{M}_t)\right]$$

$$\leq \gamma \mathbb{E}_{a \sim p_t}\left[\frac{(f_{M^\star}(x_t, a) - f_{M_t}(x_t, a))^2}{\sigma^2_{M_t}(x_t, a)} \cdot \mathbb{1}(M^\star \in \mathcal{M}_t)\right] + \frac{A\sigma^2_{M_t}(x_t)}{\gamma}$$

$$\lesssim \gamma \mathbb{E}_{a \sim p_t}\left(D^2_{\mathrm{H}}(M^\star(x_t, a), M_t(x_t, a)) \cdot \mathbb{1}(M^\star \in \mathcal{M}_t)\right) + \frac{A\sigma^2_{M_t}(x_t)}{\gamma}.$$

Then by summation over $t \in [T]$, we have

$$\sum_{t=1}^{T} \mathbb{E}_t[\mathrm{reg}_t \mathbb{1}(I_t = 2, M^\star \in \mathcal{M}_t)]$$

$$\lesssim \frac{A \sum_{t=1}^{T} \sigma^2_{M_t}(x_t)}{\gamma} + \gamma \cdot \sum_{t=1}^{T} \mathbb{E}_{a \sim p_t}\left[D^2_{\mathrm{H}}(M^\star(x_t, a), M_t(x_t, a)) \cdot \mathbb{1}(M^\star \in \mathcal{M}_t)\right].$$

Then by Lemma A.15 of Foster et al. (2021), we have that with probability at least $1 - \delta/2$,

$$\sum_{t=1}^{T} \mathbb{E}_{a \sim p_t}\left[D^2_{\mathrm{H}}(M^\star(x_t, a), M_t(x_t, a)) \cdot \mathbb{1}(M^\star \in \mathcal{M}_t)\right] \leq \log(2|\mathcal{M}|/\delta).$$

Then by the choice of $\gamma = \sqrt{A \sum_{t=1}^{T} \sigma^2_{M_t}(x_t) / \log(2|\mathcal{M}|/\delta)}$, we have with probability at least $1 - \delta/2$,

$$\frac{A \sum_{t=1}^{T} \sigma^2_{M_t}(x_t)}{\gamma} + \gamma \cdot \sum_{t=1}^{T} \mathbb{E}_{a \sim p_t}\left[D^2_{\mathrm{H}}(M^\star(x_t, a), M_t(x_t, a)) \cdot \mathbb{1}(M^\star \in \mathcal{M}_t)\right]$$

$$\lesssim \sqrt{A \sum_{t=1}^{T} \sigma^2_{M_t}(x_t) \cdot \log(2|\mathcal{M}|/\delta)}$$

$$\lesssim \sqrt{A \sum_{t=1}^{T} \sigma^2_{M^\star}(x_t) \cdot \log(2|\mathcal{M}|/\delta)},$$

where the second inequality is by Lemma G.2. We also have by Lemma A.3 of Foster et al. (2021) that with probability at least $1 - \delta/4$, for all $t \in [T]$,

$$\sum_{s=1}^{t} D^2_{\mathrm{H}}(M^\star(x_s, a_s), M_t(x_s, a_s)) \leq \frac{3}{2} \sum_{s=1}^{t} \mathbb{E}_{a \sim p_s}\left[D^2_{\mathrm{H}}(M^\star(x_s, a), M_t(x_s, a))\right] + 4\log(8T/\delta).$$

By union bound, this implies with probability at least $1 - 3\delta/4$, for all $t \in [T]$, $M^\star \in \mathcal{M}_t$. Again by Lemma A.3 of Foster et al. (2021), we have with probability at least $1 - \delta/4$, for all $t \in [T]$

$$\sum_{t=1}^{T} \mathrm{reg}_t \mathbb{1}(I_t = 2, M^\star \in \mathcal{M}_t) \leq \frac{3}{2} \sum_{t=1}^{T} \mathbb{E}_t[\mathrm{reg}_t \mathbb{1}(I_t = 2, M^\star \in \mathcal{M}_t)] + 4\log(8/\delta).$$

Thus by the union bound , with probability at least $1 - \delta$, we have

$$\sum_{t=1}^{T} \mathrm{reg}_t \mathbb{1}(I_t = 2) = \sum_{t=1}^{T} \mathrm{reg}_t \mathbb{1}(I_t = 2, M^\star \in \mathcal{M}_t)$$

$$\lesssim \sqrt{A \sum_{t=1}^{T} \sigma^2_{M^\star}(x_t) \cdot \log(|\mathcal{M}|/\delta)}.$$

$\square$

**Lemma G.5.** *For any $\lambda \geq 0$ and $\alpha > 0$, we have*

$$\sum_{t=1}^{T} \min \left\{ \lambda, \sup_{M,M' \in \mathcal{M}_t} \frac{D_H^2(M(x_t,a_t), M'(x_t,a_t))}{\alpha^2 T + \sum_{\tau=1}^{t-1} D_H^2(M(x_\tau,a_\tau), M'(x_\tau,a_\tau))} \right\} \leq d_{elu}^H(\alpha)(\lambda + \log T).$$

**Proof.** The proof follows similarly from the proof of Lemma D.5 by replacing the square divergence with squared Hellinger distance. □

**Lemma G.6.** *The regrets accumalated on rounds where $I_t = 1$ are bounded by*

$$\sum_{t=1}^{T} \mathrm{reg}_t \mathbb{1}(I_t = 1) \lesssim d_{elu}^H(\alpha)(\alpha^2 T + \log(|\mathcal{M}|T/\delta)) \log T,$$

*for any $\alpha > 0$.*

**Proof.** For any $t \in [T]$ and $M, M' \in \mathcal{M}_t$, we have

$$\sum_{\tau=1}^{t-1} D_H^2(M(x_\tau,a_\tau), M'(x_\tau,a_\tau))$$

$$\leq 2 \left( \sum_{\tau=1}^{t-1} D_H^2(M(x_\tau,a_\tau), M_\tau(x_\tau,a_\tau)) + \sum_{\tau=1}^{t-1} D_H^2(M_\tau(x_\tau,a_\tau), M'(x_\tau,a_\tau)) \right) \leq 4L. \quad (25)$$

Thus let $\sum_{t=1}^{T} \mathbb{1}(I_t = 1) = T_1$ and we have

$$\sum_{t=1}^{T} \mathrm{reg}_t \mathbb{1}(I_t = 1) \leq \sum_{t=1}^{T} \mathbb{1}(I_t = 1) = T_1.$$

Thus we have for any $\alpha \geq 0$,

$$T_1 = \sum_{I_t=1} \mathbb{1} \left\{ \sup_{M,M' \in \mathcal{M}_t} D_H^2(M(x_t,a_t), M'(x_t,a_t)) \geq \frac{1}{2} \right\}$$

$$\leq \sqrt{2} \sum_{I_t=1} \min \left\{ \frac{1}{\sqrt{2}}, \sup_{M,M' \in \mathcal{M}_t} D_H(M(x_t,a_t), M'(x_t,a_t)) \right\}$$

$$\leq \sqrt{2} \sum_{I_t=1} \min \left\{ \frac{1}{\sqrt{2}}, \sup_{M,M' \in \mathcal{M}_t} \frac{D_H(M(x_t,a_t), M'(x_t,a_t))}{\sqrt{\alpha^2 T + \sum_{\tau=1}^{t-1} D_H^2(M(x_\tau,a_\tau), M'(x_\tau,a_\tau))}} \cdot \sqrt{\alpha^2 T + 4L} \right\},$$

where the last inequality is by (25). Furthermore, by Cauchy-Schwarz inequality, we have

$$\sum_{I_t=1} \min \left\{ \frac{1}{\sqrt{2}}, \sup_{M,M' \in \mathcal{M}_t} \frac{D_H(M(x_t,a_t), M'(x_t,a_t))}{\sqrt{\alpha^2 T + \sum_{\tau=1}^{t-1} D_H^2(M(x_\tau,a_\tau), M'(x_\tau,a_\tau))}} \cdot \sqrt{\alpha^2 T + 4L} \right\}$$

$$\leq \sqrt{T_1} \sqrt{\sum_{I_t=1} \min \left\{ \frac{1}{2}, \sup_{M,M' \in \mathcal{M}_t} \frac{D_H^2(M(x_t,a_t), M'(x_t,a_t))}{\alpha^2 T + \sum_{\tau=1}^{t-1} D_H^2(M(x_\tau,a_\tau), M'(x_\tau,a_\tau))} \cdot (\alpha^2 T + 4L) \right\}}$$

$$\leq \sqrt{T_1 \cdot d_{elu}^H(\alpha)(1/2 + (\alpha^2 T + 4L) \log T)},$$

where the last inequality is by Lemma G.5. Altogether, we have

$$T_1 \lesssim \sqrt{T_1 \cdot d_{\text{elu}}^{\mathsf{H}}(\alpha)(1/2 + (\alpha^2 T + 4L) \log T)}.$$

Reorganizing the above inequality, we obtain the desired bound.

□

**Proof of Theorem 6.1.** The proof is straight-forward by combing Lemma G.6 and Lemma G.4, i.e., with probability at least $1 - \delta$,

$$R_T \lesssim \sqrt{A \sum_{t=1}^{T} \sigma_{M^\star}^2(x_t) \cdot \log(|\mathcal{M}|/\delta)} + d_{\text{elu}}^{\mathsf{H}}(\alpha)\big(\alpha^2 T + \log(|\mathcal{M}|T/\delta)\big) \log T.$$

Then, by choosing $\alpha = 1/\sqrt{T}$, we obtain the desired bound.

□

**Proof of Theorem 6.2.** The proof of this theorem is essentially the same as that of Theorem 4.1. Recall that in Theorem 4.1 we construct a hard instance for a function class $\mathcal{F}$ that only contain reward mean information, but the reward has a fixed variance upper bound $\sigma_t^2 \le \sigma^2$. What we do here is simply embed this instance in a *model class* $\mathcal{M}$ where for every model, every context, and every action, the reward distribution is a Gaussian with mean as in $\mathcal{F}$ and variance $\sigma^2$. Notice that we can change the distribution $P_{\sigma,\epsilon}^-$ and $P_{\sigma,\epsilon}^+$ in Lemma B.1 to $\mathcal{N}(\sigma - \epsilon, \sigma^2)$ and $\mathcal{N}(\sigma + \epsilon, \sigma^2)$, respectively, which still gives us $D_{\text{KL}}\big(P_{\sigma,\epsilon}^- \,\|\, P_{\sigma,\epsilon}^+\big) \le \frac{(2\epsilon)^2}{2\sigma^2} = \frac{2\epsilon^2}{\sigma^2}$. This allows us to prove the same bound as in Lemma C.1. Furthermore, the eluder dimension $d_{\text{elu}}^{\mathsf{H}}(\mathcal{M}, 0)$ defined through the Hellinger distance between models remain the same as the eluder dimension $d_{\text{elu}}(\mathcal{F}, 0)$ defined through the mean difference in the constructions of Lemma C.1 and Lemma C.2. Overall, the lower bounds in Lemma C.1 and Lemma C.2 are still applicable after we change the reward distribution from Bernoulli-style distributions to Gaussian distributions, and change the distance measure from mean difference to Hellinger distance. The arguments in Theorem 4.1 thus allow us to prove the same lower bound in the case here.

□

**Lemma G.7.** *For any integer $A, T \ge 2$, $N \le c\sqrt{T/A}$ and positive number $\Lambda > AN/c$, there exists a context space $\mathcal{X}$, a contextual bandit model class $\mathcal{M} \subset (\mathcal{X} \times \mathcal{A} \to \Delta(\mathbb{R}))$ with eluder dimension $d_{elu}^{\mathsf{H}}(\mathcal{M}, 0) = N(A - 1)$ and action set $\mathcal{A} = [A]$, and adversarially assigned variances $\sigma_1^2, \dots, \sigma_T^2$ that $\sum_{t=1}^{T} \sigma_t^2 \le \Lambda$ such that any algorithm will suffer at least $\Omega(\min\{\sqrt{NA\Lambda}, \sqrt{AT}\})$.*

**Proof of Lemma G.7.** This lower bound is based on a modification of Lemma F.6. Concretely, we illustrate here how to embed the hard case from Lemma F.6 to an equivalent model class in the distributional case.

One can add information into the context for the hard case constructed in Lemma F.6. Concretely, we don't enlarge the function class, but for each context $x$, the new context space will have $T^A$ corresponding contexts $(x, j_1, ..., j_A)_{1 \le j_1, ..., j_A \le T}$, where the second argument will be used to record the number of pulls to each action $a$ under context $x$. The function value under these contexts will be the same as under the original context. The adversarial thus can choose in the new context space $(x_t, N_t(x_t, 1), ..., N_t(x_t, A))$ to embed the hard case from Lemma F.6. The Hellinger eluder dimension is twice the eluder dimension because there are two types of variances corresponding to each original context. Thus, we obtain the desired bound.

□

**Proof of Theorem 6.3.**

This proof follows a similar argument with Theorem 5.1, with only embedding the function classes to model classes. Again, we change the Bernoulli-styled distributions in the construction of Theorem 5.1

to Gaussian distributions. The only other more crucial difference lies in the Subclass 2 at the end. We first deal with some corner cases:

Case 1: If $A > T$ and $d > T$, then by Lemma C.2 with $N = 1$, we have a lower bound of $\Omega(T)$.

Case 2: If $A > T$, $d < T$, and $d > \Lambda$, then by Lemma C.2 with $N = 1$ and the number of actions in Lemma C.2 set to $d$, we have a lower bound of $\Omega(d)$.

Case 3: If $A > T$, $d < T$, and $d < \Lambda$, then by Lemma C.1 with the number of actions in Lemma C.1 set to $d$, we have a lower bound of $\Omega(\sqrt{d\Lambda})$.

Case 4: If $A < T$, $d > T$, then by Lemma C.2 with $N = \sqrt{T/A}$, we have a lower bound of $\Omega(\sqrt{AT})$.

Case 5: If $A < T$, $d < T$, $d \leq A$, and $d \geq \Lambda$ then by Lemma C.2 with $N = 1$ and the number of actions in Lemma C.2 set to be d, we have a lower bound of $\Omega(d)$.

Case 6: If $A < T$, $d < T$, $d \leq A$, and $d > \Lambda$, then by Lemma C.1 with the number of actions in Lemma C.1 set to be d, we have a lower bound of $\Omega(\sqrt{d\Lambda})$.

Now, we consider our main cases where $A, d \leq T$ and $d \geq A$. We consider the following two subcases:

Subcase 1: If $d \geq \Omega(\sqrt{d\Lambda})$ or $d \geq \Omega(\sqrt{AT})$, then we invoke Lemma C.2 with $N = \min\{d/A, \sqrt{T/A}\}$ and obtain a lower bound of $\Omega(\min\{d, \sqrt{AT}\}) \geq \Omega(\min\{\sqrt{d\Lambda} + d, \sqrt{AT}\})$.

Subcase 2: If $d < c\sqrt{d\Lambda}$ and $d < c\sqrt{AT}$ for a small enough constant $c > 0$. Then we invoke Lemma G.7 and obtain a lower bound of $\Omega(\min\{\sqrt{d\Lambda} + d, \sqrt{AT}\})$.

Thus, we conclude our proof.

$\square$

# H    Upper Bound with Zero-One Variance (Section 7)

In this section, we consider the setting where $\sigma_t = 0$ or 1, and not reveal to the learner at the beginning of each round. The algorithm is displayed in Algorithm 6. We have the following theorem.

**Theorem H.1.** *With the choice of $\gamma = \sqrt{\frac{8A}{\log|\mathcal{F}|}}$, we have the upper bound on the expected regret of Algorithm 6*

$$\mathbb{E}[R_T] = \mathcal{O}\left(\sqrt{A\log|\mathcal{F}|\left(1 + \sum_{t=1}^{T}\sigma_t^2\right)} + A(d_{elu} + \log|\mathcal{F}|)\right).$$

We will prove upper bounds for regret of rounds with $\sigma_t = 0$ and $\sigma_t = 1$ separately. First we bound the regret of rounds with $\sigma_t = 1$.

**Lemma H.1.** *With $\gamma = \sqrt{\frac{8A}{\log|\mathcal{F}|}}$, the output actions $a_t$ at each round in Algorithm 6 satisfies*

$$\sum_{t=1}^{T}\mathbb{I}[\sigma_t = 1]\left(\max_{a\in\mathcal{A}}f^\star(x_t, a) - f(x_t, a_t)\right) \leq 4\sqrt{2A\log|\mathcal{F}|\cdot\left(1 + \sum_{t=1}^{T}\sigma_t^2\right)}.$$

**Proof.** Based on the inverse-gap weighting update rule Eq. (26), similar to (Foster and Rakhlin, 2020, Lemma 3), we have

$$\mathbb{E}\left[\sum_{t=1}^{T}\mathbb{I}[\sigma_t = 1](\max_a f^\star(x_t, a) - f^\star(x_t, a_t))\right]$$

**Algorithm 6** Variance Sensitive SquareCB for Zero-One Noise

**Input:** $\gamma$.
1: Let $\mathcal{F}_1 \leftarrow \mathcal{F}$.
2: **for** $t = 1 : T$ **do**
3:   Receive $x_t$, and calculate $\mathcal{A}_t = \{a \in \mathcal{A} : \exists f \in \mathcal{F}_t, f(x_t, a) = \arg\max_{a \in \mathcal{A}} f(x_t, a)\}$.
4:   Sample action $a_t \sim p_t$ and receive $\sigma_t$ and $r_t$, where

$$p_t(a) = \begin{cases} \frac{1}{A + \gamma\sqrt{1 + \sum_{s<t} \sigma_s^2}(\max_{a' \in \mathcal{A}_t} f_t(x_t, a') - f_t(x_t, a))} & \text{for } a \in \mathcal{A}_t, \\ 0 & \text{for } a \notin \mathcal{A}_t. \end{cases} \tag{26}$$

5:   Calculate $L_t(f)$ for all $f \in \mathcal{F}$ as

$$L_t(f) = \sum_{\tau=1}^{t} \frac{(f(x_\tau, a_\tau) - r_\tau)^2}{\sigma_\tau^2},$$

   where for those $\sigma_\tau = 0$, we define $\frac{(f(x_\tau, a_\tau) - r_\tau)^2}{\sigma_\tau^2} = 0$ if $f(x_\tau, a_\tau) = r_\tau$ and $\infty$ otherwise.
6:   Calculate $q_t(f)$ to be

$$q_{t+1}(f) = \frac{e^{-L_t(f)}}{\sum_{g \in \mathcal{F}} e^{-L_t(g)}}.$$

7:   Calculate $f_{t+1}$ to be

$$f_{t+1} = \sum_{f \in \mathcal{F}} q_{t+1}(f) \cdot f.$$

8:   **if** $\sigma_0 = 0$ **then**
9:     Update $\mathcal{F}_{t+1} = \{f \in \mathcal{F}_t, f(x_t, a_t) = r_t\}$.
10:   **else**
11:     Let $\mathcal{F}_{t+1} = \mathcal{F}_t$.

$$\leq \mathbb{E}\left[\sum_{t=1}^{T} \mathbb{I}[\sigma_t = 1] \cdot \frac{2A}{A + \gamma\sqrt{1 + \sum_{s=1}^{t-1} \sigma_s^2}}\right] + \frac{\gamma}{4}\mathbb{E}\left[\sum_{t=1}^{T} \sqrt{1 + \sum_{s=1}^{t-1} \sigma_s^2} \cdot (f_t(x_t, a_t) - f^\star(x_t, a_t))^2\right].$$

For the first term, we have analysis

$$\sum_{t=1}^{T} \mathbb{I}[\sigma_t = 1] \cdot \frac{1}{A + \gamma\sqrt{1 + \sum_{s=1}^{t-1} \sigma_s^2}} \leq \frac{2}{\gamma}\sum_{t=1}^{T}\left(\sqrt{1 + \sum_{s=1}^{t} \sigma_s^2} - \sqrt{1 + \sum_{s=1}^{t-1} \sigma_s^2}\right) = \frac{2}{\gamma}\sqrt{1 + \sum_{s=1}^{T} \sigma_s^2},$$

where the first inequality uses the fact that $\sigma_t \in \{0, 1\}$ and when $\sigma_t = 1$ we have $\frac{1}{\sqrt{1 + \sum_{s=1}^{t-1} \sigma_s^2}} =$

$\frac{1}{\sqrt{\sum_{s=1}^{t} \sigma_s^2}} \leq 2\left(\sqrt{1 + \sum_{s=1}^{t} \sigma_s^2} - \sqrt{1 + \sum_{s=1}^{t-1} \sigma_s^2}\right)$. For the second term, we have

$$\sum_{t=1}^{T} \sqrt{1 + \sum_{s=1}^{t-1} \sigma_s^2} \cdot (f_t(x_t, a_t) - f^\star(x_t, a_t))^2$$

$$\leq \sqrt{1 + \sum_{t=1}^{T} \sigma_t^2} \cdot \sum_{t=1}^{T} (f_t(x_t, a_t) - f^\star(x_t, a_t))^2$$

$$\leq \sqrt{1 + \sum_{t=1}^{T} \sigma_t^2} \cdot \log|\mathcal{F}|,$$

where the last line is according to the analysis of Vovk's aggregating algorithm (Vovk, 1995; Cesa-Bianchi and Lugosi, 2006).

Above all, with $\gamma = \sqrt{\frac{8A}{\log|\mathcal{F}|}}$, we get

$$\mathbb{E}\left[\sum_{t=1}^{T}\mathbb{I}[\sigma_t = 1](\max_a f^\star(x_t, a) - f^\star(x_t, a_t))\right]$$

$$\leq \left(\frac{2A}{\gamma} + \frac{\gamma \log|\mathcal{F}|}{4}\right)\sqrt{1 + \sum_{t=1}^{T}\sigma_t^2}$$

$$= 4\sqrt{2A\log|\mathcal{F}| \cdot \left(1 + \sum_{t=1}^{T}\sigma_t^2\right)}.$$

$\square$

The following is a useful lemma regarding Eluder dimension.

**Lemma H.2.** *We define*

$$Z_t = \mathbb{I}\left[\exists f, f' \in \mathcal{F}_t \text{ such that } |f(x_t, a_t) - f'(x_t, a_t)| \geq \frac{1}{T}\right]. \tag{27}$$

*Then we have*

$$\sum_{t=1}^{T}\mathbb{I}[\sigma_t = 0]Z_t \leq d_{elu}(1/T).$$

**Proof.** This lemma follows directly according to the definition of Eluder dimension $d_{\text{elu}}(1/T)$ in Definition 2.1, and the fact that all functions in $\mathcal{F}_t$ must agree on the previous samples where $\sigma_s = 0$. $\square$

With this lemma, we are ready to bound the regret of rounds with $\sigma_t = 0$.

**Lemma H.3.** *With $\gamma = \sqrt{\frac{8A}{\log|\mathcal{F}|}}$, the output actions $a_t$ at each round in Algorithm 6 satisfies*

$$\mathbb{E}\left[\sum_{t=1}^{T}\max_{a\in\mathcal{A}}\mathbb{I}[\sigma_t = 0](f^\star(x_t, a) - f^\star(x_t, a_t))\right] = \mathcal{O}\left(\sqrt{A\log|\mathcal{F}|\sum_{t=1}^{T}\sigma_t^2} + A(d_{elu} + \log|\mathcal{F}|)\right).$$

**Proof.** In this proof, we focus on the case $\sigma_t = 0$. We first notice that for rounds $\sigma_t = 0$, we always have $f^*(x_t, a_t) = r_t$. Hence $f^* \in \mathcal{F}_t$ for every $t \in [T]$. We define

$$a_t^* = \arg\max_{a\in\mathcal{A}} f^*(x_t, a).$$

At round $t$, we let $b_t = \arg\max_{a\in\mathcal{A}} f_t(x_t, a)$, and we define $\mathcal{B}_t \subset \mathcal{A}_t$ as

$$\mathcal{B}_t \triangleq \left\{a \in \mathcal{A}_t : f_t(x_t, b_t) - f_t(x_t, a) \leq \frac{2}{T}\right\}. \tag{28}$$

For rounds with $\sigma_t = 0$, based on $x_t, f_t$, we divide such rounds into two cases, which we denote as $\mathcal{T}_1$ and $\mathcal{T}_2$.

(a) There exists $a \in \mathcal{B}_t$ and $f \in \mathcal{F}_t$, such that $|f(x_t, a) - f_t(x_t, a)| \geq 1/T$.

(b) For all $f \in \mathcal{F}_t$ and $a \in \mathcal{B}_t$, we have $|f(x_t, a) - f_t(x_t, a)| \leq 1/T$.

**Regret in $t \in \mathcal{T}_1$.** For those $t \in \mathcal{T}_1$, first we notice that for any $a \in \mathcal{B}_t$, according to Eq. (26) we have

$$p_t(a) = \frac{1}{A + \gamma\sqrt{1 + \sum_{s<t}\sigma_s^2}(f_t(x_t, b_t) - f_t(x_t, a))} \geq \frac{1}{A + \gamma\sqrt{1 + \sum_{s<t}\sigma_s^2} \cdot 2/T} \geq \frac{1}{7A},$$

where in the last inequality we use the fact that

$$\frac{2\gamma\sqrt{1+\sum_{s<t}\sigma_s^2}}{T} \leq \frac{2\gamma\sqrt{T}}{T} = \frac{2\sqrt{8A/\log|\mathcal{F}|}}{\sqrt{T}} \leq 6A.$$

Therefore, according to the definition of $\mathcal{T}_1$ that there exists some $a \in \mathcal{B}_t$ and $f \in \mathcal{F}_t$ such that $|f(x_t,a) - f_t(x_t,a)| \geq 1/T$, for any $t \in \mathcal{T}_1$ with probability at least $1/7A$ we will sample an action $a_t$ such that $Z_t = 1$ ($Z_t$ is defined in Eq. (27)). Therefore, we have

$$\mathbb{E}\left[\sum_{t\in\mathcal{T}_1}(f^\star(x_t,a_t^\star) - f^\star(x_t,a_t))\right]$$

$$\leq \mathbb{E}\left[\sum_{t=1}^{T}\mathbb{I}[t\in\mathcal{T}_1]\right] \leq \mathbb{E}\left[\sum_{t=1}^{T}\frac{Z_t\mathbb{I}[\sigma_t=0]}{1/(7A)}\right] = 7A\mathbb{E}\left[\sum_{t=1}^{T}Z_t\mathbb{I}[\sigma_t=0]\right] \leq 7Ad_{\mathrm{elu}}(1/T),$$

where the last inequality uses Lemma H.2.

**Regret in $t \in \mathcal{T}_2$ with $a_t \in \mathcal{B}_t$.** For those $t \in \mathcal{T}_2$, to facilitate the analysis we define

$$l_t(f,x_t,a) = \frac{(f(x_t,a)-r_t)^2 - (f^\star(x_t,a)-r_t)^2}{\sigma_t^2} \qquad \forall t \in [T],$$

and

$$\Phi_t = \log\left(\sum_{f\in\mathcal{F}}\exp\left(-\sum_{s=1}^{t}l_s(f,x_s,a_s)\right)\right). \tag{29}$$

Then we have

$$q_t(f) = \frac{\exp(-\sum_{s=1}^{t-1}l_s(f,x_s,a_s))}{\sum_{g\in\mathcal{F}}\exp(-\sum_{s=1}^{t-1}l_s(f,x_s,a_s))},$$

which implies that

$$\mathbb{E}[\Phi_{t-1} - \Phi_t] = -\mathbb{E}\left[\log\frac{\sum_{f\in\mathcal{F}}\exp\left(-\sum_{s=1}^{t}l_s(f,x_s,a_s)\right)}{\sum_{f\in\mathcal{F}}\exp\left(-\sum_{s=1}^{t-1}l_s(f,x_s,a_s)\right)}\right]$$

$$= -\mathbb{E}\left[\log\left(\sum_{f\in\mathcal{F}}q_t(f)\exp\left(-l_t(f,x_t,a_t)\right)\right)\right]$$

$$\geq -p_t(a_t^\star)\log\left(\sum_{f\in\mathcal{F}}q_t(f)\exp\left(-l_t(f,x_t,a_t^\star)\right)\right)$$

$$= -p_t(a_t^\star)\log\left(\sum_{f\in\mathcal{F}}q_t(f)\mathbb{I}[f(x_t,a_t^\star)=f^\star(x_t,a_t^\star)]\right)$$

$$= -p_t(a_t^\star)\log\left(1 - \sum_{f\in\mathcal{F}}q_t(f)\mathbb{I}[f(x_t,a_t^\star)\neq f^\star(x_t,a_t^\star)]\right)$$

$$\geq p_t(a_t^\star)\sum_{f\in\mathcal{F}}q_t(f)\mathbb{I}[f(x_t,a_t^\star)\neq f^\star(x_t,a_t^\star)]. \tag{30}$$

Hence we have

$$\sum_{a\in\mathcal{B}_t}p_t(a)\left(f^\star(x_t,a_t^\star) - f^\star(x_t,a)\right)$$

$$\overset{(i)}{\leq} \sum_{a\in\mathcal{B}_t}p_t(a)\left(f^\star(x_t,a_t^\star) - f_t(x_t,a_t^\star) + f_t(x_t,a) - f^\star(x_t,a)\right) + \frac{2}{T}$$

$$\overset{(ii)}{\leq} \sum_{a \in \mathcal{B}_t} p_t(a) \left( f^\star(x_t, a_t^\star) - f_t(x_t, a_t^\star) \right) + \frac{3}{T}$$

$$\leq \frac{1}{p_t(a_t^\star)} \cdot p_t(a_t^\star) |f_t(x_t, a_t^\star) - f^\star(x_t, a_t^\star)| + \frac{3}{T}$$

$$= \frac{1}{p_t(a_t^\star)} \cdot p_t(a_t^\star) \left| \sum_{f \in \mathcal{F}} q_t(f) f(x_t, a_t^\star) - f^\star(x_t, a_t^\star) \right| + \frac{3}{T}$$

$$= \frac{1}{p_t(a_t^\star)} \cdot p_t(a_t^\star) \sum_{f \in \mathcal{F}} q_t(f) \mathbb{I}[f(x_t, a_t^\star) \neq f^\star(x_t, a_t^\star)] + \frac{3}{T}$$

$$\overset{(iii)}{\leq} \left( A + \gamma \sqrt{1 + \sum_{t=1}^{T} \sigma_t^2} \right) \cdot \mathbb{E}[\Phi_{t-1} - \Phi_t] + \frac{3}{T},$$

where in $(i)$ we use the fact that according to the definition of $\mathcal{B}_t$ in Eq. (28),

$$f_t(x_t, a) - f_t(x_t, a_t^\star) \geq f_t(x_t, b_t) - f_t(x_t, a_t^\star) - \frac{2}{T} \geq -\frac{2}{T},$$

and $(ii)$ uses the definition of $\mathcal{T}_2$ that for any $a \in \mathcal{B}_t$ and $f \in \mathcal{F}_t$ we always have $|f_t(x_t, a) - f^\star(x_t, a)| \leq 1/T$, and $(iii)$ uses Eq. (30) and also the fact that

$$p_t(a_t^\star) = \frac{1}{A + \gamma \sqrt{1 + \sum_{s<t} \sigma_s^2}(f_t(x_t, b_t) - f(x_t, a_t^\star))} \geq \frac{1}{A + \gamma \sqrt{1 + \sum_{t=1}^{T} \sigma_t^2}}.$$

Suming this up for every $t \in \mathcal{T}_2$, we obtain that

$$\mathbb{E}\left[ \sum_{t \in \mathcal{T}_2} \mathbb{I}[a_t \in \mathcal{B}_2](f^\star(x_t, a_t^\star) - f^\star(x_t, a_t)) \right]$$

$$\leq \left( A + \gamma \sqrt{1 + \sum_{t=1}^{T} \sigma_t^2} \right) \cdot \left( 1 + \sum_{t=1}^{T} \mathbb{E}[\Phi_{t-1} - \Phi_t] \right) + \frac{3}{T} \cdot T$$

$$\leq \left( A + \gamma \sqrt{1 + \sum_{t=1}^{T} \sigma_t^2} \right) \log |\mathcal{F}| + 3,$$

where in the last inequality we use the definition of $\Phi_t$ in Eq. (29) that $\Phi_0 = \log |\mathcal{F}|$ and

$$\Phi_T \geq \log \left( \exp \left( -\sum_{s=1}^{T} l_s(f^\star, x_s, a_s) \right) \right) = 0.$$

**Regret in $t \in \mathcal{T}_2$ with $a_t \notin \mathcal{B}_t$.** Next, for $a_t \notin \mathcal{B}_t$, according to the definition of $\mathcal{A}_t$, there exists a function $\tilde{f} \in \mathcal{F}_t$ such that $\tilde{f}(x_t, a_t) \geq \tilde{f}(x_t, a')$ for any $a' \in \mathcal{A}$. This implies that

$$f_t(x_t, a_t) \overset{(i)}{\leq} f_t(x_t, b_t) - \frac{2}{T} \overset{(ii)}{\leq} \tilde{f}(x_t, b_t) + \frac{1}{T} - \frac{2}{T} = \tilde{f}(x_t, a_t) - \frac{1}{T},$$

where in $(i)$ we use the definition of $\mathcal{B}_t$ in Eq. (28), and in $(ii)$ we use the definition of $\mathcal{T}_2$ that for any $f \in \mathcal{F}_t$, $|f(x_t, b_t) - f_t(x_t, b_t)| \leq 1/T$. Therefore, in those rounds with $\sigma_t = 0$, $t \in \mathcal{T}_2$ and $a \notin \mathcal{B}_t$, we always have $Z_t = 1$ ($Z_t$ is defined in Eq. (27)), which implies that

$$\mathbb{E}\left[ \sum_{t \in \mathcal{T}_2} \mathbb{I}[a_t \notin \mathcal{B}_2] \left( f^\star(x_t, a_t^\star) - f^\star(x_t, a_t) \right) \right] \leq \mathbb{E}\left[ \sum_{t=1}^{T} \mathbb{I}[\sigma_t = 0] Z_t \right] \leq d_{\mathrm{elu}}(1/T),$$

where the last inequality uses Lemma H.2.

Finally we combined these these bounds in $\mathcal{T}_1$ and $\mathcal{T}_2$ together, and obtain that

$$\mathbb{E}\left[\sum_{t=1}^{T}\mathbb{I}[\sigma_t = 0](f^{\star}(x_t, a_t^{\star}) - f^{\star}(x_t, a_t))\right]$$

$$= \mathbb{E}\left[\sum_{t\in\mathcal{T}_1}(f^{\star}(x_t, a_t^{\star}) - f^{\star}(x_t, a_t))\right] + \mathbb{E}\left[\sum_{t\in\mathcal{T}_2}(f^{\star}(x_t, a_t^{\star}) - f^{\star}(x_t, a_t))\right]$$

$$\leq 7Ad_{\text{elu}}(1/T) + \mathbb{E}\left[\sum_{t\in\mathcal{T}_2}\mathbb{I}[a_t \in \mathcal{B}_2]\left(f^{\star}(x_t, a_t^{\star}) - f^{\star}(x_t, a_t)\right)\right]$$

$$+ \mathbb{E}\left[\sum_{t\in\mathcal{T}_2}\mathbb{I}[a_t \notin \mathcal{B}_2]\left(f^{\star}(x_t, a_t^{\star}) - f^{\star}(x_t, a_t)\right)\right]$$

$$\leq 7Ad_{\text{elu}}(1/T) + \left(A + \gamma\sqrt{1 + \sum_{t=1}^{T}\sigma_t^2}\right)\log|\mathcal{F}| + 3 + d_{\text{elu}}(1/T)$$

With our choice of $\gamma = \sqrt{\frac{8A}{\log|\mathcal{F}|}}$, we have

$$\mathbb{E}\left[\sum_{t=1}^{T}\mathbb{I}[\sigma_t = 0](f^{\star}(x_t, a_t^{\star}) - f^{\star}(x_t, a_t))\right] = \mathcal{O}\left(\sqrt{A\log|\mathcal{F}|\sum_{t=1}^{T}\sigma_t^2} + A(d_{\text{elu}} + \log|\mathcal{F}|)\right).$$

$\square$

Finally, combining the regret bound for rounds with $\sigma_t = 1$ and $\sigma_t = 0$ together, we can prove Theorem H.1.

**Proof of Theorem H.1.** We decompose the regret

$$R_T = \sum_{t=1}^{T}\left(\max_{a\in\mathcal{A}}f^{\star}(x_t, a) - f^{\star}(x_t, a_t)\right)$$

$$= \sum_{t=1}^{T}\mathbb{I}[\sigma_t = 0]\left(\max_{a\in\mathcal{A}}f^{\star}(x_t, a) - f^{\star}(x_t, a_t)\right) + \sum_{t=1}^{T}\mathbb{I}[\sigma_t = 1]\left(\max_{a\in\mathcal{A}}f^{\star}(x_t, a) - f^{\star}(x_t, a_t)\right).$$

According to Lemma H.1, we have

$$\mathbb{E}\left[\sum_{t=1}^{T}\mathbb{I}[\sigma_t = 1]\left(\max_{a\in\mathcal{A}}f^{\star}(x_t, a) - f^{\star}(x_t, a_t)\right)\right] = \mathcal{O}\left(\sqrt{A\log|\mathcal{F}|\left(1 + \sum_{t=1}^{T}\sigma_t^2\right)}\right),$$

and according to Lemma H.3, we have

$$\mathbb{E}\left[\sum_{t=1}^{T}\mathbb{I}[\sigma_t = 0]\left(\max_{a\in\mathcal{A}}f^{\star}(x_t, a) - f^{\star}(x_t, a_t)\right)\right] = \mathcal{O}\left(\sqrt{A\log|\mathcal{F}|\sum_{t=1}^{T}\sigma_t^2} + A(d_{\text{elu}} + \log|\mathcal{F}|)\right).$$

Summing these two together, we obtain that

$$\mathbb{E}[R_T] = \mathcal{O}\left(\sqrt{A\log|\mathcal{F}|\sum_{t=1}^{T}\sigma_t^2} + A(d_{\text{elu}} + \log|\mathcal{F}|)\right).$$

$\square$

