# OpenReview forum: "How Does Variance Shape the Regret in Contextual Bandits?"
_NeurIPS.cc/2024/Conference — NeurIPS 2024 poster_

### Official Review · Reviewer_CuA6 · 2024-06-23

**Soundness:** 3
**Presentation:** 2
**Contribution:** 3
**Rating:** 5
**Confidence:** 3

**Summary:**

*** In line 12 there is a comment with acronyms of one of the authors, for the AC's consideration if it is a desk reject ***

The paper studies the setting of adversarial contextual MAB under the assumption of access to a realizable general function class for approximating the context-dependent rewards.
The authors prove both upper and lower regret bounds, that are variance dependent, i.e., dependent on the cumulative data variance rather than on $T$ which is the number of episodes. The authors present the regent bounds in three different settings (1) for strong adversary, (2) for weak adversary, (3) where learning from function class.
In all the cases, their bounds are also dependent on the Eluder dimension. They present a matching lower bound for each setting.
The presented algorithm is adapted from SquareCB of Foster and Rakhlin (2020), but additionally maintains a confidence function set, to learn faster when the functions in the confidence set have larger disagreement.

**Strengths:**

1.	Variance-dependent regret bound is an interesting benchmark in RL. An application of it to contextual bandits is nice and worth the community's attention.
2.	The work is extensive – the authors fully analyze three different settings, and prove both upper and lower bounds.

**Weaknesses:**

1.	The related literature review is unclear.  In the presented previous results, there should be a clear separation between variance-dependent bounds and minimax regret bounds.

2.	The difference between the three settings is unclear to me. I would expect the authors to use the accepted terms, i.e., oblivious and adaptive adversary. Moreover, to be comparable to previous literature, I think the most interesting case is strong adversary + learning from function class.

3.	I do not see why the eluder diminution is necessary in all the bounds. It would be appreciated if the authors could provide an intuitive explanation.

4.	It seems that the work over a confidence set of functions implies a running time complexity of $|F|$ that is not discussed.

5.	The paper seems to be written at the last minute, as there are comments left inside the paper. See for additional example line 185 "(our contribution)".

6.	The use of the Hellinger-Eluder dimension is given without any justification or an appropriate citation of previous works. The classic Eluder dimension introduced by Russo and Van-Roy is defined with respect to the $\ell_2$ norm only. If the authors are using other versions of it, they have to cite related work, if such exists, or justify the used guarantees themselves.  In this case, [1] presents a version of the Eluder dimension to bounded metrics, that also holds to the squared Hellinger distance.


[1] Eluder-based regret for stochastic contextual MDPs, Levy et al. 2024.

**Questions:**

1.	Please provide a specific comparison of your results with the known variance-dependent bounds in previous literature.

2.	What do you mean by $d_{elu}(0)$ ? For my understanding, the Eluder dimension is meaningful only for $\alpha \in (0,1] $.

3.	Please refer to the weaknesses.

**Limitations:**

Mentioned in the "Weaknesses" section.

---

> ### Author Rebuttal · Authors · 2024-08-06
>
> Thank you very much for providing these useful feedback and questions.
>
> Weakness:
>
> - *Unclear related work.*
>
> We will improve the related work section based on the suggestion.
>
> - *Difference between the three settings.*
>
> The separation between strong and weak adversary is different from that between oblivious and adaptive adversary. The terms 'strong' and 'weak' come from literature studying adversarial corruption in contextual bandits (e.g., He et al., 2022). In our case, 'strong adversary' decides the variance after seeing the decision of the learner, while 'weak adversary' decides it before. The separation between 'function class' and `model class' has been studied in previous works like [Wang et al., 2024]. The latter provides  the full distributional information given context and decision, while the former only the mean.
>
> [He et al., 2022] Nearly Optimal Algorithms for Linear Contextual Bandits with Adversarial Corruptions.
>
> *The most interesting case is strong adversary + learning from function class.*
>
> We argue that weak adversary and learning from model class are also of interest. In particular, as we showed, against weak adversary, one may obtain a better regret bound. Also, with a model class, the learner is able to generalize the learned knowledge beyond the 'mean' domain, which is the main consideration in the field of 'distributional RL.' The benefit of learning with a model class is also discussed in, e.g., [Wang et al., 2023, 2024].
>
> - *Why the eluder dimension is necessary in all the bounds?*
>
> Some intuition has been mentioned in Line 41-54. While in the non-contextual case (e.g., multi-armed bandits), eluder dimension is not necessary, it is necessary in the contextual case as we showed. Consider the case where the function set size |F| >> 2 and the action number A=2. Suppose that the ground truth function $f^\star$ always chooses the better action under every context. Every time, the adversary chooses the context so that one of the functions choose an action that disagrees with all other $|F|-1$ functions' choices. This way, to figure out $f^\star$, the regret must scale with some ``disagreement coefficient'' of $|F|$, instead of just $A$. The eluder dimension takes the role of the disagreement coefficient.
>
> - *Running time O(|F|) not discussed.*
>
> We will clarify this in our revision. Indeed, in this work, we only focus on the statistical complexity but not the computational complexity.
>
> - *Comments left inside the paper. Additional example: line 185 "(our contribution)".*
>
> We are sorry for the mistake. The label ``(our contribution)'' is actually not a mistake. It refers to the equation right below Line 179.
>
> - *The use of the Hellinger-Eluder dimension is given without any justification or an appropriate citation of previous works. The classic Eluder dimension introduced by Russo and Van-Roy is defined with respect to the norm only. If the authors are using other versions of it, they have to cite related work, if such exists, or justify the used guarantees themselves. In this case, [1] presents a version of the Eluder dimension to bounded metrics, that also holds to the squared Hellinger distance.*
>
>  The use of the Hellinger-Eluder dimension is given without any justification or an appropriate citation of previous works. The classic Eluder dimension introduced by Russo and Van-Roy is defined with respect to the $\ell_2$norm only. If the authors are using other versions of it, they have to cite related work, if such exists, or justify the used guarantees themselves. In this case, [1] presents a version of the Eluder dimension to bounded metrics, that also holds to the squared Hellinger distance.
>
>
> Thanks for bringing to our attention the missing citation. Our motivation is the same as [1] to generalize the eluder dimension to general divergence.
>
> [1] Eluder-based regret for stochastic contextual MDPs, Levy et al. 2024.
>
>
> Question:
> - *Please provide a specific comparison of your results with the known variance-dependent bounds in previous literature.*
>
> Previous work on variance-dependent contextual bandits mostly focused on linear contextual bandits (see Zhao et al., 2023 and the related work therein).  Literature on variance-dependent contextual bandits with general function approximation is scarce. [Wang et al., 2024] studied the setting same as our Section 6. Their result is listed in our Table 1. While they conjectured that their bound could be improved, we disproved it by showing a matching lower bound. The work of [Wei et al., 2020] also investigated second-order bounds for contextual bandits, but they focused on the agnostic setting (while we focused on the realizable setting). Though their algorithm can be applied to our setting, the regret is highly sub-optimal. In general, their result is incomparable to ours.
>
> - *What does $d_{\mathrm{elu}}(0)$ mean?*
>
> In fact, $d_{\mathrm{elu}}(\alpha)$ remains meaningful when $\alpha=0$ (see our Definition 2.1, and simply set $\alpha=0$). For example, a $d$-dimensional linear function class has $d_{\mathrm{elu}}(0)=d$.

---

> ### Comment · Reviewer_CuA6 · 2024-08-08
>
> I thank the authors for their response.
> Not sure I understand their response related to the question about $d_{eluder}(0)$. As far as I understand the Eluder dimension, $\alpha$ should be an error parameter. Are you using $1-\alpha$? Would be happy if the authors could elaborate more.
> Besides that I have no further questions.

---

> > ### Author Response · Authors · 2024-08-09
> >
> > We provide a further explanation below for $d_{elu}(0)$.
> >
> > Our definition of eluder dimension (Definition 2.1) follows that in Russo and Van Roy (2014). See their Definition 3 and 4. Note that their $\epsilon$ is our $\alpha$.
> >
> > By the definitions, $d_{elu}(0)$ is well-defined as long as the concept of ``$0$-dependent'' is well-defined (see Russo and Van Roy's Definition 3). By their Definition 3, an action $a$ is $0$-dependent with respect to $\mathcal{F}$ on $\{a_1, \ldots, a_n\}$ if any pairs $f, \tilde{f}\in \mathcal{F}$ satisfying $f(a_i)=\tilde{f}(a_i)$ for all $i\in\{1,2,\ldots, n\}$ also satisfies $f(a)=\tilde{f}(a)$.
> >
> > Take the $d$-dimensional linear function class as an example, i.e., $\mathcal{F}$ consists of functions of the form $f_\theta(a)=\theta^\top a$. We demonstrate it through $d=4$. Below we show that $a=(2,3,4,0)$ is $0$-dependent on $\{a_1, a_2, a_3\}$ where $a_1=(1,0,0,0)$, $a_2=(0,1,0,0)$, and $a_3=(0,0,1,0)$. To verify this, assume that $f=f_\theta$ and $\tilde{f} = f_{\tilde{\theta}}$ agree on $\{a_1, a_2, a_3\}$, i.e., $\theta^\top a_i = \tilde{\theta}^\top a_i$ for all $i=1,2,3$. By our choices of $a_1, a_2, a_3$, it must be that $\theta_i = \tilde{\theta}_i$ for all $i=1,2,3$. This further implies $\theta^\top a = \tilde{\theta}^\top a$. We thus have shown that $a$ is $0$-dependent on $\{a_1, a_2, a_3\}$.
> >
> > It can be verified that for any finite function class $\mathcal{F}$, $d_{elu}(0)\leq |\mathcal{F}|$ always holds, though it is possible for some infinite function class to have $d_{elu}(0)=\infty$.
> >
> > Note that by the definition of eluder dimension, $d_{elu}(\alpha)$ is decreasing with $\alpha$. That is the reason why we choose to state with $d_{elu}(0)$ in Theorem 4.1, 6.2, and 6.3. In these lower bounds, our construction ensures $d_{elu}(0)\leq d$, which implies $d_{elu}(\alpha)\leq d$ for all $\alpha\geq 0$. This ensures that our lower bound simultaneously hold for any choice of $\alpha$.
> >
> >
> > [Russo and Vay Roy, 2014] Eluder Dimension and the Sample Complexity of Optimistic Exploration. (link: https://web.stanford.edu/~bvr/pubs/Eluder.pdf).

---

> > > ### Comment · Reviewer_CuA6 · 2024-08-10
> > >
> > > I thank the authors for their response and have no further questions.

---

### Official Review · Reviewer_u4Bf · 2024-07-09

**Soundness:** 3
**Presentation:** 4
**Contribution:** 3
**Rating:** 6
**Confidence:** 3

**Summary:**

This paper considers contextual bandits with function approximation, and studies the influence of the variance information on the regret bound. It studies three regimes: weak adversary where the variance is revealed before each time step; strong adversary where the variance depends on the chosen action at each time step; and learning with a model class where the reward follows a Gaussian distribution with the mean and the variance depending on the context and the arm. By incorporating the variance, both the upper and lower bounds of the regret are refined, subject to some gaps in some scenarios.

**Strengths:**

- The proposed problem is interesting and the result also answers the conjecture that $\tilde{O}(\sqrt{A\Lambda \log|\mathcal{F}|}+A\log |\mathcal{F}|)$  is not true, when we incorporate the variance into the algorithm design. The eluder dimension plays an important role in this bound and has been used to characterize the bounds.
- The paper studies three different cases and provides sufficient insights in this line of problems. While there are some gaps between the upper and lower bounds, the discussions are thorough and can lead to further research.

**Weaknesses:**

See the questions section

**Questions:**

- Can the authors summarize the main technical novelty used in the paper?
- Concerning the third case, learning with the model class, the authors indicate that the variance information is not required as input in line 105. Does it mean that the variance, given the model, the context and the action, the variance $\sigma_{M_t}^2(x_t,a)$ is revealed, as presented in line 7 in Algorithm 3? If we only assume that the variance is unknown but it only depends on the context and the arm, can the algorithm be modified such that it can work with estimating the variance on the fly, like [1]?
- The  lower bound result Theorem 6.2 is obtained via instances with the same variance. While the lower bound is indeed a minimax lower bound, as this paper is considering the role of the variance, is it possible to derive a lower bound for the case where the variances are heterogeneous?

[1] Jourdan, M., Rémy, D., & Emilie, K. Dealing with unknown variances in best-arm identification. International Conference on Algorithmic Learning Theory. PMLR. 2023.

---

> ### Author Rebuttal · Authors · 2024-08-07
>
> - *Can the authors summarize the main technical novelty used in the paper?*
>
> We have the following technical contributions.
>
> 1. Algorithm design through checking disagreement: Our algorithms in Sec 4 and Sec 6 decide whether to use inverse gap weighting based on the degree of disagreement in the function confidence set. Such a design is, to our knowledge, crucial to obtain the tight bound and not seen in previous work. The analysis framework for this algorithm is also new.
>
> 2. Refined online regression oracle: The choice of the online regression oracle (i.e., Prod) in Sec 4 and its associated analysis are non-trivial. The usefulness of such an online regression oracle has not been discovered before.
>
> 3. In Sec 6, we adapt the inverse-gap weighting technique to learning from model class. This was mentioned as a future work by [Wang et al., 2024]. In this case, we identified the Gaussian-noise condition under which tight bound can be obtained without revealing variance. We believe this is an important step towards resolving the general case.
>
> 4. In Sec 6, we identify the important difference between the achievable bounds using $\sum_{t=1}^T
> \sigma(x_t,a_t)^2$ and $\sum_{t=1}^T \max_a \sigma(x_t,a)^2$ as the total variance measure. This answers the open questions in [Wang et al., 2024]. Similarly, in Sec 4 and 5, we identify the difference between the achievable bounds under strong and weak adversary.
>
>
>
>
> - *Concerning the third case, learning with the model class, the authors indicate that the variance information is not required as input in line 105. Does it mean that the variance, given the model, the context and the action, the variance $\sigma_{M_t}^2(x_t,a)$ is revealed, as presented in line 7 in Algorithm 3?*
>
> Yes, the variance will be revealed.
>
> - *If we only assume that the variance is unknown but it only depends on the context and the arm, can the algorithm be modified such that it can work with estimating the variance on the fly, like [1]?*
>
> It would be hard to estimate in general. For instance, if the context space is super huge and no context is repeated twice, then since there is no model class, then there is no way to generalize the variance information since each time, the variance can be arbitrary. So, estimation of the variance on the fly is not possible. On the other hand, if the context space is small, estimating the variance might be possible, but we suspect that would render the regret bound depending on the size of the context space.
>
> - *The lower bound result Theorem 6.2 is obtained via instances with the same variance. While the lower bound is indeed a minimax lower bound, as this paper is considering the role of the variance, is it possible to derive a lower bound for the case where the variances are heterogeneous?*
>
> Thanks for your suggestion. Our lower bounds can be adapted to any collection of variances (with no ordering) through the following trick. The idea is that for any given collection of variance, we can divide the variances with doubling scales into $\log T$ categories. For the scale with the largest cumulative variance, we put the variances of this scale to all appear first and apply our lower bound to these steps. By pigeonhole principle, our lower bound is worsened by a rate of at most $\log T$. Currently, our lower bounds do not extend to cases where the sequence of variance is given. It is an interesting direction for future research.
>
> [1] Jourdan, M., Rémy, D., & Emilie, K. Dealing with unknown variances in best-arm identification. International Conference on Algorithmic Learning Theory. PMLR. 2023.

---

> > ### Comment · Reviewer_u4Bf · 2024-08-11
> >
> > Thank the authors for their response! Please consider incorporating the discussions of lower bound under the heterogeneous variances scenario into their manuscript.
> >
> > Regarding the question that the variances are revealed, I still believe this setup might be strong. I understand that when the context space is large, the estimation of the variance may not be possible. From another point of view, this indicates the knowledge of the variance is unavoidable in this work. Therefore, can the authors give practical scenarios where the variance information is given to justify their setup?

---

> ### Author Response · Authors · 2024-08-13
>
> Thanks for the feedback. First, we agree that the assumption in Section 4 that variance being revealed is not ideal, and our ultimate goal is to remove it (some initial attempts have been made in Appendix G). At the current stage, however, this assumption is necessary for our technique, and we hope that the our technique could inspire future work that obtains variance-agnostic result.
>
> Still, there are cases where the learner is able to obtain variance information / variance estimation before making decisions, and the use of our algorithm can be justified.
>
> For example, our algorithm works well for the case where the variance is upper bounded by a fixed quantity in all rounds, and the variance is much smaller than the reward range. This could simulate recommendation systems where the reward comes from user rating. Let's say the range of the rating is 0 to 10, but the user feedback has some noise and it is modeled to be within (mean - 1.5, mean + 1.5). In this case, the sigma in our algorithm can be chosen as 3/10, which significantly improve the existing regret bound by variance-unaware algorithms (e.g., SquareCB).
>
> It is also possible that the learner can perform context-aware variance estimation, and plug it into our algorithm. For example, in clinical trials, the context may include information like the type of the drug and the age of the patient. We could create a finite number of categories for these contexts, and estimate the variance for each of them using historical data. This way, we can obtain variance estimation for given contexts before making decisions, and then apply our algorithm.

---

> > ### Comment · Reviewer_u4Bf · 2024-08-13
> >
> > Thank the reviewer for their response! I do not any further questions.

---

### Official Review · Reviewer_W3F3 · 2024-07-12

**Soundness:** 3
**Presentation:** 3
**Contribution:** 3
**Rating:** 7
**Confidence:** 4

**Summary:**

This paper studies contextual bandits with general function approximation, emphasizing the impact of variance information. It proves lower bounds dependent on the Eluder dimension and variance information in both strong and weak adversary settings. It also proposes algorithms for both cases. When the adversary is weak, the proved regret upper bound is optimal, otherwise, there is a $\sqrt{d}$ gap from optimal. The paper also studies the learning problem when the function class contains the distributional information and provides optimal regret bound.

**Strengths:**

1. The paper studies contextual bandits with general function approximation. The proved lower bound dependent on the Eluder dimension and variance information is very novel.
2. The paper provides a comprehensive analysis for different settings, e.g, weak/strong adversaries, and distributional settings. The proved results are insightful and optimal in most cases.
3. The presentation is very clear, including comparison with previous works.

**Weaknesses:**

1. In the known variance section, this paper only focuses on a regime where $A \le d_{elu} \le \sqrt{AT}$. The case $d_{elu} \le A$ is also very important, especially when the action set is large and function approximation benefits. In such cases, the proved upper bound and lower bound do not match, so the contribution in this section is a bit overclaimed. (matching upper bound when variance is known)
2. Some missing related works.
More works on variance-dependent results of heteroscedastic bandits:

[1] Zhou et al. Nearly minimax optimal reinforcement learning for linear mixture markov decision processes, Colt 2021

[2] Zhou and Gu. Computationally Efficient Horizon-Free Reinforcement Learning for Linear Mixture MDPs Neurips 2022

[3] Zhao et al. Optimal Online Generalized Linear Regression with Stochastic Noise and Its Application to Heteroscedastic Bandits

3. This description of "checking disagreement" (Line 166) is not very accurate. A similar definition of equation (4) has been used in previous works as weights ([4] [5]), bonus functions ([6][7]), and selection rules ([8][9]). I suggest adding more discussions on this.

[4] Agarwal et al. Vo q l: Towards optimal regret in model-free rl with nonlinear function approximation. Colt 2023

[5] Ye et al.  Corruption-robust algorithms with uncertainty weighting for nonlinear contextual bandits and markov decision processes. ICML2023

[6] Di et al. Pessimistic Nonlinear Least-Squares Value Iteration for Offline Reinforcement Learning, ICLR 2024

[7] Huang et al. Horizon-Free and Instance-Dependent Regret Bounds for Reinforcement Learning with General Function Approximation AISTATS

[8] Gentile et al. Fast Rates in Pool-Based Batch Active Learning ICML 2024

[9] Zhao et al. A Nearly Optimal and Low-Switching Algorithm for Reinforcement Learning with General Function Approximation

4. Some minor errors: Line 12 worst case over what?

In the proof of Lemma E.5, there is an abusive use of $\epsilon$, it represents both the Gaussian noise and a changeable parameter for $\sigma$.

**Questions:**

1. Is it possible to prove a variance-dependent upper bound with $d_{elu}$ dependence in the first term of Theorem 4.2?
2. What difficulty did you meet when trying to recover Zhao et al. (2023)’s result for unknown variance?

**Limitations:**

The limitations and societal impact of this work have been addressed.

---

> ### Author Rebuttal · Authors · 2024-08-06
>
> Thank you very much for providing these useful feedback and questions.
>
> Answers to Weaknesses:
> - *Mismatch upper and lower bound for large action space cases*
>
> We have such results for cases where $d_{\mathrm{elu}} < A$ (an upper bound $\tilde{\mathcal{O}}(d_{\mathrm{elu}}\sqrt{T\sigma^2\log|\mathcal{F}|})$). Please view Theorem 5.2 in our paper for more details. We admit that this upper bound does not match the provided lower bounds, and we will change our presentation accordingly in the next version.
>
> - *Some missing related works. More works on variance-dependent results of heteroscedastic bandits:*
>
> Thanks for pointing out these related references. We will add them in the next version of our paper.
>
> - *This description of "checking disagreement" (Line 166) is not very accurate. A similar definition of equation (4) has been used in previous works as weights ([4] [5]), bonus functions ([6][7]), and selection rules ([8][9]). I suggest adding more discussions on this.*
>
> Here 'check agreement' means checking whether there exists an action where two functions in the function set deviated a lot. Thanks for pointing out these references. We will add them in the next version of our paper.
>
> - *Some minor errors: Line 12 worst case over what?*
>
> These are words meant to be deleted. We will delete them in our next version.
>
> Answers to Questions:
> - *Is it possible to prove a variance-dependent upper bound with dependence in the first term of Theorem 4.2?*
>
> Yes, we can obtain such an upper bound: $\tilde{\mathcal{O}}(d_{\mathrm{elu}}\sqrt{T\sigma^2\log|\mathcal{F}|})$, which serves as a special case of Theorem 5.2 in our paper.
>
> - *What difficulty did you meet when trying to recover Zhao et al. (2023)’s result for unknown variance?*
>
> The main difficulty we met is that we are dealing with bandit with function approximation in our paper, compared to the setting of linear bandit in Zhao et al. (2023). Technically, to prove our results, we are required to apply the general Freedman's inequality, rather than the Freedman's inequality specialized for linear setting.

---

> > ### Comment · Reviewer_W3F3 · 2024-08-11
> >
> > Thanks for your useful responses. I will keep my score.

---

### Official Review · Reviewer_aAKW · 2024-07-12

**Soundness:** 3
**Presentation:** 3
**Contribution:** 3
**Rating:** 7
**Confidence:** 3

**Summary:**

This paper studies the problem of obtaining variance-aware regret bounds for realizable contextual bandits with general function approximation. They show that, despite intuition suggested by prior works, there is an unavoidable dependence on the Eluder dimension in getting variance-aware bounds. They investigate several different settings (depending on the strength of the adversary's environment design and the agent's knowledge of distributional assumptions) and show minimax lower bounds each involving the Eluder dimension. They also provide matching regret upper bounds under various assumptions on knowledge of distributions or variances.

**Strengths:**

* The problem is well-motivated, the surrounding literature is well-explained, and the authors take care to explain why the main message is surprising, while also explaining the intuition for the unavoidable dependence on Eluder dimension.
* The algorithms and needed modifications are well-explained with authors taking care to make simplifying assumptions to enhance presentation and understanding.
* Along the way, some new technicalities had to be developed such as employing a square regression oracle with variance-based regret employing a more complicated action selection scheme based on checking discerning actions.

**Weaknesses:**

* The main weakness is that the only upper bound results matching the lower bounds require some knowledge of the variance at every single round or under a stronger model class assumption (which then involves picking up a dependence on the size of the model class of variances).
* There could be more discussion of other function classes beyond the generic finite regressor class ${\cal F}$. For instance, how do the lower bounds and results here about variance-based bounds and Eluder dimension compare with the understanding for linear function classes, high-dimensional classes (like those mentioned for SquareCB in Foster and Rakhlin, 2020), neural network, or non-parametric classes? Do the lower bounds hold in greater generality?

**Questions:**

Please refer to strengths and weaknesses above.

**Limitations:**

No braoder impact concerns.

---

> ### Author Rebuttal · Authors · 2024-08-06
>
> Thank you very much for providing these useful feedback and questions.
>
> Answers to Weaknesses:
> -  *The main weakness is that the only upper bound results matching the lower bounds require some knowledge of the variance at every single round or under a stronger model class assumption (which then involves picking up a dependence on the size of the model class of variances).*
>
> We obtained matched upper bound and lower bound for the variance-aware cases in our paper. For agnostic variance cases, we have both upper bound and lower bound with only a factor of $\sqrt{d_{\mathrm{elu}}}$ between them. Even with this gap, the dependence over the variances $\sqrt{\sum_{t=1}^T \sigma_t^2}$ is tight, and our bounds are sufficient to show the separation between variance aware and agnostic variance cases.
>
> -  *There could be more discussion of other function classes beyond the generic finite regressor class
> . For instance, how do the lower bounds and results here about variance-based bounds and Eluder dimension compare with the understanding for linear function classes, high-dimensional classes (like those mentioned for SquareCB in Foster and Rakhlin, 2020), neural network, or non-parametric classes? Do the lower bounds hold in greater generality?*
>
> Even though we did not discuss about infinite function class in our paper. But our algorithms and theoretical guarantee can generalize to infinite function classes. For example, for linear classes, using the similar structure of Algorithm 1 in our paper and some proper regression oracle for linear class similar as the one in Foster and Rakhlin (2021), we can obtain an upper bound of $\tilde{\mathcal{O}}(\sqrt{dA\sum_{t=1}^T\sigma_t^2})$. For generalized linear classes, we can get an upper bound of $\tilde{\mathcal{O}}(\sqrt{T\sum_{t=1}^T\sigma_t^2})$. And for other classes including neural networks and nonparametric class, we can get similar variance-dependent bound as long as there are some proper variance-aware regression oracles.
>
> As for the lower bound part, since our lower bound is constructive, i.e. we show that there exists a function class $F$ which achieves the lower bound, we do not know how we can generalize our results to other function classes. But to obtain an instance-dependent lower bound for all possible function class is an interesting future research direction.

---

> > ### Comment · Reviewer_aAKW · 2024-08-13
> >
> > Thank you for the response. It seems the strong adversary case actually covers the unknown variance situation so the results of the paper are stronger than I initially thought, even though there is higher dependence on $d$. As such, I am satisfied by the discussion and raise my score.

---

### Decision · Program_Chairs · 2024-09-25

**Decision:**

Accept (poster)

**Comment:**

This paper makes a solid contribution to the important problem of adapting to the variance in contextual bandits.